# VAV2 signaling promotes regenerative proliferation in both cutaneous and head and neck squamous cell carcinoma

L. Francisco Lorenzo-Martín [1,2,3], Natalia Fernández-Parejo[1,2,3,16], Mauricio Menacho-Márquez[1,2,3,15,16], Sonia Rodríguez-Fdez [1,2,3], Javier Robles-Valero[1,2,3], Sonia Zumalave [4], Salvatore Fabbiano [1,2,15], Gloria Pascual [5,6], Juana M. García-Pedrero[3,7], Antonio Abad[1,2,3], María C. García-Macías [1,2,3], Nazareno González [1,2], Pablo Lorenzano-Menna[8,9], Miguel A. Pavón[10,11], Rogelio González-Sarmiento[1,2,12], Carmen Segrelles[3,13], Jesús M. Paramio [3,13], José M. C. Tubío[4], Juan P. Rodrigo [3,7], Salvador A. Benitah [5,6,14], Myriam Cuadrado [1,2,3] & Xosé R. Bustelo [1,2,3✉]

Regenerative proliferation capacity and poor differentiation are histological features usually linked to poor prognosis in head and neck squamous cell carcinoma (hnSCC). However, the pathways that regulate them remain ill-characterized. Here, we show that those traits can be triggered by the RHO GTPase activator VAV2 in keratinocytes present in the skin and oral mucosa. VAV2 is also required to maintain those traits in hnSCC patient-derived cells. This function, which is both catalysis- and RHO GTPase-dependent, is mediated by c-Myc- and YAP/TAZ-dependent transcriptomal programs associated with regenerative proliferation and cell undifferentiation, respectively. High levels of *VAV2* transcripts and VAV2-regulated gene signatures are both associated with poor hnSCC patient prognosis. These results unveil a druggable pathway linked to the malignancy of specific SCC subtypes.

[1] Centro de Investigación del Cáncer, CSIC-University of Salamanca, 37007 Salamanca, Spain. [2] Instituto de Biología Molecular y Celular del Cáncer, CSIC-University of Salamanca, 37007 Salamanca, Spain. [3] Centro de Investigación Biomédica en Red de Cáncer (CIBERONC), CSIC-University of Salamanca, 37007 Salamanca, Spain. [4] Center for Research in Molecular Medicine and Chronic Diseases (CiMUS), University of Santiago de Compostela, 15782 Santiago de Compostela, Spain. [5] Institute for Research in Biomedicine, 33011 Barcelona, Spain. [6] The Barcelona Institute of Science and Technology, Barcelona 33011, Spain. [7] Hospital Universitario Central de Asturias, Oviedo University, 33011 Oviedo, Spain. [8] Laboratory of Molecular Oncology and National University of Quilmes, Buenos Aires B1876BXD, Argentina. [9] National Council of Scientific and Technical Research (CONICET), National University of Quilmes, Buenos Aires B1876BXD, Argentina. [10] Institut Català d'Oncologia, 08908 L'Hospitalet de Llobregat, Spain. [11] Centro Biomédica de Investigación en Red de Enfermedades Respiratorias (CIBERESP), 08908 L'Hospitalet de Llobregat, Spain. [12] Instituto de Investigación Biomédica de Salamanca, 37007 Salamanca, Spain. [13] Centro de Investigaciones Energéticas, Medioambientales y Tecnológicas, 28040 Madrid, Spain. [14] Catalan Institution for Research and Advanced Studies (ICREA), 33011 Barcelona, Spain. [15] Present address: Instituto de Inmunología Clínica y Experimental de Rosario (IDICER, CONICET-UNR). Facultad de Ciencias Médicas Universidad Nacional de Rosario (M.M.-M.) and CellPress editorial office (S.F.), S2000LRJ Rosario, Argentina. [16] These authors contributed equally: Natalia Fernández-Parejo, Mauricio Menacho-Márquez. ✉email: xbustelo@usal.es

Cutaneous (cSCC) and hnSCCs develop from the stratified epithelia of the skin and the mucosal lining of the upper aerodigestive tract, respectively. These tumors represent nowadays a clinical challenge due to increasing incidence worldwide, aggressiveness, poor prognosis, and limited efficacy of available therapeutic tools. Risk factors for these tumors include ultraviolet ray exposure (in the case of cSCCs), alcohol intake, smoking, and human papillomavirus (HPV) infection (in the case of hnSCCs)[1,2]. These SCCs share histological features similar to those found in the most undifferentiated basal layers of healthy epithelia. However, they can display variable numbers of cells that had undergone both terminal differentiation and growth arrest. This variability is clinically relevant, since tumors with less differentiated features usually exhibit worse disease outcomes[1]. These data suggest that pathobiological programs that regulate regenerative proliferation plus delayed differentiation drive both the tumorigenesis and malignancy of this type of tumors. In line with this, it has been proposed that therapies that elicit the terminal differentiation of cancer cells could be of high utility in hnSCC[1]. To date, however, we still have a poor understanding of the molecular and signaling programs that regulate these malignant traits. Recent work in this area has revealed that nuclear factors such as AP1 and E2F family members, c-MYC, the YAP/TAZ complex, TP63, and ACTL6A can trigger those processes[3–8]. These proteins seem to act in a concerted manner, as exemplified by the presence of composite cis-gene DNA regulatory elements for many of them in the target genes[5,6].

Recent tumor molecular taxonomy studies have revealed that the activation of RHO GTPase-regulated biological processes is a common signaling feature for most SCCs[9]. Consistent with this, the gene encoding the RAC1 GTPase has been found mutated at low frequency in cSCC and hnSCC[10]. High levels of active RAC1 are also found in a large variety of hnSCCs[11,12]. This activation has been linked to the development of chemo- and radio-resistance in patients[12]. It is not presently known whether the role of this GTPase is associated with the regulation of its canonical cytoskeletal-related functions[13,14] and/or to regenerative proliferation in those tumors. However, the observation that the elimination of RAC1 activity reduces the clonogenicity of hnSCC cell lines suggest that this GTPase can be involved in the regulation of the latter pathobiological program[12,15]. Unfortunately, efforts at blocking RAC1 signaling pathways in tumors have failed so far due to the intrinsic undruggability of RHO GTPases and the unwanted side-effects of inhibitors for downstream elements such as the Pak family of serine/threonine kinases[10].

An alternative way to targeting these pathways is by inhibiting the catalytic activity of RHO guanosine nucleotide exchange factors (GEFs), the enzymes that promote the stimulation step of RHO proteins in cells[10]. Supporting this idea, genetic analyses have shown that either the single (Tiam1) or the compound (Vav2 plus Vav3) depletion of some RHO GEFs can abate carcinogen-induced skin tumor formation while maintaining skin homeostasis in mice[16,17]. Catalytically hypomorphic Vav2 knock-in mice also show impaired skin tumorigenesis using the above experimental protocols[18]. Yet, very little is known regarding the specific RHO GEFs that become deregulated in human tumors and the pathways engaged by them[10]. Addressing these issues is not an easy task, given that the RHO GEF family is composed of more than 70 members in humans[10].

In this work, we report that the RHO GEF VAV2 is frequently overexpressed in both cSCC and hnSCC cases. We also demonstrate that this GEF is associated with the engagement of a pathobiological program linked to stem cell-like regenerative proliferation in epithelial cells located in the skin and head and neck areas. Conversely, we show that high levels of endogenous VAV2 are required to maintain the tumorigenic activity of

hnSCC patient-derived cells. Along those lines, we have found that the expression levels of the VAV2 mRNA and VAV2-regulated gene signatures correlate with disease outcome in hnSCC patients.

## Results

**VAV2 mRNA levels correlate with poor prognosis in hnSCC.** Previous data suggested that, out of the 70 RHO exchange factors present in humans, the GEF VAV2 could be potentially involved in both hnSCC and cSCC tumorigenesis. This GEF belongs to the VAV family, a group of GEFs (VAV1, VAV2, VAV3) whose exchange activity is regulated by direct tyrosine phosphorylation events[19,20]. In favor of the connection of VAV2 with this type of tumors, work with human hnSCC cell lines has revealed that this GEF is frequently tyrosine phosphorylated and involved in the stimulation of RAC1 downstream of the epidermal growth factor receptor (EGFR) in a significant number of the interrogated cell lines[11]. Furthermore, the use of catalytically deficient Vav2 knock-in mice has demonstrated that the inhibition of the enzymatic activity of Vav2 impairs both the papilloma and cSCC formation that are typically induced upon the topic administration of the carcinogen 7,12-dimethylbenz(a)anthracene (DMBA) either alone or in combination with the tumor promoter 12-O-tetradecanoylphorbol-13-acetate (TPA)[11]. To investigate this possibility, we first analyzed using in silico techniques the expression of VAV2 transcripts in cSCC and hnSCC patients. To this end, we interrogated four independent gene expression datasets from either cSCC (GEO GSE13355 and GSE45216) or hnSCC (GEO GSE30784 and TCGA; for further information, see both Supplementary Table 1 and "Methods" section) patients. In the case of hnSCC, the TCGA collection harbored samples with (21% of cases) and without information on HPV status. The GEO GSE30784 lacks information on HPV status, although the percentage of HPV⁻ samples has been estimated to be in the 75% range[21] (Supplementary Table 1). These datasets were chosen because: (i) They contained expression data from both healthy and tumor samples. (ii) They included a minimum of 100 samples (to facilitate the generation of statistically robust data). (iii) They included samples with clinical information (e.g., dysplasia, stage of tumor progression, etc.). Using this approach, we found that the VAV2 mRNA is consistently upregulated in human cSCC (Fig. 1a, left panel) and hnSCC (Fig. 1a, second and third panels from left) when compared to healthy tissue samples. In the case of hnSCCs, higher expression is detected in HPV⁻ than in HPV⁺ tumors (Fig. 1a, third panel from left). Moderate overexpression levels of VAV2 transcripts are also found in precancerous conditions such as cutaneous actinic keratosis (Fig. 1a, left panel) and oral epidermal dysplasia (Fig. 1a, second panel from left). However, the elevation in VAV2 transcripts is much less noticeable in skin samples from psoriasis patients (Fig. 1a, left panel). The mouse Vav2 mRNA also shows a progressive increase in expression along the progression of cSCCs (Fig. 1a, right panel) when interrogated in the available GSE21264 gene expression microarray dataset (Supplementary Table 1), indicating that the deregulation of this gene is conserved in SCCs from different species. We did not observe any statistically significant elevation in VAV1, a mRNA encoding a VAV family member that shows a hematopoietic-specific pattern of expression[19,20,22], in any of those pre- and neoplastic stages (Supplementary Fig. 1a). By contrast, the VAV1 mRNA does show a small, although statistically significant enrichment in the case of samples from psoriatic patients (Supplementary Fig. 1a, left panel). These data suggest that the changes seen in VAV2 mRNA levels are probably intrinsic to tumor cells rather than an indirect consequence of the presence of varying amounts of infiltrating hematopoietic cells in

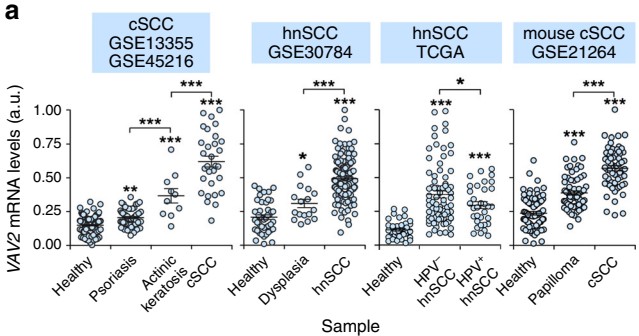

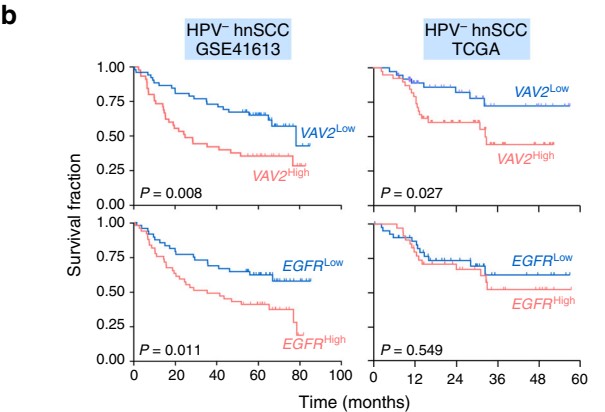

**Fig. 1 VAV2 mRNA levels correlate with poor prognosis in HPV⁻ hnSCC.**
**a** Expression of *VAV2* mRNA in indicated samples (bottom) and gene expression datasets (top). *$P = 0.029$ (dysplasia versus healthy, hnSCC), 0.047 (HPV⁻ versus HPV⁺ hnSCC); **$P = 0.005$ (psoriasis versus healthy, cSCC); ***$P < 0.0001$ (all other analyses) using ANOVA and Tukey's HSD tests (from left to right, $n = 220, 229, 155$, and 273 patient samples, respectively). When data were generated using two datasets at the same time, we used frozen robust multiarray analyses (fRMA) to avoid batch effects as indicated in Methods. Data represent the mean ± SEM. **b** Survival of HPV⁻ hnSCC patients according to the abundance of *VAV2* and *EGFR* mRNAs in tumors in indicated datasets (top). The Mantel–Cox test P value is indicated for each transcript. Source data for this figure are provided as a Source Data file.

the interrogated samples. Taken together, these observations indicate that the *VAV2* mRNA is frequently overexpressed in both cSCCs and HPV⁻ hnSCCs. This is SCC type-specific, because we could not find any statistically significant elevation of *VAV2* transcripts in SCCs from the lung and cervix (Supplementary Fig. 1b, Supplementary Table 1).

We next investigated whether the abundance of *VAV2* transcripts correlated with disease outcome. We focused these analyses on hnSCC, given that no data are available on cSCC due to the typically high survival rates exhibited by these patients[23]. To this end, we carried out in silico analyses using three independent gene expression datasets (TCGA, GEO GSE41613, and GEO GSE42743, Supplementary Table 1). These datasets were chosen because: (i) They contained information of at least 90 independent clinical cases. (ii) They have information on long-term survival, HPV status, and other clinical criteria of patients. These analyses indicated that the levels of the *VAV2* mRNA directly correlated with poor HPV⁻ patient prognosis in these three independent datasets (Figs. 1b and S1c). The stratification power conferred by the *VAV2* mRNA levels is in fact better in terms of statistical significance than that obtained when using the expression level of the transcripts for the EGFR (Fig. 1b, bottom panels), a tyrosine kinase receptor encoded by a gene frequently

amplified in hnSCC (Supplementary Fig. 1d)[2]. It is also better than that conferred by the levels of *ACTL6A* (Supplementary Fig. 1e, left panel), a transcript encoding a hnSCC protumorigenic factor that is upregulated at similar levels between HPV⁺ and HPV⁻ tumors (Supplementary Fig. 1f)[8]. Unlike the case of *ACTL6A* (Supplementary Fig. 1e, two left panels), the diagnostic power of the *VAV2* and *EGFR* mRNA levels is lost when using patient cohorts that have not been classified according to HPV status (Supplementary Fig. 1e, third and fourth panels from left). Further multivariable analyses indicated that the *VAV2* mRNA levels do not show any statistically significant association with tumor stage, smoking status, or therapeutic response.

The upregulation of the *VAV2* mRNA is not associated with gene amplification events (Supplementary Fig. 1d). It is also specific for the *VAV2* locus, because the two genes that flank the *VAV2* gene (*SARDH*, *BRD3*) do not show any consistent variation in expression in the human and mouse tumor datasets interrogated in this study (Supplementary Fig. 1g). The analysis of Pan-Cancer data also indicates that *VAV2* is seldom mutated in hnSCC cases[24–26] (Supplementary Fig. 1d). However, it is worth noting that during the elaboration of this work this locus was found to be mutated at higher rates in hereditary cases of oral SCC[27]. The recurrent gain-of-function mutation found in those tumors (Val[872] to Ile) targets a residue located in the most C-terminal VAV2 SH3 (C-SH3) domain, a region previously shown to be involved in the formation of the intramolecular interactions that maintain the inhibitory structure of all Vav family members when they are not tyrosine-phosphorylated[28]. Taken together, these results indicate that VAV2 is commonly upregulated in specific SCC subtypes. They also indicate that this upregulation correlates with poor prognosis in the case of HPV⁻ hnSCC patients.

**Wild-type VAV2 is overexpressed in hnSCCs.** We next performed immunoprecipitation analyses to monitor the levels of VAV2 in cellular extracts from 83 HPV⁻ hnSCC patient samples. As control, we included lysates from healthy tissues that were in the vicinity of the tumors extracted from the same patients. Consistent with the foregoing in silico mRNA expression analyses, we found that VAV2 is detected at much higher levels in the tumor than in the healthy tissue samples (Fig. 2a, b). Further Western blot analyses revealed that the endogenous VAV2 is also tyrosine phosphorylated in most of those samples (Fig. 2a, c). In addition to the estimation of the protein content present in the tissue extracts using the Bradford's method, the presence of similar amounts of total protein in the tissue lysates used in these immunoprecipitation analyses was rechecked by running aliquots of them in SDS-PAGE gels and, after transferring to nitrocellulose filters, staining the separated proteins with Ponceau S solution (Fig. 2a). We also found using standard immunohistochemical techniques that 41.8% and 14.7% out of a total of 232 independent cases (96.55% of them with a HPV⁻ status) of hnSCC samples that were surveyed display medium to high levels of VAV2 immunoreactivity, respectively (Fig. 2d). This expression is concentrated in the cancer cells rather than in the surrounding stromal components (Fig. 2d), further demonstrating that the upregulation of VAV2 is a cancer cell-intrinsic phenomenon. VAV2 staining does not significantly correlate with the differentiation grade in our patient cohort ($P < 0.86$). Nevertheless, it is worth mentioning that VAV2 staining was consistently absent in all terminal differentiation areas of the samples (e.g., keratin pearls) even in the case of VAV2-positive tumors (Fig. 2e, asterisk).

The above immunohistochemical studies also revealed that VAV2 is detected in the basal but not in the more differentiated

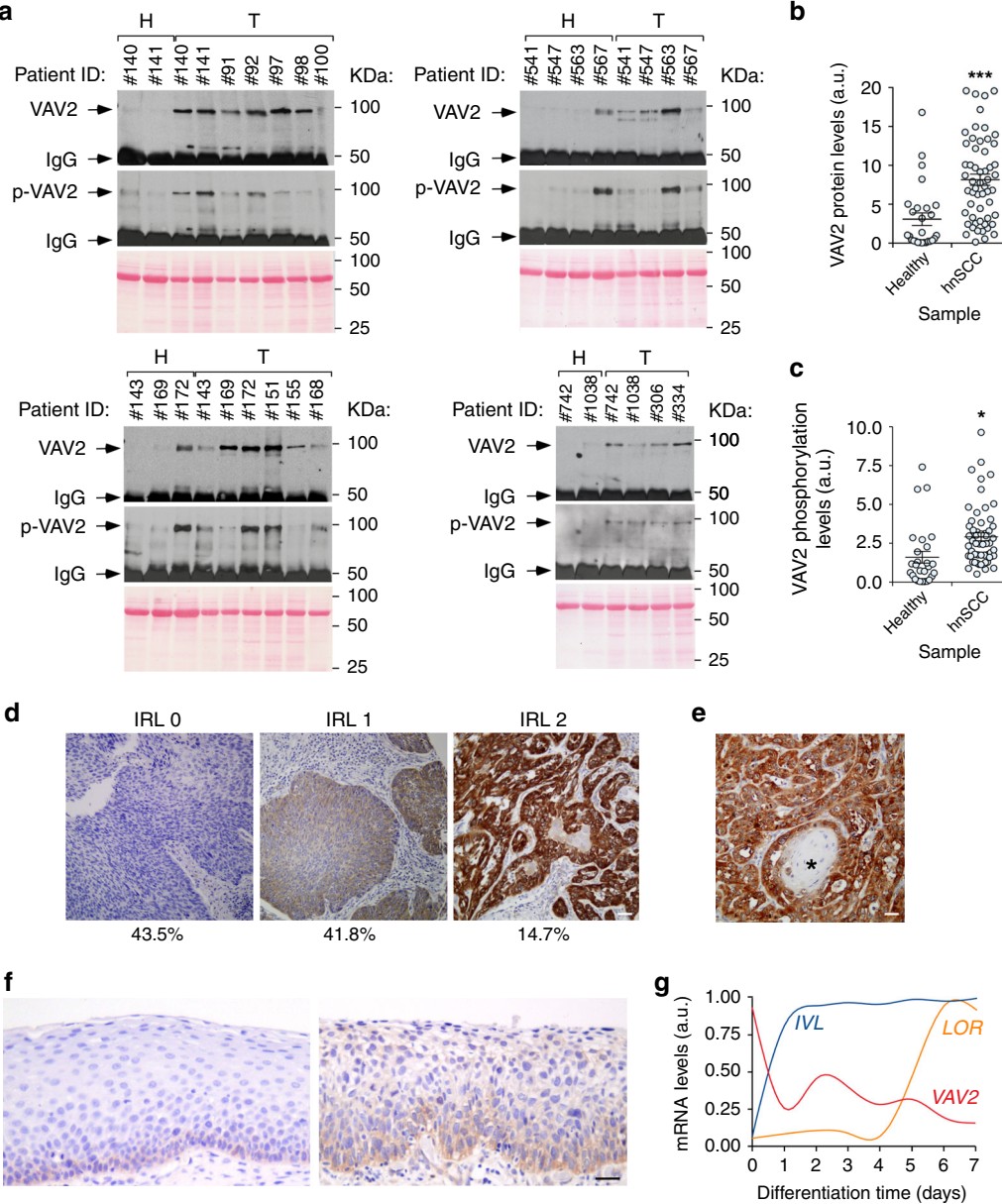

**Fig. 2 VAV2 protein is overexpressed in hnSCC. a** Levels of total (VAV2) and tyrosine-phosphorylated (p-VAV2) VAV2 in healthy (H) and tumor (T) tissues from hnSCC patients ($n = 83$). The band corresponding to the antibody used for the immunoprecipitation step is indicated (IgG). Amount of total protein content in the extracts used for the immunoprecipitation was tested in parallel filters using Ponceau staining (bottom panels). KDa, kilodalton. **b** VAV2 abundance according to data from **a**. ***$P < 0.0001$ (two-tailed Mann–Whitney test, $n = 83$ patient samples). a.u., arbitrary units. **c** VAV2 tyrosine phosphorylation levels according to data obtained in **a**. *$P = 0.011$ (two-tailed Mann–Whitney test, $n = 83$ patient samples). Please, note that the total levels of tyrosine phosphorylation of VAV2 in these analyses are probably underestimated given the long experimental procedure associated with tumor collection and subsequent lysis. **d** Example of tumor sections showing the indicated VAV2 immunoreactivity levels (IRL, top). The percentages of tumor samples (% out of a total of 232 tumors analyzed) showing the indicated VAV2 immunoreactivity levels are shown at the bottom. $n = 232$ tumor samples. Scale bar, 200 μm. **e** Representative image showing the lack of VAV2 immunoreactivity in terminally differentiated areas (asterisk) of VAV2-positive tumors. Scale bar, 100 μm. **f** Representative images of the immunohistochemical staining of VAV2 in healthy (left) and dysplastic (right) samples from human oral epithelium. Sections were counterstained with hematoxylin. $n = 232$ tumor samples. Scale bar, 100 μm. **g** Expression levels of the indicated transcripts according to the differentiation status of keratinocytes. Data were obtained from the GEO GSE52651 dataset (see "Methods"). IVL involucrin (an mRNA expressed from the early stages of keratinocyte terminal differentiation); LOR loricrin (a transcript encoding a major protein component of the cornified cell envelope that is found in terminally differentiated epidermal cells). In **b** and **c**, data represent the mean ± SEM. Source data for this figure are provided as a Source Data file.

layers of the normal oral epithelium obtained from the same patients (Fig. 2f, left panel). However, we found that the VAV2 immunoreactivity expands to suprabasal cell layers when analyzed in samples exhibiting histological signs of dysplasia (Fig. 2f, right panel). The levels of immunoreactivity found in the foregoing cases were lower than those detected in tumors (Fig. 2d, e). The preferential expression of the *VAV2* mRNA in keratinocyte precursors and its downmodulation in more mature derivatives is also observed in the skin when using gene expression data from organotypic cultures that recapitulate this

differentiation process (Fig. 2g). By contrast, differentiation-associated transcripts show the expected upregulation as these cell precursors progressively give rise to a more mature progeny (Fig. 2g). Taken together, these results further indicate that the levels of VAV2 become highly elevated in a significant number of hnSCC cases when compared to both normal and dysplastic tissue conditions.

**VAV2 promotes a protumorigenic niche in the mouse epidermis.** To investigate the effect of upregulated VAV2 signaling in primary epithelial cells, we first made use of a previously described $Vav2^{Onc/Onc}$ gain-of-function knock-in strain that expresses a mutant version of the protein (Δ1-186, referred to hereafter as Vav2$^{Onc}$). This protein exhibits constitutive GEF activity due to the removal of the N-terminal inhibitory regions that, together with the C-SH3, maintain the inactive state of Vav family proteins in the absence of tyrosine phosphorylation[28,29]. Due to this, the activity of this active version must reflect the output of Vav2$^{WT}$ under optimal tyrosine phosphorylation conditions according to the current regulatory model for Vav family proteins[19,20,30]. Importantly, Vav2$^{Onc}$ is expressed from the endogenous locus and, therefore, exhibits the same pattern and levels of expression that the endogenous wild-type (WT) counterpart[29]. The analysis of these mice indicates the presence of extensive hyperplasia in the epidermal layers of the skin, palate, and tongue (Fig. 3a, b). This phenotype is associated with an increase in cell layers composed of both immature and highly proliferative (keratin 5 and 6 positive) keratinocyte precursors (Fig. 3c). Corneum strata are observed in all cases (Fig. 3a, c), indicating that Vav2$^{Onc}$ delays but does not abate the terminal differentiation of keratinocytes. In the case of the skin, we also observed frequent cases of dermal dysplasia (Fig. 3a, top panels).

We did not find tumor development in the skin or head and neck areas of Vav2$^{Onc/Onc}$ mice even after long periods of time (12 months of age). However, we did observe that those mice develop skin tumors with faster kinetics (Fig. 3d), higher multiplicity (Fig. 3e), and of larger size (Fig. 3f) than controls upon being subjected to a single topic administration of DMBA at newborn stages. The malignant progression of these tumors is also accelerated, as inferred from the increased percentages of cSCCs found in $Vav2^{Onc/Onc}$ mice (Fig. 3g and Supplementary Fig. 2). These results indicate that upregulated Vav2 signaling creates a preneoplastic niche that favors full epidermal transformation when combined with additional genetic lesions.

**The effect of Vav2 in epidermis is keratinocyte autonomous.** Since the hyperplasia can develop as a consequence of both intrinsic and extrinsic factors to epithelial cells, we next investigated the direct effect of upregulated Vav2 signaling in primary mouse and human keratinocytes. To this end, we compared the ability of keratinocytes obtained from the skin of newborn WT and $Vav2^{Onc/Onc}$ mice to form epidermis-like structures in organotypic 3D cultures (Fig. 4a). Similar to the results obtained in mice, we found that the primary keratinocytes from newborn $Vav2^{Onc/Onc}$ mice develop a thicker and more proliferative epidermis than WT cells (Fig. 4a, b and Supplementary Fig. 3a). To assess the level of phylogenetic conservation of this phenotype, we next performed similar organotypic experiments using immortalized human keratinocytes ectopically expressing Vav2$^{Onc}$. These cells also form thicker and more proliferative layers than controls (Fig. 4c, d and Supplementary Fig. 3a, b). As in the case of the mouse primary keratinocytes, the epidermis formed by human keratinocytes ectopically expressing Vav2$^{Onc}$ is characterized by the expansion of layers of highly undifferentiated

keratinocytes that still keep the ability to terminally differentiate when reaching the outermost layers of the epithelium (Fig. 4c, bottom panels). This phenotype is lost when a catalytically-dead version of Vav2$^{Onc}$ (E200A mutant) is ectopically expressed in human keratinocytes (Supplementary Fig. 3b and Fig. 4c, d), indicating that Vav2 triggers the epithelial hyperplasia using a catalysis-dependent mechanism.

To identify the downstream Rho GTPases involved in this process, we performed G-ELISA assays to monitor the GTP loaded (active) status of the endogenous Rac1, RhoA, and Cdc42 in the mouse and human keratinocytes utilized above. As positive controls, we used human keratinocyte derivatives stably expressing mutant versions of Rac1 (F28L mutant), RhoA (F30L mutant), and Cdc42 (F28L mutant) (Supplementary Fig. 3c). These mutant proteins show constitutive activity due to high intrinsic rates of GDP/GTP exchange in the absence of upstream stimulation[31]. We observed that both the endogenously (Fig. 4e, left panel) and the ectopically expressed (Fig. 4e, right panel) Vav2$^{Onc}$ proteins trigger the stimulation of Rac1 and RhoA in mouse and human keratinocytes, respectively. However, we could not find any statistically significant elevation of the GTP levels of endogenous Ccd42 (Fig. 4e, right panel). As control, the cell lines expressing the GTPase mutant proteins show high GTP levels of activation of the expected Rho protein (Fig. 4e, right panel). In agreement with this, we observed that human keratinocytes expressing Rac1$^{F28L}$ and, to a lesser extent the other constitutively activated Rho GTPases elicit a hyperplasic response when tested in 3D cultures (Fig. 4f, g). This response is more clearly seen in keratinocytes coexpressing Rac1$^{F28L}$ and RhoA$^{F30L}$ (Fig. 4f, g and Supplementary Fig. 3d). As in the case of Vav2$^{Onc}$-expressing cells (Fig. 4a, c), this phenotype is associated with increased numbers of proliferative and undifferentiated strata of keratinocytes (Fig. 4h). In line with the implication of Rac1 and RhoA in this process, we found that the epithelial hyperplasia elicited by ectopically expressed Vav2$^{Onc}$ can be blocked by inhibitors of well-known proximal downstream elements of the Rac1 (Pak kinases) and RhoA (Rock kinases) pathways (Supplementary Fig. 3e, f)[10]. Collectively, these results indicate that the biological effects triggered by Vav2$^{Onc}$ in the epidermis are keratinocyte autonomous and catalysis-dependent.

**Vav2 controls a protumorigenic and stem cell-like program in keratinocytes.** We next performed microarray analyses using samples from the epidermis of WT and $Vav2^{Onc/Onc}$ mice to get information on the biological programs associated with upregulated Vav2 signaling in the epidermis. We found a total of 2,153 probe sets (1,881 genes) whose expression is differentially deregulated in $Vav2^{Onc/Onc}$ mice (Supplementary Data 1). Interestingly, ≈37.4% and 40.5% of the up- and downregulated subsets (referred to hereafter as "redundant transcriptome") display an opposite regulation to that found in the previously described response of the skin of $Vav2^{-/-};Vav3^{-/-}$ mice to the tumor promoter 12-O-tetradecanoylphorbol-13-acetate (TPA)[17] (Supplementary Fig. 4, Supplementary Table 2). This suggests that a significant fraction of the transcriptome that is elicited by TPA during the promotion phase of skin tumorigenesis can also be triggered by Vav2$^{Onc}$ in a signaling autonomous manner. Functional annotation analyses revealed that these gene expression changes are consistent with the phenotype observed in $Vav2^{Onc/Onc}$ mice, including the deregulation of genes associated with proliferative and epidermal differentiation programs (Fig. 5a). The use of Gene Set Enrichment analyses (GSEA) confirmed the foregoing programs and, in addition, further underscored the impact of increased Vav2 signaling on the overall differentiation program of keratinocytes. These

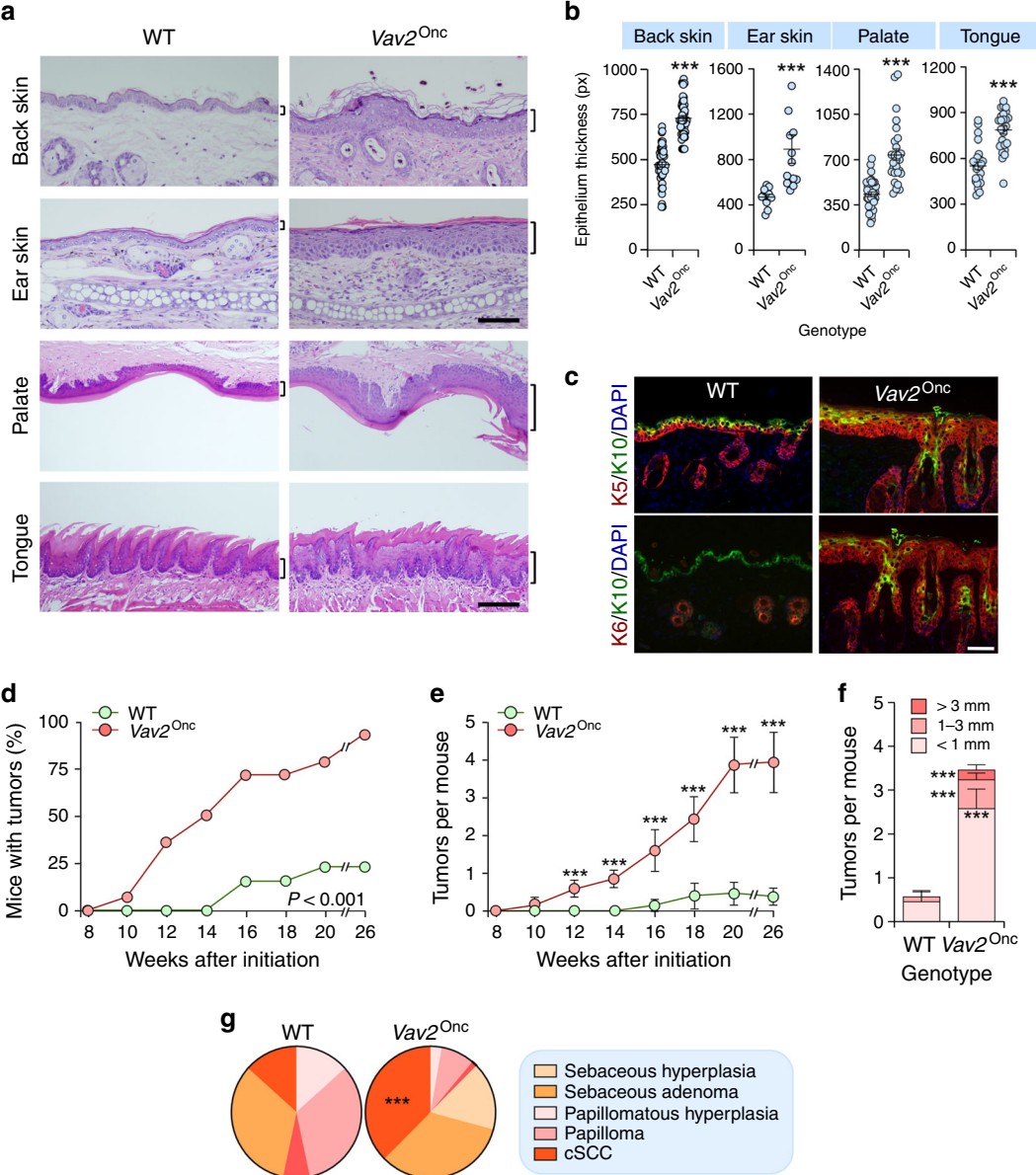

**Fig. 3 Upregulated VAV2 signaling creates a protumorigenic niche. a** Histological sections of the epithelia from mice of the indicated genotypes. The thickness of the epithelium is indicated with a bracket. Scale bar, 100 (back and ear skin) and 200 (palate and tongue) μm ($n = 6$ animals per genotype). **b** Thickness of indicated epithelial linings according to the data gathered in **a**. ***$P < 0.0001$ (two-tailed Student's $t$-test, $n = 6$ animals per genotype). px pixel. **c** Expression of keratin 5 (K5; top, red color), keratin 6 (K6; bottom, red color), and keratin 10 (K10; both panels, green color) in the skin from the back of animals of indicated genotypes. In all cases, the nuclei of cells were labeled with DAPI (blue color). Scale bar, 50 μm ($n = 3$). Penetrance (**d**) and multiplicity (**e**) of tumors generated in mice of the indicated genotypes upon a single topic DMBA application at postnatal day 3. The differences between genotypes were statistically significant from week 12 onwards ($P < 0.0001$, two-tailed Student's $t$-test). $n = 24$ and 31 WT and $Vav2^{Onc/Onc}$ mice, respectively. **f** Size distribution of the tumors at the endpoint of the experiment shown in **d**. ***$P < 0.0001$ (two-tailed Student's $t$-test, $n$ as in **d**). **g** Tumor type distribution at the endpoint of the experiment shown in **d**. $P < 0.0001$ (Chi-squared test, $n = 15$ and 72 WT and $Vav2^{Onc/Onc}$ tumors, respectively). In **b**, **e**, and **f**, data represent the mean ± SEM. Source data for this figure are provided as a Source Data file.

effects included the downregulation and upregulation of gene signatures characteristic of late epidermal differentiation- and epidermal cell progenitor-associated programs (Fig. 5b, upper panels), respectively. Perhaps more importantly, we found the enrichment of gene signatures related to cSCC tumor initiating cells and embryonic stem cells in the Vav2^Onc-induced transcriptome (Fig. 5b, bottom panels). No statistically significant overlaps are found with the transcriptomes of normal hair follicle stem cells, indicating that Vav2^Onc triggers an aberrant dedifferentiation process more similar to general stem cells than to those found in the most undifferentiated skin precursors. In

line with the induction of the combination of proliferative and stem cell like-programs, we found that the epidermis of Vav2^Onc/Onc mice reacts to the topic administration of a single dose of TPA with more rapid and wide proliferative responses than the skin from control animals (Fig. 5c). We also observed that the primary keratinocytes from newborn Vav2^Onc/Onc mice generate higher numbers of induced pluripotent stem (iPS) cell colonies than the WT counterparts upon transfection with the Yamanaka's cocktail (Fig. 5d). These observations indicate that the upregulation of Vav2 signaling leads to a stem cell-like state of regenerative proliferation and delays in keratinocyte

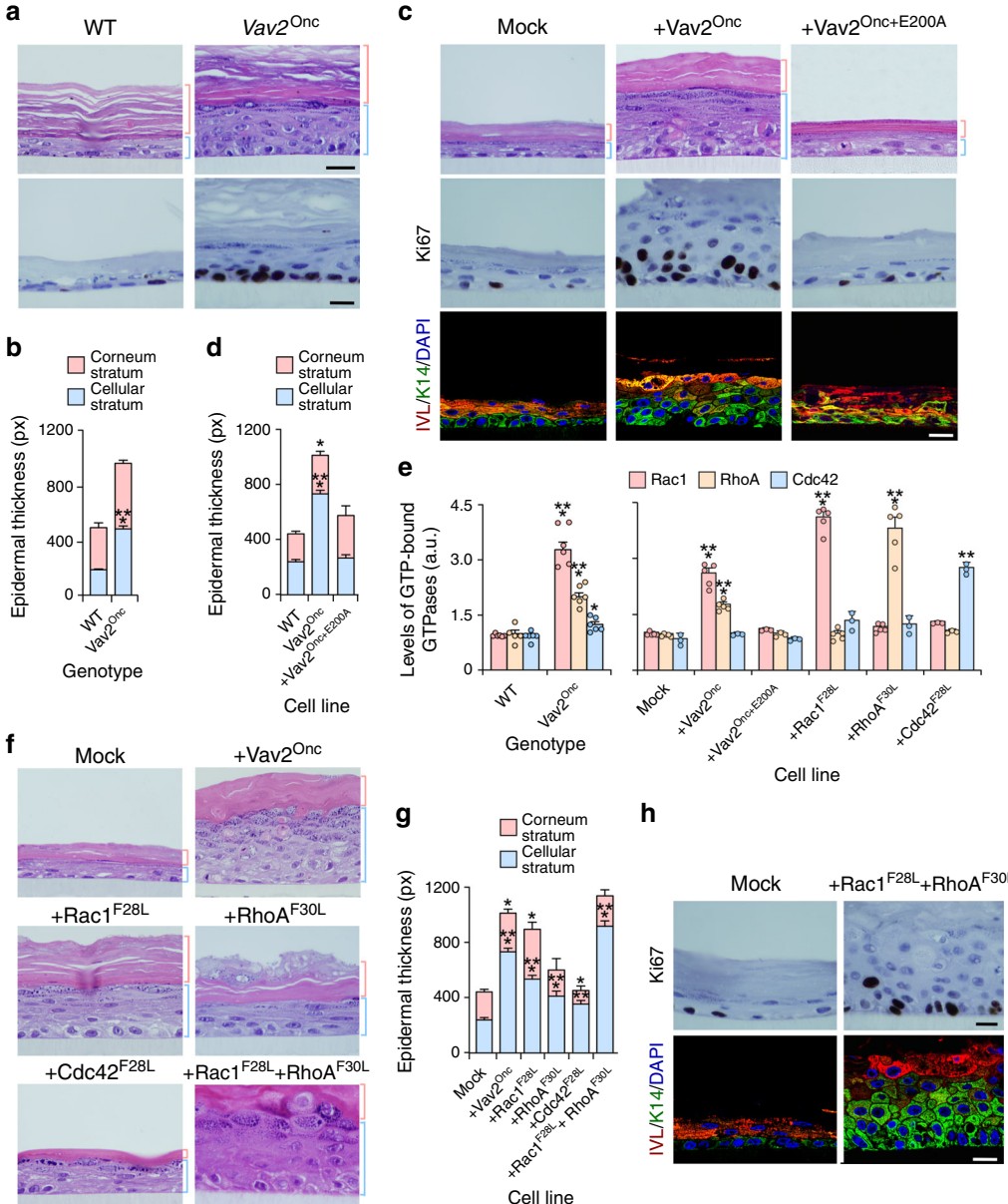

**Fig. 4 The effect of Vav2 in epidermis is keratinocyte autonomous. a** Hematoxylin-eosin- (top) and Ki67- plus hematoxylin-stained (bottom) tissue sections from organotypic cultures using neonatal keratinocytes of indicated genotypes. The thickness of the corneum and cellular epidermal strata is indicated with red and blue brackets, respectively. Scale bar, 10 μm ($n = 6$ independent cultures). **b** Thickness of the indicated skin strata according to data from **a**. ***$P < 0.0001$ (two-tailed Student's $t$-test, $n = 6$ independent cultures). **c** Representative images of indicated organotypic cultures after staining with hematoxylin-eosin (top panels) and antibodies to Ki67 (middle panels), involucrin (IVL, bottom, red color), and keratin 14 (K14, green color). Some of the sections were subsequently counterstained with either hematoxylin (middle row of panels) or DAPI (bottom panels, blue color). Scale bar, 10 μm. $n = 8$ (Mock and Vav2^Onc) and 5 (Vav2^Onc+E200A). **d** Thickness of the indicated skin strata according to data obtained in **c**. *$P = 0.029$; ***$P < 0.0001$ (two-tailed Student's $t$-test, $n$ as in panel c). **e** GTP-bound levels of Rac1, RhoA, and Cdc42 in indicated cells. *$P = 0.013$; **$P = 0.002$; ***$P < 0.0001$ (ANOVA and Tukey's HSD test, $n = 3$ independent experiments). **f** Histological sections of 3D cultures of human keratinocytes expressing indicated proteins. Scale bar, 10 μm ($n = 5$ independent cultures). **g** Thickness of the indicated epidermal strata according to data from **f**. *$P = 0.047$ (Vav2^Onc corneum stratum), 0.011 (Rac1^F28L corneum stratum), 0.033 (Cdc42^F28L corneum stratum); **$P = 0.003$ (Cdc42^F28L cellular stratum); ***$P < 0.0001$ (all other tests) (ANOVA and Dunnett's multiple comparison test, $n = 5$ independent cultures). **h** Immunohistochemical staining of mock- and Rac1^F28L + RhoA^F30L-expressing human keratinocytes with antibodies to Ki67 (top panels), IVL (bottom, red color) and K14 (bottom, green color). Sections were counterstained with either hematoxylin (top panels) or DAPI (bottom panels, blue color). Scale bar, 10 μm ($n = 3$). In **b**, **d**, **e**, and **g**, data represent the mean ± SEM. Source data for this figure are provided as a Source Data file.

differentiation that probably contribute to the epidermal hyperplasia seen in Vav2^Onc/Onc mice.

In line with the protumorigenic status of the epidermis of Vav2^Onc/Onc mice, bioinformatics analyses revealed that the Vav2^Onc-associated transcriptome shares a large degree of overlap with gene expression programs present in mouse papilloma and cSCC (Fig. 5e, left panels), human cSCC (Fig. 5e, middle panels), human hnSCCs (Fig. 5e, right panels), and human oral epidermal dysplasia (Fig. 5e, right panels). Conversely, GSEAs confirmed the enrichment of gene signatures

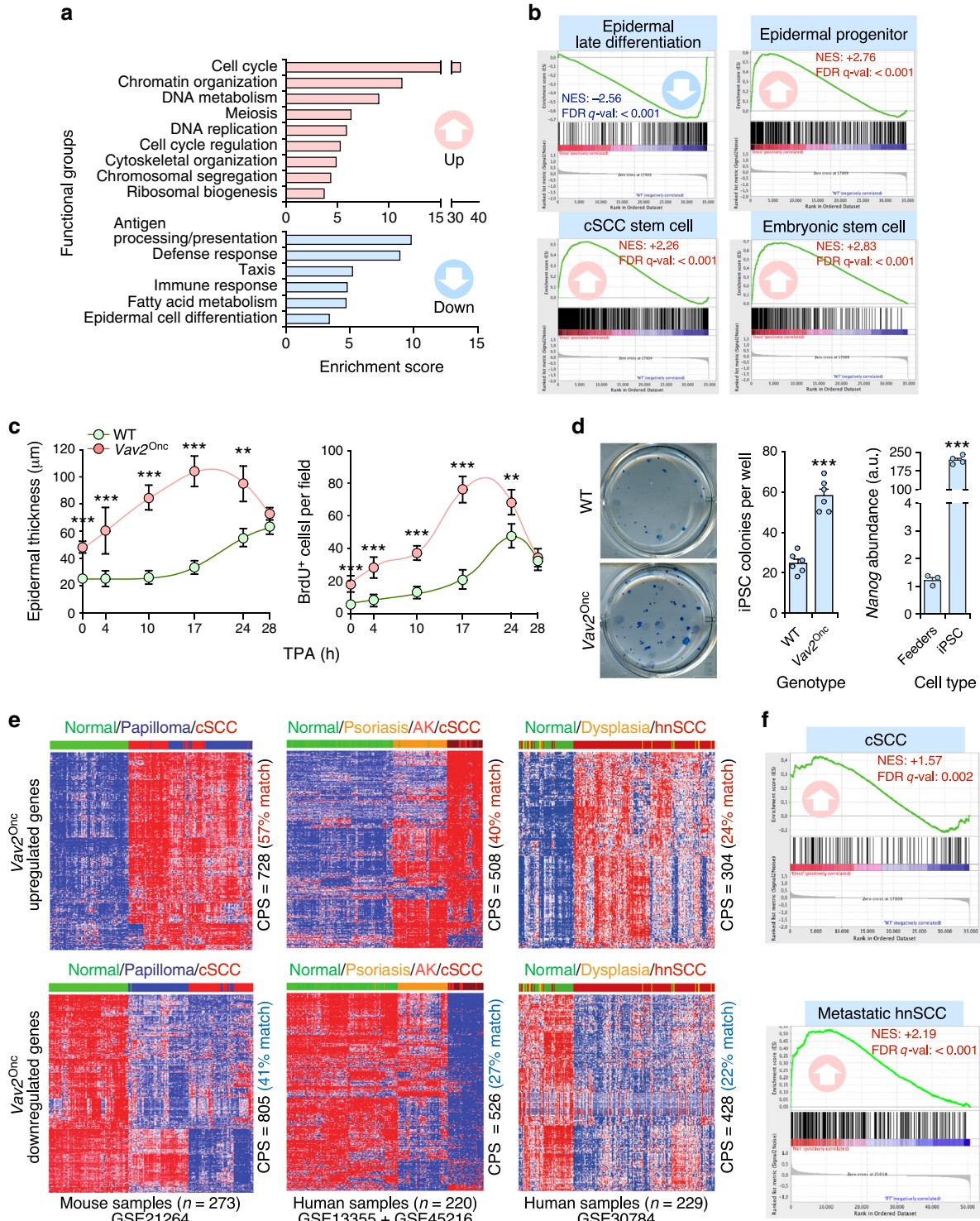

characteristic of metastatic hnSCC patient-derived cells and mouse cSCCs (Fig. 5f). Taken together, these results indicate that the upregulation of Vav2 signaling is associated with the generation of an undifferentiated, regenerative population of epidermal precursors that bear many biological and transcriptomal features with squamous epithelial tumors originated from both the skin and head and neck areas.

**Regulatory elements of the Vav2[Onc] transcriptome.** We next carried out further in silico analyses to obtain clues about the distal transcriptional factors involved in the regulation of the

Vav2[Onc]-dependent gene expression program. We found that the upregulated genes of this transcriptome are enriched in binding sites for the AP1 family, c-Myc, E2F, Nfy, and TEAD (Fig. 6a, b). By contrast, the downregulated genes seem to be under the regulation of a different subset of transcriptional factors such as Mitf, Mef2, FoxO, and RXRα (Fig. 6a, b). Consistent with these data, GSEAs demonstrated the presence of gene signatures previously linked to those factors in the Vav2[Onc]-dependent transcriptome (Supplementary Fig. 5a).

Given that upregulated genes are easier to tamper with than downregulated ones in the context of both biological and

**Fig. 5 Vav2 controls a protumorigenic stem cell-like program in keratinocytes. a** Main functional categories encoded by the $Vav2^{Onc}$-dependent transcriptome. For all of them, $P < 0.001$ (Fisher's exact test). **b** Gene signatures dysregulated in the $Vav2^{Onc}$-dependent transcriptome. FDR $q$ value < 0.0001 (GSEA statistical test). NES normalized enrichment score. Negative and positive enrichments are indicated with downward and upward arrows, respectively. **c** Time-course showing the changes in the thickness (left) and BrdU incorporation (right) of the epidermis of mice of the indicated genotypes in response to topic applications of TPA. $**P = 0.002$ (epidermal thickness, 24 h), 0.009 (BrdU incorporation, 24 h); $***P < 0.0001$ (all other tests) using two-tailed Student's $t$-test, $n = 3$ independent experiments. **d** Representative image (left panel) and quantification (middle panel) of the number of induced pluripotent stem cell (iPSC) colonies generated by keratinocytes of the indicated genotypes upon transduction with the Yamanaka's cocktail. As a control, the quantification of the pluripotency marker *Nanog* in the cultures is shown (right panel). $***P < 0.0001$ (two-tailed Student's $t$-test, $n = 3$ independent experiments). **e** Fraction of the upregulated (top panels) and downregulated (bottom panels) $Vav2^{Onc/Onc}$-driven transcriptome showing conservation with mouse cSCC (left panels), human cSCC (middle panels), and human hnSCC (right panels) transcriptomes. The number of tumor samples is indicated at the bottom. The level of conserved probe sets (CPS) is shown on the right of each panel. AK, actinic keratosis. **f** GSEA of the indicated gene sets (top) in the $Vav2^{Onc/Onc}$-regulated transcriptome. The NES and FDR values are indicated within each graph (GSEA statistical test). Positive enrichments are indicated with upward arrows. The conversion of mouse to human probe set has been made using the hgu133plus2 platform. In **c** and **d**, data are given as the mean ± SEM. Source data for this figure are provided as a Source Data file.

therapeutic systems, we decided to focus our validation studies on the transcriptional factors associated with the upregulated gene subset of the $Vav2^{Onc}$-dependent transcriptome. Using immunohistochemical analyses, we confirmed that c-Fos (an AP1 family member), c-Myc, Cyclin D1 (used as a surrogate for the activity of E2F family proteins), and YAP are both upregulated and nuclear-localized in the epidermis of $Vav2^{Onc/Onc}$ mice (Fig. 6c, two columns of panels on the left). Similar results were obtained using immunohistochemical experiments with epidermal sections from the organotypic cultures generated by both mouse $Vav2^{Onc/Onc}$ (Fig. 6c, third and fourth set of column panels from the left) and $Vav2^{Onc}$-expressing human (Fig. 6c, rest of panels) keratinocytes. We also found higher levels of nuclear-localized c-Fos, c-Myc, and YAP in the epidermis formed by keratinocytes ectopically coexpressing the active versions of both Rac1 and RhoA (Fig. 6d). The quantification of these immunohistochemical experiments is shown in Supplementary Fig. 5b.

The immunoreactive signals obtained with each of those antibodies always display much higher intensities in the $Vav2^{Onc}$ samples that in controls, indicating that they are not merely the consequence of the expansion of proliferative cell layers that already contain very high levels of activation of these transcriptional factors. Consistent with this, we also observed that the immunoreactivity of c-Fos, c-Myc, and YAP is detected in upper cell layers of the organotypic cultures that are negative for the proliferative markers Cyclin D1 (Fig. 6c) and Ki67 (Fig. 4c). To further confirm that the stimulation of these factors is the result of the direct downstream signaling of $Vav2^{Onc}$, we investigated whether the transient expression of the active version of this GEF could promote the stimulation of these transcriptional factors using luciferase reporter assays in 2D keratinocyte cultures. In agreement with the organotypic data, we found that the transient expression of $Vav2^{Onc}$ leads to the stimulation of the transcriptional activity of the endogenous AP1 factors (Fig. 6e), c-MYC (Fig. 6f), E2F proteins (Fig. 6g), and the YAP/TAZ/TEAD complex (Fig. 6h). These effects are severely reduced when using the catalytically dead $Vav2^{Onc+E200A}$ protein (Fig. 6e to h). Immunoblot analyses confirmed the proper expression of the ectopically expressed proteins used in these experiments (Fig. 6i). These data indicate that $Vav2^{Onc}$ stimulates all these transcriptional factors in a catalysis-dependent and signaling autonomous-manner.

Further linking Vav2 with the stimulation of these factors, we observed using in silico analyses that the expression of well-known c-MYC- (Fig. 6j) and YAP/TAZ/TEAD-regulated (Fig. 6k) gene signatures is directly proportional to the abundance of the *VAV2* transcripts in both human cSCC and HPV$^-$ hnSCC patient samples. The correlation between *VAV2* transcript levels and the

c-MYC- (Fig. 6j, bottom right panel) and YAP/TAZ-regulated (Fig. 6k, bottom right panel) gene signatures is lost in most cases when the hnSCC samples are not stratified according to HPV status, further linking this $Vav2^{Onc}$-driven pathobiological program to HPV$^-$ tumor cases.

**The hyperplasia induced by Vav2 is c-MYC- and YAP-dependent.** Given the above results, we next investigated the potential implication of c-MYC and YAP/TAZ in the hyperplasia triggered by $Vav2^{Onc}$ and $Rac1^{F28L} + RhoA^{F30L}$ using a pharmacological approach. To this end, we chose concentrations of inhibitors of c-MYC (10058-F4) and YAP (Verteporfin) that did not cause significant interference with the growth and differentiation of control cells in the 3D cultures (Fig. 7a, top panels; Fig. 7b). With this strategy, we expected to unveil vulnerabilities specifically associated with the gain-of-function events elicited by the VAV2–GTPase signaling axis in these cells. We found that the inhibition of c-MYC blocks the epidermal hyperplasia induced by $Vav2^{Onc}$- and $Rac1^{F28L} + RhoA^{F30L}$-expressing keratinocytes (Fig. 7a, compare left and middle columns of panels; Fig. 7b). This effect is similar to that elicited by the incubation of these organotypic cultures with Pak and Rock inhibitors (see above, Supplementary Fig. 3e, f). We did not find any histological signs of cell differentiation (Fig. 7a), suggesting that the inhibition of c-MYC preferentially impacts on the proliferation of those cells. As a result, the epidermis formed by $Vav2^{Onc}$- and $Rac1^{F28L} + RhoA^{F30L}$-expressing keratinocytes under those conditions closely resemble those generated by the control, mock-transduced cells (Fig. 7a, b). The addition of the YAP inhibitor does not block the hyperplasia exhibited by $Vav2^{Onc}$-expressing cells. However, it does prompt the differentiation of all the immature keratinocyte cell layers but the most basal one (Fig. 7a, compare first and third columns of panels from left; Fig. 7b). This indicates that the YAP/TAZ/TEAD complex is mainly involved in sustaining the undifferentiated phenotype of suprabasal cell layers. To demonstrate the inhibition of the expected targets by the foregoing inhibitors, we evaluated c-MYC and YAP/TAZ/TEAD transcriptional activity in human keratinocyte 2D cultures transiently transfected with the indicated $Vav2^{Onc}$ versions plus luciferase reporter plasmids containing either c-MYC- or TEAD-responsive promoters, respectively. We observed that the addition of 10058-F4 blocks both the basal and the $Vav2^{Onc}$-driven MYC activity in those cells (Fig. 7c, left panel). On the other hand, Verterporfin reduces the basal and the $Vav2^{Onc}$-triggered transcriptional activity of TEAD in these culture conditions (Fig. 7c, right panel). It has been previously shown that the activation and inactivation of the YAP/TAZ/TEAD promotes and blocks epithelial hyperplasia, respectively[3–5]. However, dependent on the experimental

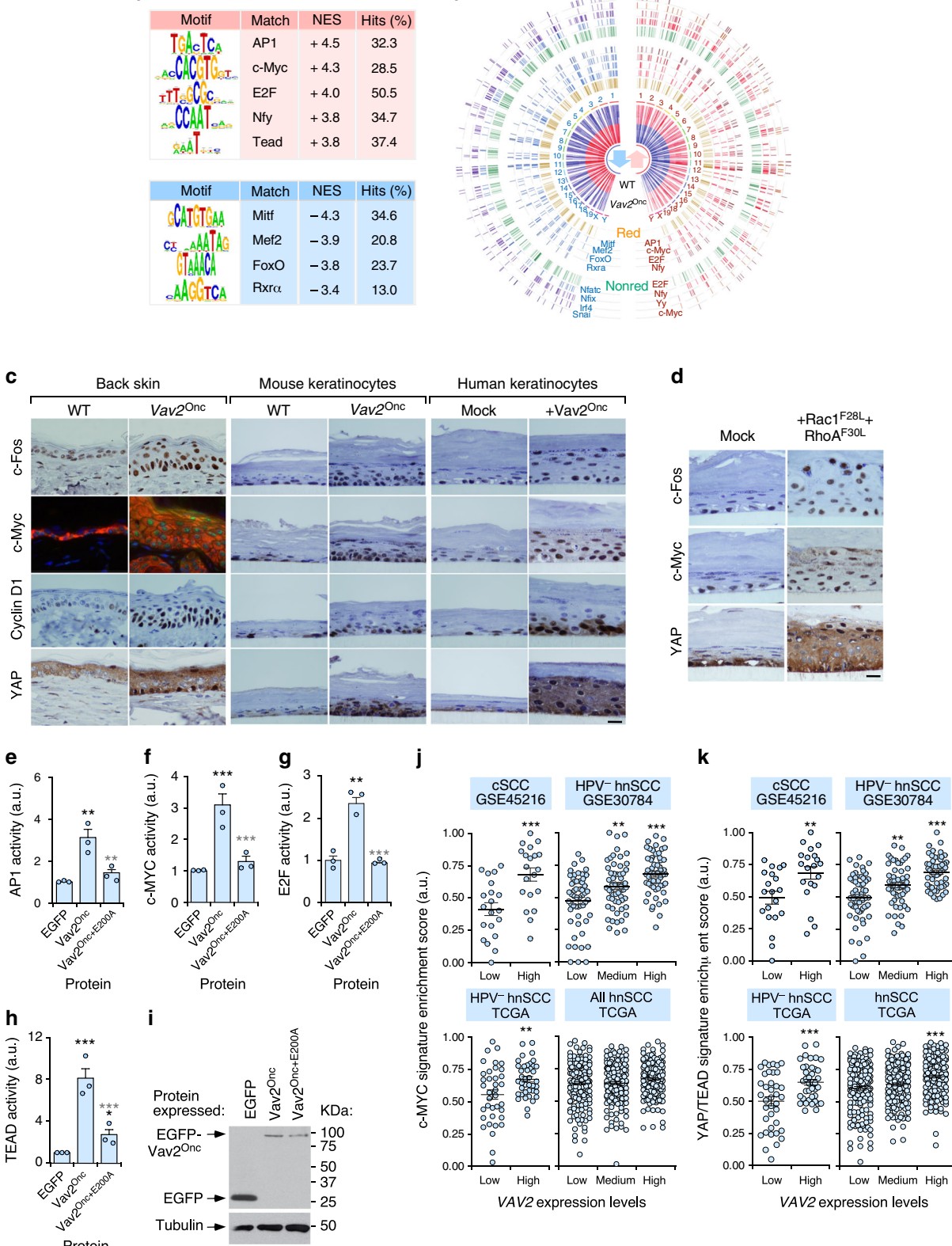

setting used, c-MYC has been associated with either regenerative proliferation or terminal differentiation in keratinocytes[7]. Consistent with this latter mechanism of action and the data obtained in our current work, we found that the stable ectopic expression of c-MYC in human keratinocytes (Fig. 7d) elicits a Vav2$^{Onc}$-like phenotype in the 3D organotypic cultures (Fig. 7e). These results indicate that c-MYC and the YAP/TAZ/TEAD complex are key

distal elements of the VAV2 pathway that promotes the acquisition of SCC-like molecular features by epithelial cells.

**Wild-type VAV2 triggers keratinocyte hyperplasia.** Given that the overexpression of VAV2$^{WT}$ is the most common event in hnSCC, we next investigated whether the overexpression of the

**Fig. 6 Transcriptional factors involved in the Vav2^Onc-dependent transcriptome. a** Transcription factor binding sites found enriched in the promoter regions of up- (red) and downregulated (blue) genes present in the "redundant" fraction of the *Vav2^Onc*-driven transcriptome. The NES and percentage of hits for each transcription factor binding site are also indicated. **b** Circos plot of the differentially expressed genes in the *Vav2^Onc* mouse skin. For each gene, the expression heatmap, the chromosomal location (1–19, X, Y), the binding sites for the indicated transcription factors, and the mirror-image behavior in the *Vav2^Onc*-regulated transcriptome and the Vav2;Vav3-dependent fraction of the TPA-stimulated gene expression program in the skin of mice are indicated. Red, redundant; Nonred, nonredundant. **c** Expression of indicated proteins (left) in the epidermis of the back skin of WT and *Vav2^Onc/Onc* mice (left), 3D organotypic cultures from WT and *Vav2^Onc/Onc* primary keratinocytes (middle), and 3D cultures of human keratinocytes (right). In the case of the immunofluorescence experiments using c-Myc antibodies (back skin, green color), sections were decorated with K14 (red color) and DAPI (blue color). Scale bar, 10 μm (*n* = 3 independent analyses for each experimental group). **d** Expression of indicated proteins (left) in 3D cultures of control and Rac1^F28L + RhoA^F30L-expressing human keratinocytes. Scale bar, 10 μm (*n* = 3 independent cultures). **e–h** Transiently transfected Vav2^Onc triggers the rapid activation of endogenous AP1 (**e**), c-MYC (**f**), E2F- (**g**), and TEAD (**h**) proteins in human keratinocytes. Activity was measured using luciferase-encoding vectors containing promoter regions for each of the interrogated transcriptional factors, as described in Methods. Data represent the mean ± SEM. Black and gray stars indicate the *P* value of the indicated experimental values when compared to EGFP- and EGFP-Vav2^Onc-transfected cells, respectively. *P = 0.011 (Vav2^Onc+E200A versus Vav2^Onc, TEAD); **P = 0.008 (Vav2^Onc versus EGFP, AP1), 0.010 (Vav2^Onc+E200A versus Vav2^Onc, AP1), 0.002 (Vav2^Onc versus EGFP, E2F); ***P < 0.0001 (all other tests) (ANOVA and Tukey's HSD test, *n* = 3 independent experiments). **i** Expression of indicated proteins in one of the experiments performed in **e** to **h**. Tubulin was used as loading control (bottom panel) (*n* = 3 independent experiments). **j, k** Correlation between the levels of the *VAV2* mRNA and the expression of c-MYC- (**j**) and YAP/TEAD-regulated- (**k**) gene signatures in the indicated (top) cSCC (*n* = 40) and hnSCC (*n* = 685) gene expression datasets. **P = 0.007 (*VAV2* medium, MYC signature, GSE30784), 0.005 (*VAV2* high, MYC signature, HPV⁻ TCGA), 0.008 (*VAV2* high, YAP signature, GSE45216), 0.002 (*VAV2* medium, YAP signature, GSE30784); ***, P < 0.0001 (all other experiments) using the ANOVA and Dunnett's multiple comparison test). Data represent the mean ± SEM. Source data for this figure are provided as a Source Data file.

full-length version of this protein could induce a Vav2^Onc-like phenotype in nontransformed keratinocytes. To this end, we generated two independent pools of human keratinocytes expressing high levels of ectopically expressed Vav2^WT (Fig. 8a). Immunoprecipitation experiments indicated that this protein is tyrosine phosphorylated (Fig. 8b) and that it coimmunoprecipitates with a large molecular weight tyrosine-phosphorylated protein in these cells (Fig. 8b, asterisk). Based on previous results, it is likely that this protein corresponds to the autophosphorylated EGFR[32,33]. As control, we included in these experiments the previously described pools of human keratinocytes expressing either Vav2^Onc or the catalytically dead Vav2^Onc+E200A mutant protein (Fig. 8a). We observed that the keratinocytes ectopically expressing Vav2^WT generate thicker epithelial layers than control cells when tested in organotypic cultures (Fig. 8c, d). This phenotype, however, is milder than the exhibited by Vav2^Onc-expressing cells (Fig. 8c, d). Consistent with previous data (Fig. 4b, d), the keratinocytes expressing Vav2^Onc+E200A generate epithelial structures similar to controls (Fig. 8c, d). These results indicate that the overexpression of the full-length Vav2, the version of the protein commonly found deregulated in cSCC and hnSCCs (see above, Figs. 1 and 2, Supplementary Fig. 1), also promotes epithelial hyperplasia when overexpressed in human keratinocytes.

**Endogenous VAV2^WT is required for hnSCC maintenance.** Having proved that Vav2 activity is associated with the engagement of a regenerative proliferation and a cell undifferentiation state in normal keratinocytes, we next investigated whether VAV2 was also important to maintain those traits in already transformed hnSCC cells. To this end, we knocked down the endogenous *VAV2* transcripts using several independent shRNAs in a widely-used hnSCC cell line (SSC-25)[34] and in two HPV⁻ patient-derived cells obtained from two independent oral SCC cases (VdH01, VdH15)[35]. Unlike the case of SCC-25 cells that have been previously characterized (Supplementary Fig. 6a–c)[36], the genomic alterations present in the VdH01 and VdH15 cells remain unknown. Due to this, we characterized the whole genome of both patient-derived cells through next generation sequencing (Supplementary Note 1, Supplementary Fig. 6 and Supplementary References). When tested in organotypic cultures, we observed that all the control cells bearing an empty lentiviral

vector (referred to as "pLKO" cells) form in all cases unstructured tumor masses composed of highly proliferative (Ki67⁺) and undifferentiated (keratin 14⁺, involucrin⁻) cell layers according to immunohistochemical staining (Fig. 9a). By contrast, the *VAV2* knockdown cells (Supplementary Fig. 7) generate under the same conditions thinner and more epithelioid structures (Fig. 9a and Supplementary Fig. 8a) that, in some cases (e.g., VHd15 cells) can even form a corneum stratum (Fig. 9a). However, unlike the case of normal keratinocytes, the knockdown cells cannot enucleate properly during the formation of this latter cell layer (Fig. 9a). The organotypic cultures from *VAV2* knockdown cells also display a reduction in Ki67⁺ levels and the presence of upper cell layers composed of differentiated, involucrin⁺ cells (Fig. 9a and Supplementary Fig. 8b). To analyze the behavior of the *VAV2* knockdown patient-derived cells in vivo, we transplanted them orthotopically in the tongue of *Vav1^−/−;Vav2^−/−;Vav3^−/−* mice. As these mice are lymphopenic[37,38] and generated in our own animal facility, they represent a cheap alternative to the use of the commercial immunodeficient mouse strains typically used for this type of experiments. Using this model, we found that the hnSCC patient-derived cells bearing the *VAV2* shRNA show reductions in both the tumorigenic (Fig. 9b, c) and metastatic (Fig. 9d) potential when compared to controls. Interestingly, we also observed that the small tumors generated by the *VAV2* knockdown cells undergo a process of keratinization and differentiation without proper enucleation similar to that found in the organotypic cultures (Fig. 9e). These results indicate that the elimination of endogenous *VAV2* in hnSCC cells induces an effect opposite to that elicited by upregulated VAV2 signaling in primary, nontransformed keratinocytes. Consistent with this, we observed that the transcriptional activity of the endogenous c-MYC, YAP/TAZ/TEAD, AP1, and E2F transcriptional factors is severely reduced in the *VAV2* knockdown patient-derived cells and SSC-25 cells when compared to the respective controls (Fig. 9f). Likewise, the 3D structures generated by those cells show lower c-MYC (Supplementary Fig. 8c, upper panels), YAP (Supplementary Fig. 8c, middle panels), and c-FOS (Supplementary Fig. 8c, bottom panels) immunoreactivity than controls. Similar results are obtained when the expression of those three transcriptional factors is measured in tumors from the xenograft experiments (Fig. 9g, h and Supplementary Fig. 8d). Further demonstrating the importance of this VAV2 catalysis-dependent pathway for

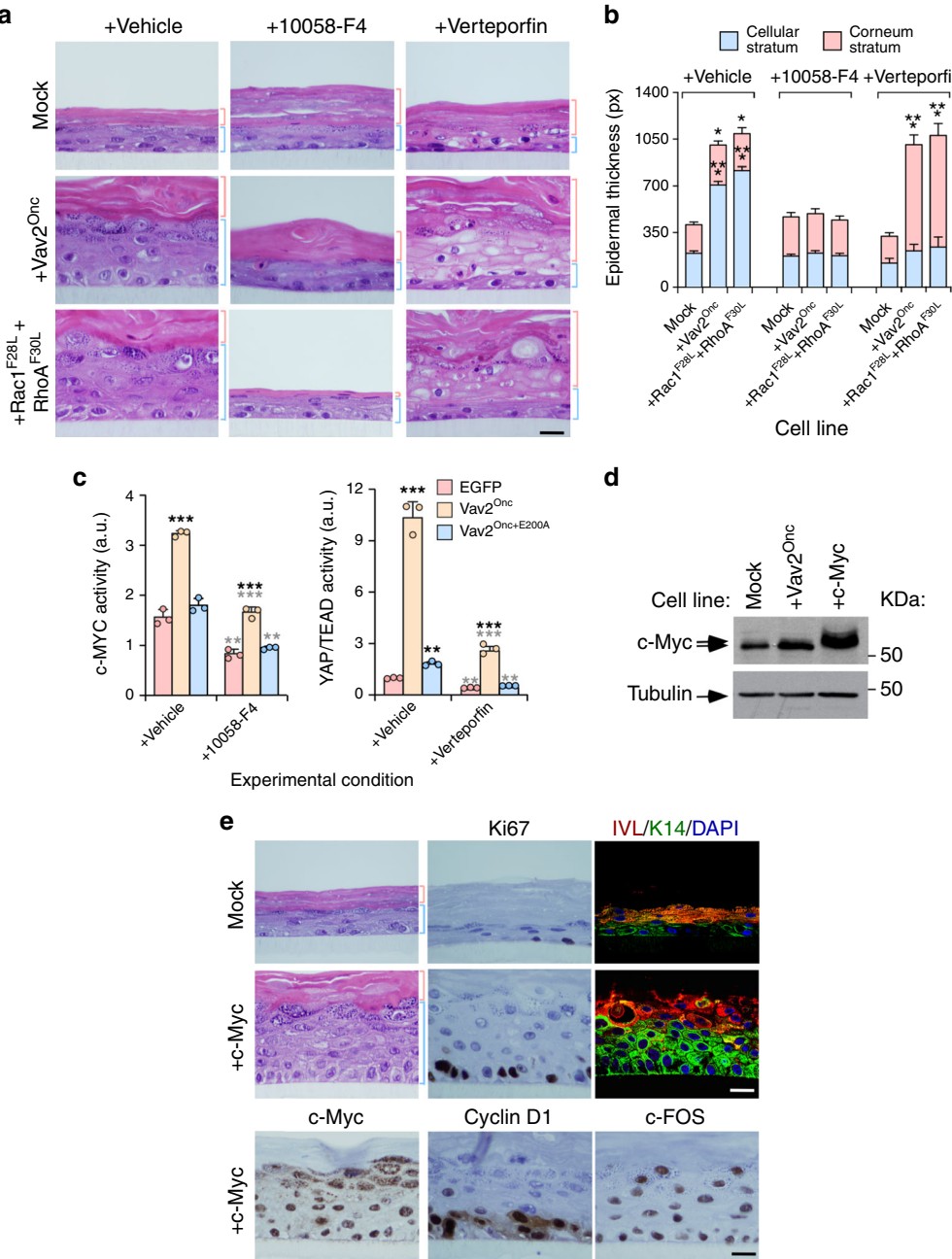

**Fig. 7 Vav2^Onc-driven hyperplasia is c-MYC- and YAP-dependent. a** Histological sections of 3D organotypic cultures of human keratinocytes expressing the indicated proteins (left) that have been treated with the inhibitors shown on top. Scale bar, 10 μm ($n = 3$ independent cultures). **b** Thickness of the cellular and corneum strata using data from **a**. *$P = 0.012$ (Vav2^Onc vehicle, stratum corneum), 0.037 (Rac1^F28L + RhoA^F30L vehicle, stratum corneum); ***$P < 0.0001$ (all other tests) (ANOVA and Dunnett's multiple comparison test, $n = 3$ independent cultures). **c** Demonstration that the transcriptional activity of c-MYC and TEAD is inhibited by MYC (10058-F4) and YAP/TEAD interaction (Verteporfin) inhibitors, respectively. Black and gray stars indicate the $P$ value of the indicated experimental values when compared to EGFP-transfected and vehicle-treated cells, respectively. **$P = 0.003$ (EGFP, 100058-F4, MYC activity), 0.002 (Vav2^Onc+E200A, 100058-F4, MYC activity), 0.001 (Vav2^Onc+E200A, vehicle, YAP/TEAD activity), 0.001 (EGFP, Verteporfin, YAP/TEAD activity), 0.002 (Vav2^Onc+E200A, Verteporfin, YAP/TEAD activity); ***, $P < 0.0001$ (all other tests) (ANOVA and Tukey's HSD test, $n = 3$ independent experiments). **d** Expression of endogenous and ectopic c-Myc in indicated cell lines. Tubulin was used as loading control (bottom). $n = 3$ independent experiments. **e** Staining of indicated human keratinocytes (left) with hematoxylin-eosin (left and middle two rows of panels) and antibodies to Ki67 (two top middle panels), IVL (two right panels on top, red color), K14 (two right panels on top, green color), ectopically expressed c-Myc (left bottom panel), endogenous Cyclin D1 (middle bottom panel) and endogenous c-FOS (right bottom panel). When indicated, some sections were counterstained with DAPI (two right panels on top. blue color). Scale bar, 10 μm ($n = 3$ independent cultures). In **b** and **c**, data represent the mean ± SEM. Source data for this figure are provided as a Source Data file.

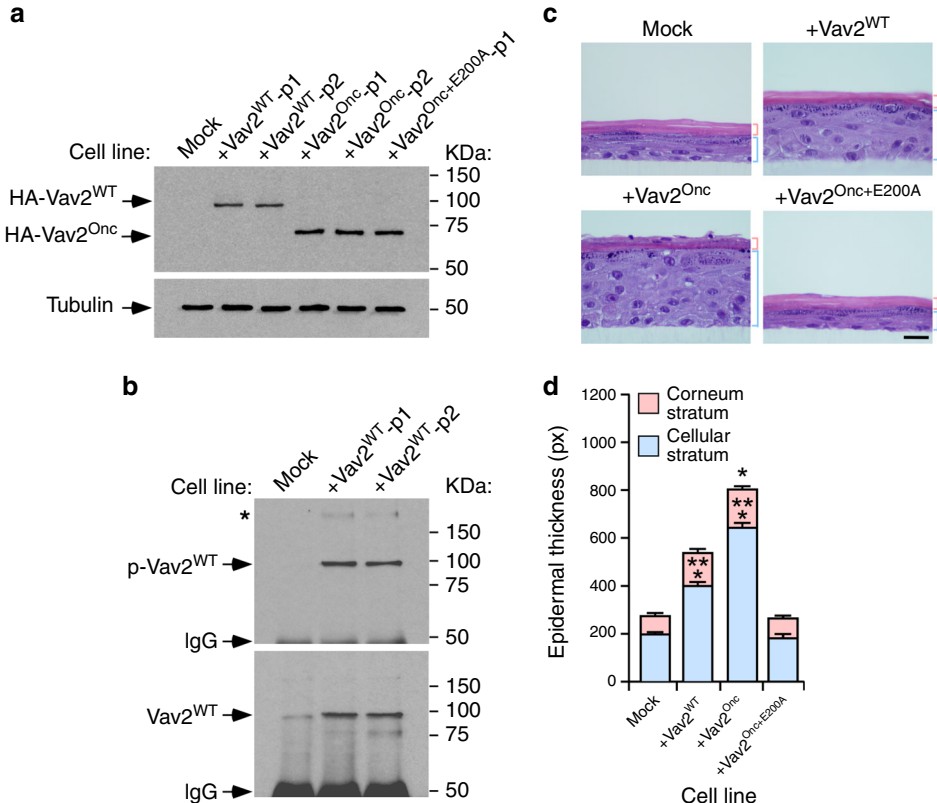

**Fig. 8 Wild-type VAV2 also triggers keratinocyte hyperplasia. a** Immunoblots showing the expression of the indicated Vav2 proteins in human keratinocytes (top panel). Tubulin was used as loading control in all cases (bottom panel) ($n = 3$ independent experiments). p1 and p2 refer to two independent pools of cells. **b** Tyrosine-phosphorylation levels (top panel) of immunoprecipitated Vav2$^{WT}$ (bottom panel) in the keratinocyte cell lines. Similar data were obtained in two additional independent experiments. p, tyrosine-phosphorylated. IgG, immunoglobulin band derived from the antibody used in the immunoprecipitation step. The asterisk marks a high molecular weight tyrosine-phosphorylated protein that coimmunoprecipitates with Vav2. **c** Representative images of hematoxylin/eosin-stained organotypic cultures generated by the indicated cells (top). The thickness of the corneum and cellular epidermal strata is indicated with red and blue brackets, respectively. Scale bar, 10 µm ($n = 3$ independent cultures). **d** Thickness of the cellular and corneum strata using data from **c**. *$P = 0.013$; ***$P < 0.0001$ (ANOVA and Dunnett's multiple comparison test, $n = 3$ independent cultures). Data represent the mean ± SEM. Source data for this figure are provided as a Source Data file.

hnSCC tumorigenicity, we observed that the addition of chemical inhibitors for RAC1 (1A116)[39], c-MYC, and YAP promotes a *VAV2* knockdown-like phenotype when added to the organotypic cultures of VdH15 cells (Supplementary Fig. 9). Together with the conservation found for the Vav2$^{Onc}$-regulated gene signatures in hnSCC patients (Fig. 5e, f), these data indicate that endogenous VAV2 is important to maintain the proliferation and undifferentiation state of hnSCC cells.

**Prognostic value of Vav2$^{Onc}$-regulated gene signatures.** The foregoing data, coupled with the prognostic power exhibited by the levels of *VAV2* mRNA transcripts in hnSCC patients (Fig. 1b), suggested that the Vav2$^{Onc}$-regulated gene expression landscape could be used to develop better prognostic tools for this tumor type. Concurring with this hypothesis, we found using bioinformatics techniques that the levels of expression of the whole Vav2$^{Onc}$-dependent transcriptome (Supplementary Data 1) correlate with poor HPV$^-$ hnSCC patient prognosis (Fig. 10a). This was observed in the three independent datasets that were chosen for the in silico studies (GSE41613 [$P = 0.0092$, Fig. 10a], TCGA [$P = 0.0068$], and GSE42743 [$P = 0.0361$]). Based on these observations, we aimed at identifying a minimal gene signature that could predict long-term survival in those patients. As a first step towards this direction, we developed a

first-generation Vav2$^{Onc}$ gene signature (Vav2$^{Onc}$-Sig) composed of a reduced number (187) of humanized probe sets (Fig. 10b, dark red area; Supplementary Table 3) that fulfilled the following a priori criteria: (i) Statistically significant deregulation in the Vav2$^{Onc}$-expressing mouse epidermis when compared to the epidermis of WT mice. (ii) Opposite pattern of regulation in the *Vav2$^{Onc}$* epidermis and the TPA-stimulated skin of *Vav2$^{-/-}$*; *Vav3$^{-/-}$* animals (Supplementary Fig. 4, Fig. 10b, and Supplementary Table 2; to make sure that they were regulated by the endogenous Vav2 proteins in a fully signaling autonomous manner). (iii) Similar patterns of deregulation when using human tumors versus healthy tissue samples (Figs. 5e, f and 10b; to make sure that such a deregulation was conserved across species). In good agreement with its association with the VAV2 pathway, the levels of expression of the Vav2$^{Onc}$-Sig correlated well with the abundance of *VAV2* transcripts in HPV$^-$ hnSCC samples (Fig. 10c). We also demonstrated using principal component analyses that the expression of the Vav2$^{Onc}$-Sig in those tumors is not indirectly caused by programs commonly found in most tumorigenic processes[9,40]. Hence, this signature shows a specific enrichment in human cSCC, hnSCC, actinic keratosis cases (Fig. 10d, area shaded in light red) and a very limited number of non-small cell lung tumors (Fig. 10d, blue dots). By contrast, it is not found enriched in psoriasis samples (Fig. 10d, area shaded in

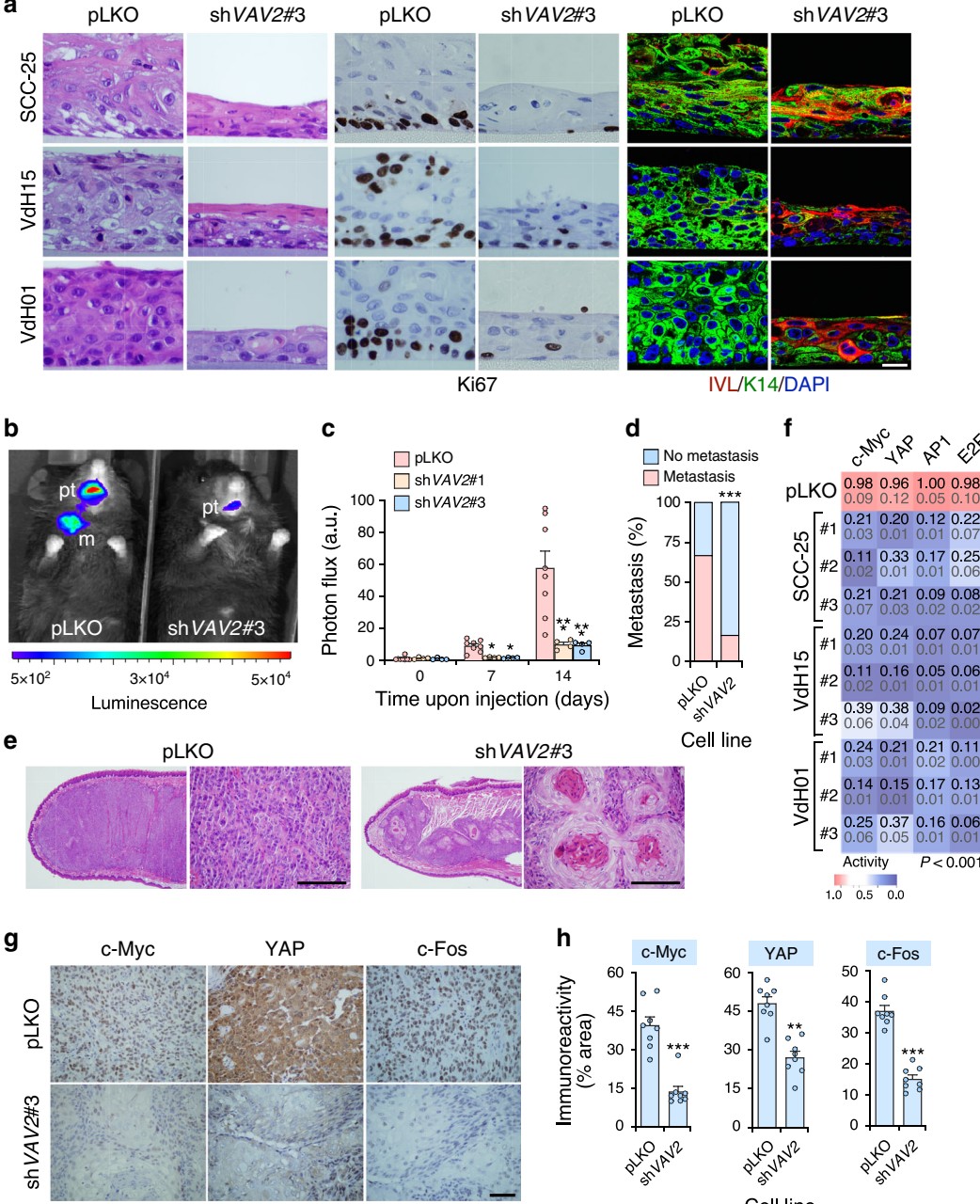

**Fig. 9 Endogenous VAV2 is required for the fitness of hnSCC cells. a** Images of sections from organotypic cultures of the indicated control (pLKO) and *VAV2* knockdown (sh*VAV2*) cell lines upon staining with hematoxylin-eosin (two left columns of panels) and antibodies to Ki67 (two middle columns of panels), IVL and K14 (two right columns of panels). Some of the sections were counterstained with either hematoxylin (two middle columns of panels) or DAPI (two right columns of panels, blue color). The specific cell line and experimental modification used is shown on the left and top, respectively. pLKO, cells transduced with an empty lentivirus. sh*VAV2*#3, cells transduced with a lentivirus encoding the *VAV2* shRNA #3. The same notation will be used in the rest of panels of this figure. Scale bar, 10 μm (*n* = 5 independent cultures). **b** Luminescence readings at the endpoint of the tongue orthotopic xenograft experiment using the indicated VdH15 cells (*n* = 8 per cell type derivative). pt, primary tumor; m, metastasis. **c** Photon flux of the indicated xenografted animals according to data from panel **b**. Data represent the mean ± SEM. *P = 0.010 and 0.012, respectively (day 7); ***P < 0.0001 (day 14) (Kruskal–Wallis test, *n* = 8). **d** Lymph node metastatic events detected in animals used in panel **b**. Data represent percentage. ***P < 0.0001 (Chi-squared test, *n* = 8). **e** Histological sections of the SCC tumors formed by the indicated VdH15 cells in the experiments shown in **b**. Scale bar, 75 μm (*n* = 8 per cell type derivative). **f** Luciferase activity recorded in the indicated cells (left) upon transient transfection with plasmids encoding the luciferase gene under the regulation of promoters regulated by the indicated transcriptional factors (top). For each condition, the mean and the SEM are indicated in black and gray, respectively. The color gradient is proportional to the level of inhibition achieved when compared to control cells (whose values are shown in red). #1, #2 and #3 (left) refer to the independent shRNAs used to knockdown endogenous *VAV2*. For all of them, *P* < 0.001 (ANOVA and Dunnett's multiple comparison test, *n* = 3 independent experiments). **g** Expression of indicated proteins (top) in tumors formed in the experiment shown in **b**. Scale bar, 50 μm (*n* = 8). **h** Quantification of immunohistochemical data from **g**. **P = 0.001; ***P < 0.0001 (two-tailed Student's *t*-test, *n* = 8). Data represent the mean ± SEM. Source data for this figure are provided as a Source Data file.

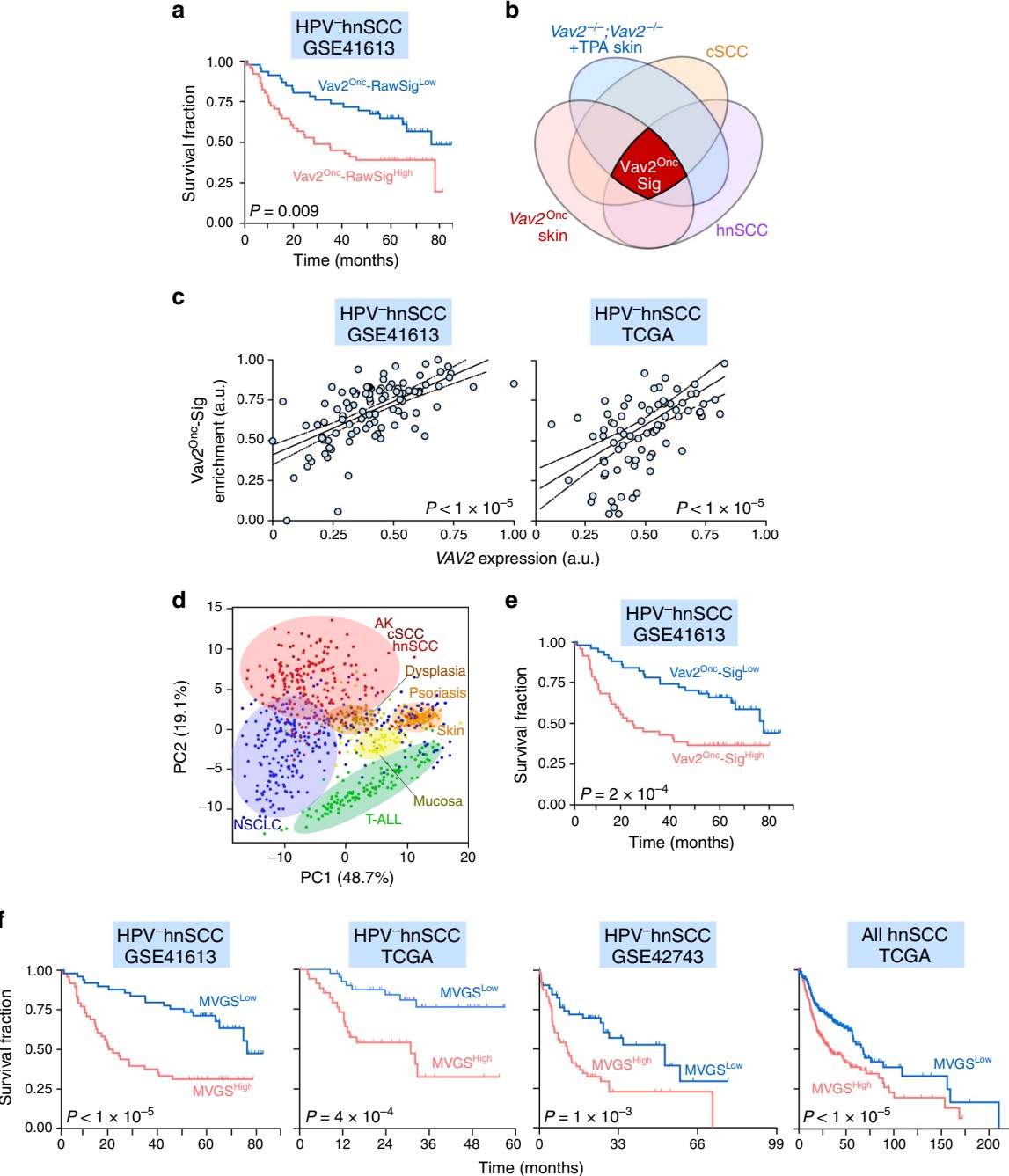

**Fig. 10 A prognostic Vav2^Onc-regulated gene signature for HPV⁻ hnSCC. a** Survival of HPV⁻ hnSCC patients ($n = 97$) according to the expression of the raw Vav2^Onc-driven transcriptome. The Mantel–Cox test $P$ value is indicated. **b** Venn diagram representing the criteria used to select the Vav2^Onc-Sig (dark red color). The datasets used include: the Vav2^Onc-driven transcriptome (Vav2^Onc skin), the Vav2^{−/−};Vav3^{−/−}-dependent fraction of the transcriptome induced by TPA (12-O-tetradecanoyl-phorbol-13-acetate) in the mouse skin (Vav2^{−/−};Vav3^{−/−}+TPA skin), and the human cSCC- and hnSCC-associated transcriptomes. **c** Correlation between VAV2 mRNA levels and the enrichment of the Vav2^Onc-Sig in the indicated gene expression datasets (top). The $P$ value of the F-test assessing the significance of the non-zero slope is indicated. **d** Principal component analysis showing the enrichment of the Vav2^Onc-Sig across the indicated tumor-associated transcriptomes. NSCLC, non-small cell lung cancer; T-ALL, T-cell acute lymphoblastic leukemia. **e** Survival of HPV⁻ hnSCC ($n = 97$) patients according to the levels of the Vav2^Onc-Sig present in tumors. The Mantel–Cox test $P$ values obtained in each analysis are indicated. **f** Survival of hnSCC patients according to levels of the MVGS (minimal Vav2^Onc gene signature) using the indicated datasets (top). The Mantel–Cox test $P$ values obtained in each analysis are indicated. In **a**, **c**, **e**, and **f**, the datasets used for the in silico analyses are shown on top of each graph. Source data for this figure are provided as a Source Data file.

light brown), the large majority of non-small cell lung cancer patient samples (Fig. 10d, area shaded in light blue), and unrelated tumors such as T cell acute lymphoblastic leukemia (Fig. 10d, area shaded in light green). When tested prognostically using the GSE41613 gene expression dataset, we observed that the

Vav2^Onc-Sig can stratify HPV⁻ hnSCC patients according to long-term survival criteria with better resolution (Fig. 10e, $P = 2.4 \times 10^{-4}$) than the abundance of the VAV2 mRNA (Fig. 1b, $P = 0.0079$) and the overall Vav2^Onc-regulated transcriptome (Fig. 10a, $P = 0.0092$). Similar results were found using the cohort

of HPV⁻ hnSCC patients of the TCGA ($P = 0.0068$) and the GSE42743 ($P = 0.036$) datasets. Based on this information, we next generated a second-generation, minimal Vav2^Onc gene signature (MVGS) from the Vav2^Onc-Sig using the Cox proportional-hazards regression model. This MVGS, which only contains 47 probe sets (corresponding to 41 independent Vav2^Onc-regulated genes, Supplementary Table 4), has even better prognostic power than the foregoing gene signatures to predict the long-term survival of the HPV⁻ patients contained in the GSE41613 (Fig. 10f, left panel), the TCGA (Fig. 10f, second panel from left) and in the GSE42743 (Fig. 10f, third panel from left) datasets. In this case, that signature could also stratify the whole cohort of TCGA patients that were not subclassified according to HPV status (Fig. 10f, right panel). However, the differences in the median survival time between the high- and low-risk groups is much smaller in this latter case than in the HPV⁻ patient cohorts (Fig. 10f, compare the right panel with the rest of graphs). This result further indicates that the VAV2-dependent pathobiological program reported here is mainly involved in HPV⁻ tumor malignancy.

## Discussion

Clinical and Pan-Cancer data support the idea that stem cell-like regenerative proliferation and delays in cell differentiation contribute to the malignancy of SCCs and, possibly, other cancer types[1,41–44]. These traits can be elicited by the expansion of cancer stem cell populations within tumors and/or through the dedifferentiation of cancer cells. This process can be even more plastic in the case of cSCC and hnSCC due to intrinsic biological properties of the cell types that originate those tumors. For example, it is currently known that bidirectional transitions exist between hair follicle stem cells and more differentiated epithelial cells as well as between basal and suprabasal keratinocytes under specific physiological conditions[45,46]. Unfortunately, we still have a shallow understanding of the molecular and signaling events underpinning the emergence of these malignant traits and, as a consequence, on how to effectively deal with them at the therapeutic level. In this work, we have shown that the RHO GEF VAV2 is a common driver for these processes in keratinocytes from different anatomical locations. Consistent with this, we have demonstrated that the upregulation of VAV2 catalytic activity promotes the development of a hyperplasic phenotype that, when combined with additional genetic lesions, facilitates tumor development and progression. Perhaps more importantly, we have observed that endogenous VAV2 is required for the maintenance of those malignant traits in both HPV⁻ hnSCC patient-derived cells and established hnSCC cell lines. These effects are associated with the stimulation of a number of transcriptional factors (AP1, c-MYC, E2F, NFY, TEAD) that coordinately work in the orchestration of gene expression changes linked to both stem cell-like regenerative proliferation and delays in cell differentiation in this type of tumors[1,2,43,44,47,48]. Our gain- and loss-of-function experiments have validated the functional correlation of VAV2 activity with the expression of these transcriptional factors in both hyperplasic epithelia and patient-derived cells. They also demonstrated that their stimulation is a direct downstream effect of Vav2^Onc catalysis-dependent signaling. These findings, together with recent observations indicating that the depletion of RAC1 restores chemo and radiosensitivity to hnSCC cell lines[12], suggest that the inhibition of the enzyme activity of this GEF could be a potential therapeutic avenue in a significant percentage of SCCs. In agreement with this possibility, we have recently shown using a Vav2 knock-in mouse strain that the expression of a catalytic hypomorphic Vav2 mutant (L332A) impairs the development of DMBA- and DMBA plus TPA-induced skin tumors[18].

Further supporting its contribution to both cSCC and hnSCC tumorigenesis, we have observed that the abundance of both VAV2 and the VAV2 transcripts becomes exacerbated in some of these tumors. The levels of VAV2 transcripts and Vav2^Onc-driven gene signatures also correlate with the poor prognosis of HPV⁻ hnSCC patients. These results indicate that approximately 30% of cancer patients could be stratified differently based on these molecular features. This proportion is comparable to, and even higher than those found for patients bearing genetic alterations in bona-fide hnSCC drivers such as the EGFR, KRAS, ACTL6A, and c-MYC[1,2]. However, unlike those cases, the deregulation of the VAV2 pathway is not associated with locus-specific genetic alterations. The molecular basis for the aberrant upregulation of VAV2 in these tumors remains to be addressed. Possible explanations include the amplification of super-enhancers and the spurious activation of transcriptional factors[49]. It is also possible that processes associated with increased mRNA translation could contribute to the upregulation of VAV2 signaling. This possibility is consistent with the presence in the 3' untranslated region of the VAV2 mRNA of two cytoplasmic polyadenylation elements usually associated with increased translational efficiencies[50]. Finally, it can be the product of a positive feed-back mechanism in which the normal expression of VAV2 seen in normal basal epithelial cells is further amplified as they progress along the transformation process. In any case, this dysregulation seems to be rather specific since the expression of genes located in the vicinity of VAV2 do not exhibit any kind of dysregulation in the tumor expression datasets analyzed in this work. VAV2 is seldom mutated in hnSCCs according to cBioPortal-archived cancer genome data[24–26], indicating that the overexpression of the WT protein will be the most relevant type of deregulation found for this pathway in this tumor type. Consistent with this, we have found that human keratinocytes ectopically expressing Vav2^WT can also generate a hyperplasic epithelium when tested in organotypic cultures. Conversely, the elimination of the highly expressed endogenous VAV2 protein negatively affects the transforming potential of HPV⁻ hnSCC patient-derived cells. It is worth noting, however, that a recent study has found a recurrent gain-of-function mutation for VAV2 in cases of familial oral SCC[27]. The results obtained with the Vav2^Onc/Onc mouse strain in the present study suggest that this gain-of-function VAV2 mutation, in combination with other genetic alterations, should contribute to the etiology of this type of hereditary cancers. In addition, they provide a pathobiological framework to understand the contribution of mutant VAV2 to this rare hereditary cancer.

Although cSCC and hnSCC share many molecular and pathobiological features with other SCCs[1,2,9,40,51], we have not found any overt deregulation of VAV2 expression in SCCs from other locations. Likewise, the Vav2^Onc-driven diagnostic gene signatures described in this work are only conserved in a very small subset of non-small cell lung cancer patients (Fig. 10d). The reason for such a specificity can be several-fold. It might represent a mechanism to optimally stimulate the YAP/TAZ/TEAD complex, given that the activation of this transcriptomal complex takes place in a Hippo-independent manner in this type of epithelia[3]. Alternatively, it might constitute an adaptive solution to promote the chronic, RAC1-dependent activation of JNK that could contribute to overcome the terminal differentiation-based tumor suppressor mechanism that specifically operates in these cells[52,53]. Supporting this view, the expression of Vav2^Onc does not elicit the terminal differentiation in epithelial cells that has been found before upon the expression of oncogenic versions of phosphatidylinositol 3-kinase α[52]. However, it is also possible that other SCCs could utilize redundant pathobiological programs that make it unnecessary the upregulation of VAV2 signaling. Further work will be needed to address these possibilities.

The full understanding of the role of RHO GEFs and GTPases in cancer has been historically hijacked by a cell biology-centered vision on their roles in cytoskeletal regulation. However, recent data gathered from animal models, genome-wide expression analyses, and Pan-Cancer projects are beginning to illuminate new, and in some cases quite unsuspected roles for these proteins in both tumor promotion and suppression[10,54]. This integrative approach has been instrumental to identify the VAV2-dependent program reported here. Our in silico analyses of gene expression data also have revealed that, in addition to VAV2, four additional RHO GEFs can probably play protumorigenic functions in both cSCC and hnSCC. It will be interesting to investigate in the near future the specific roles of these additional RHO GEF family members in these tumors. Given the results obtained with the single inactivation of VAV2, we postulate that these additional GEFs will play nonredundant functions in these tumors. The experimental approach used here can be also utilized to pinpoint therapeutically interesting RHO signaling elements in any tumor type with a more rational and patient-oriented aprioristic basis.

## Methods

**Ethics statement**. Animal work was done according to protocols approved by the Bioethics committee of Salamanca University. The use of patient samples was conducted in accordance to the Declaration of Helsinki and approved by the Institutional Ethic Committees of the Institute for Research in Biomedicine, Vall d'Hebron Research Institute, and the Hospital Universitario Central de Asturias. All patient samples have been collected with a priori patient consent.

**Plasmids**. To generate the lentiviral vector encoding HA-tagged Vav2 (pCCM33)[55], a cDNA fragment encoding mouse wild type Vav2 was amplified by PCR using as template the pCMV$^-$Vav2-HA plasmid[56]. Primers were modified to introduce a SpeI site (underlined)/HA epitope (bold face) at the 5' end and a NotI site (underlined) at the 3' end in the amplified product. The sequence of the forward and reverse primers were 5'-AAT AAC TAG TGC CAC CAT G**TA CCC ATA CGA CGT CCC AGA CTA CGC T**GA GCA GTG GCG GCA ATG CGG CC-3' and 5'-ATA GAC CGC GGC CGC TCA CTG GAT GCC CTC CTC TTC TAC GTA-3', respectively. The amplified PCR cDNA fragment was digested with the indicated enzymes and cloned into the SpeI/NotI-linearized lentiviral vector (pLVX-IRES-Hyg, Clontech). To generate the lentiviral vector encoding HA-tagged Vav2$^{Onc}$ (pCCM34)[55], a cDNA fragment encoding the HA-tagged version of Vav2 (fragment 187-868) was amplified by PCR using the primers 5'-AAT AAC TAG TGC CAC CAT GTA CCC ATA CGA CGT CCC AGA CTA CGC TAA AAT GGG AAT GAC TGA GGA CGA C-3' and 5'-ATA GAC CGC GGC CGC TCA CTG GAT GCC CTC CTC TTC TAC GTA-3' (restriction sites underlined) and the pCMV-Vav2-HA vector as template[56]. Upon digestion with SpeI and NotI, the cDNA fragment was cloned into the linearized pLVX-IRES-Hyg vector (Clontech, Catalog No. 632185). To generate the vector encoding HA-Vav2$^{Onc+E200A}$ (pFLM12), the pCCM34 vector was subjected to site-directed mutagenesis using the primers 5'-GAC AAG AGA AGC TGC TGC TTG TTA GCG ATT CAG GAG ACC GAG GCC AAG TAC-3' and 5'-CAT GAA CCG GAG CCA GAG GAC TTA GCG ATT GTT CGT CGT CGA AGA GAA CAG-3' (changes used to generate the mutated codon are underlined). The pMIEG3, pMIEG3-Rac1$^{F28L}$, pMIEG3-RhoA$^{F30L}$, and pMIEG3-Cdc42$^{F28L}$ plasmids have been reported before[57]. To generate the plasmid encoding EGFP-Vav2$^{Onc}$ (pNM115), the plasmid pKES19[30] was digested with BstXI, filled-in, and cloned into the SmaI-linearized pEGFP-C2 vector (Clontech, Catalog No. 632481). The plasmid encoding EGFP-Vav2$^{Onc+E200A}$ (pFLM07) was obtained by site-directed mutagenesis of pNM115 using the primers 5'-GAC AAG AGA AGC TGC TGC TTG TTA GCG ATT CAG GAG ACC GAG GCC AAG TAC-3' and 5'-CAT GAA CCG GAG CCA GAG GAC TTA GCG ATT GTT CGT CGT CGA AGA GAA CAG-3' (the altered nucleotides used to create the E200A mutation are underlined). Plasmids encoding c-Myc and the Yamanaka's factors (Oct4, Sox2, Klf4, and c-Myc) were provided by P.M. Fernández-Salguero (Department of Biochemistry and Molecular Biology, Extremadura University, Badajoz, Spain).

The luciferase plasmids used to assay promoter activation included: pAP1-Luc (AP1 reporter, obtained from Stratagene), pBV-Myc-Del1-Luc (c-Myc reporter, obtained from Addgene, Catalog No. 16601), pGL2-Chk1-WT-Luc (E2F reporter, obtained from A.M. Zubiaga, Department of Genetics, Physical Anthropology and Animal Physiology, University of the Basque Country, Bilbao, Spain), p8xGTIIC-Luc (TEAD reporter, obtained from Addgene, Catalog No. 34615), and pRL-SV40 (*Renilla* luciferase, obtained from Promega, Catalog No. E2231). All plasmids were DNA sequence verified.

**In silico analyses of gene expression data from human tumors**. All the gene expression datasets used in our study are detailed in Supplementary Table 1. The rationale for the selection of the expression datasets used for differential expression analyses has been described in the main text. Specifically, GEO datasets GSE45216 and GSE13355 (human cSCC, $n = 220$ samples)[58,59], GSE30784 (human hnSCC, $n = 229$ samples)[21], GSE21264 (mouse cSCC, $n = 273$ samples)[60] and the TCGA hnSCC dataset ($n = 573$ samples)[24] were used to interrogate VAV2 mRNA expression levels (using a 0 to 1 normalization of the expression values to that end). Other datasets included TCGA CESC ($n = 319$)[61] and LUSC ($n = 824$)[62], and the GEO T-ALL dataset GSE26713 ($n = 117$; used in Fig. 10)[63]. The R (version 3.5.1) and Perl programs were used to carry out the statistical analyses and final text processing, respectively. Signal intensity values from microarray data were obtained from CEL files after robust multichip average (RMA). Differentially expressed genes were identified using linear models for microarray data (Limma)[64]. Adjusted $P$ values for multiple comparisons were calculated applying the Benjamini–Hochberg correction (false discovery rate, FDR). The head and neck, cervix and lung SCC TCGA datasets were obtained and analyzed through the TCGAbiolinks R package[65]. When data were analyzed as multiple batches (GSE45216 and GSE13355), fRMA were used to remove batch effects[66]. Overall survival analyses were performed through Kaplan-Meier estimates according to the expression level of the indicated transcripts using the GSE41613 ($n = 97$ samples)[67], the GSE42743 ($n = 103$)[67], and the TCGA (see above) hnSCC datasets. The median of the expression distribution for each of those transcripts was used to establish the low and high expression groups and, subsequently, the Mantel-Cox test was applied to statistically validate the differences between the survival distributions.

**Determination of HPV status of human tumor samples**. HPV detection was performed using p16 immunohistochemistry, high-risk HPV DNA detection by in situ hybridization and genotyping by GP5$^+$/GP6$^+$ PCR[68]. In the first case, tissue microarrays were cut into 3 μm sections and dried on Flex IHC microscope slides (DakoCytomation). Immunohistochemistry was then performed using an automatic staining workstation (Dako Autostainer, Dako Cytomation) with the Envision system and diamino-benzidine chromogen as substrate. The primary antibodies used were those to both p53 (clone DO-7, DAKO) and p16 (clone E6H4 (Roche MTM laboratories AG). In the second case, we performed in situ hybridization with biotinylated HPV DNA probes specific for HPV types 16, 18, 31, 33, 35, 39, 45, 51, 52, 56, 58, 59, and 68 (DakoCytomation) according to the manufacturer's instructions. The results from these two techniques were always evaluated by two independent pathologists. In the latter case, isolated DNAs from tumor samples were subjected to GP5$^+$/6$^+$-PCR using an enzyme-immuno-assay readout to detect the 14 high-risk HPV 16, 18, 31, 33, 35, 39, 45, 51, 52, 56, 58, 59, 66, and 68 subtypes. Subsequent genotyping of the infection was performed using bead-based arrays on the Luminex platform. When GP5$^+$ = 61 PCR was positive, type-specific PCR for HPV16 was performed using primers located in the E7 gene[68].

**Immunoprecipitation experiments**. Healthy mucosa, when included in the experiments, were obtained from the same hnSCC patients (due to this, they were assigned the same identification number shown in Fig. 2a). Tumoral and healthy tissues (Supplementary Table 5) were mechanically homogenized in RIPA buffer (10 mM Tris-HCl [pH 8.0], 150 mM NaCl, 1% Triton X-100 (Sigma-Aldrich, Catalog No. X100), 1 mM Na$_3$VO$_4$ (Sigma-Aldrich, Catalog No. S6508), 1 mM NaF (Sigma-Aldrich, Catalog No. S7920), and a mixture of protease inhibitors (CØmplete; Roche, Catalog No. 11836145001) using a gentleMACS dissociator (Miltenyi Biotec, Catalog No. 130-093-235) and gentleMACS M tubes (Myltenyi Biotec, Catalog No. 130-096-335). Extracts were precleared by centrifugation at 13,200 rpm for 10 min at 4 °C. Protein concentration in the resulting supernatants was quantified using the Bradford reactive (Bio-Rad, Catalog No. 5000006). 400 μg of protein extracts in a final volume of 500 μl of RIPA buffer were incubated with 1.5 μl of a homemade antibody to Vav2 (Lab catalog No. 580-2) for 2 h at 4 °C in a rotating wheel. Immunocomplexes were collected with 35 μl of Gammabind G-Sepharose beads (GE Healthcare, Catalog No. GE17-0885-01) at 50% concentration for another 2 h at 4 °C in a rotating wheel. After three times washes with the lysis buffer at 4 °C, the beads were resuspended in SDS-PAGE buffer, boiled for 5 min, and transferred onto nitrocellulose filters using the iBlot Dry Blotting System (Thermo Fisher Scientific, Catalog No. IB21001). Membranes were blocked in 5% bovine serum albumin (Sigma-Aldrich, Catalog No. A7906) in TBS-T (25 mM Tris-HCl (pH 8.0), 150 mM NaCl, 0.1% Tween-20 (Sigma-Aldrich, Catalog No. P7949)) for at least 1 h and then incubated overnight at 4 °C with antibodies to Vav2 (1:1000 dilution, see above). After stripping, the same filters were used to carry out a second Western blot analysis using antibodies to phospho-tyrosine residues (1:1000 dilution; Santa Cruz Biotechnologies, Catalog No. sc-7020). After three washes with TBS-T to eliminate the primary antibody, the membrane was incubated with the appropriate secondary antibody (1:5000 dilution, GE Healthcare) for 30 min at room temperature. Immunoreacting bands were developed using a standard chemiluminescent method (Thermo Fisher Scientific, Catalog No. 32106). Blots were quantified using the ImageJ software (NIH, www.imagej.nih.gov).

In addition to protein quantitation, the total amount of protein present in each tissue lysate was confirmed by running aliquots of the total cellular extracts used in the immunoprecipitation experiments in SDS-PAGE gels and, after transfer to

nitrocellulose filters as above, staining with Ponceau S solution (Sigma-Aldrich, Catalog No. P7170).

In the case of keratinocyte cultures, exponentially growing cells were washed in a phosphate-buffered saline solution and lysed with the aid of a scrapper in 1 ml of RIPA buffer supplemented with CØmplete. Cellular extracts were kept 5 min on ice and subsequently centrifuged at 13,200 rpm for 10 min at 4 °C to eliminate cell debris. After protein quantitation, lysates were incubated with 1.5 µl of the homemade antibody to Vav2 and processed as above.

**Western blotting analyses of total cell extracts**. In the case of tissue extracts, samples were transferred to RIPA buffer and mechanically homogenized using the gentleMACS dissociator. In the case of keratinocytes maintained in culture, cells were washed with chilled phosphate buffered saline solution and then directly lysed in RIPA buffer at 4 °C. Extracts were precleared by centrifugation at 13,200 rpm for 10 min at 4 °C, denatured by boiling in SDS-PAGE sample buffer, separated electrophoretically, and transferred onto nitrocellulose filters using the iBlot Dry Blotting System. Membranes were blocked as above and then incubated overnight at 4 °C with appropriate antibodies to Vav2 (1:1000 dilution, homemade), GFP (1:5,000 dilution; Covance, Catalog No. MMS-118P), the hemagglutinin (HA) epitope (1:1,000 dilution; Cell Signaling, Catalog No. 3724), Rac1 (1:1,000 dilution; BD Biosciences, Catalog No. 610651), RhoA (1:1,000 dilution; Cell Signaling, Catalog No. 2117), Cdc42 (1:1,000 dilution; Santa Cruz Biotechnology, Catalog No. sc-87), c-Myc (1:1,000 dilution; Merck, Catalog No. 06-340), and tubulin α (1:2,000 dilution; Calbiochem, Catalog No. CP06). After three washes with TBS-T to eliminate the primary antibody, immunoreacting bands were visualized as indicated above.

**Immunohistochemistry using patient samples**. Surgical tissue specimens from patients with hnSCC who underwent resection of their tumors at the Hospital Universitario Central de Asturias between 1990 and 2010 were retrospectively collected. The formalin-fixed, paraffin-embedded tissues were cut into 3 µm sections and dried on Flex IHC microscope slides (Agilent, Catalog No. K8020). The sections were deparaffinized with standard xylene and hydrated through graded alcohols into water. Antigen retrieval was performed using Envision Flex Target Retrieval solution, high pH (Dako Omnis, Catalog No. GV804). Staining was done at room temperature on an automatic staining workstation (Dako Omnis, Catalog No. Autostainer Plus) with rabbit monoclonal antibody to VAV2 (Abcam, Catalog No. ab52640; 1:50 dilution). Hematoxylin was used for counterstaining. The slides were randomly analyzed and signals scored as negative (0), weak-to-moderate (1) or strong (2). 96.55% of all the tumor samples used in these analyses were HPV⁻.

**Mouse experiments**. $Vav2^{Onc/Onc}$ and $Vav1^{-/-};Vav2^{-/-};Vav3^{-/-}$ mice have been described elsewhere[17,29]. Animals were kept in ventilated rooms in pathogen-free facilities under controlled temperature (23 °C), humidity (50%), and illumination (12-h-light/12-h-dark cycle) conditions. For carcinogenesis experiments in newborn mice, 3-day-old mice received a single application of 5 µg of DMBA (Sigma-Aldrich, Catalog No. D3254) in 200 µl of acetone. The number, size, and incidence of tumors were monitored weekly. Time-course studies of the in vivo response of the epithelia of WT and $Vav2^{Onc/Onc}$ mice to TPA using both determination of epithelial hyperplasia and BrdU incorporation were carried out as indicated before[17]. Upon euthanasia, tumors and tissue samples were collected and processed for both histological and immunohistochemical analyses.

For the orthotopic xenograft experiments, exponentially growing hnSCC patient-derived cells were detached using Accutase (CELLnTEC, Catalog No. CnT-Accutase-100), centrifuged at 300 × g for 5 min, and resuspended in culture medium at a density of $1.7 × 10^6$ cells per ml. $5 × 10^4$ SCC cells were then introduced orthotopically using an ultrafine 8 mm-needle (BD, Catalog No. 320927) in the tongue of 6- to 8-week-old $Vav1^{-/-};Vav2^{-/-};Vav3^{-/-}$ mice previously anesthetized by injecting intraperitoneally 200 µl of a mixture of 10 mg/ml ketamine (Merial, Catalog No. Imalgene 1000) and 2 mg/ml xylazine (Bayer, Catalog No. Rompun) in phosphate buffered saline. Growth of tumors was then periodically visualized using an IVIS Lumina In Vivo Imaging System (PerkinElmer, Catalog No. CLS136334) upon administering 50 µl of a 5 mg/ml solution of D-luciferine (Goldbio, Catalog No. LUCK-500) to mice. This procedure was performed with animals anesthetized as above. Mice were reanimated after finishing the recordings by injecting intraperitoneally 200 µl of a 1 mg/ml atipamezole (Ecuphar, Catalog No. Antisedan) solution in phosphate buffered saline solution. Upon euthanasia, tumors were collected and processed for pathological and immunohistochemical analyses.

When needed, tissues were extracted, fixed in 4% paraformaldehyde, paraffin embedded, cut in 2–3 µm thick sections, and stained with hematoxylin-eosin. The thickness of the epidermal layer and/or total skin thickness were measured in vertical cross-sections using the software ImageJ. In the case of tumor sections, they were analyzed blindly by a pathologist to classify them according to malignancy grade and level of differentiation.

**Immunohistochemical studies in mice**. Immunohistochemical staining was performed using a Ventana Discovery Ultra instrument (Roche, Catalog No. 05987750001). Paraffin-embedded sections were dewaxed, subjected to either citrate buffer [pH 6.0] or Tris EDTA [pH 8.0] for heat-induced antigen retrieval, and incubated for 40 min with the appropriate primary antibody to c-Fos (1:50 dilution, Santa Cruz Biotechnology, Catalog No. sc-166940), c-Myc (1:50 dilution, Abcam, Catalog No. ab32072), YAP (1:200 dilution, Novus Biologicals, Catalog No. NB110-58358), Cyclin D1 (1:200 dilution, Roche, Catalog No. 790-4508), keratin 14 (1:300 dilution, Biolegend, Catalog No. 905301), or involucrin (1:100 dilution, Sigma-Aldrich, Catalog No. I9018). For standard staining, the Discovery OmniMap anti-Rb HRP detection system (Roche, Catalog No. 760-4311) was used for detection and hematoxylin for counterstaining as indicated by the manufacturer. Quantitation of immunohistochemical signals were done using the ImageJ software by adjusting the signal threshold to the staining intensity and measuring the area of the resulting particles.

**Cells**. Primary keratinocytes were isolated as described elsewhere[17]. Briefly, the skin from newborn mice of the indicated genotypes was treated with 250 U/ml of dispase (Roche, Catalog No. 04942078001) overnight at 4 °C. Next day, the epidermis was separated from the dermis and treated with Acutase for 30 min at 37 °C to extract the keratinocytes. Cultures were maintained in CnT07 medium (CELLnTEC, Catalog No. CnT-07) on type I collagen-precoated plates (BD Biosciences, Catalog No. 356400).

Immortalized, primary neonatal human keratinocytes (KerCT cells) were purchased from the American Type Culture Collection (Catalog No. CRL-4048) and cultured in KGM-Gold medium (Lonza, Catalog No. 00192060). These cells have been immortalized by ectopically expressing human TERT and CDK4. SCC-25 cells were obtained from S.A.B. and cultured in KSFM medium (Thermo Fisher Scientific, Catalog No. 17005-042) supplemented with 25 µg/ml bovine pituitary extract (Thermo Fisher Scientific, Catalog No. 17005-042) and 0.5 ng/ml human epidermal growth factor (Thermo Fisher Scientific, Catalog No. 17005-042).

hnSCC (tongue-located) patient-derived cells were described elsewhere[35]. VdH15 cells were cultured in KSFM medium supplemented with 25 µg/ml bovine pituitary extract and 0.5 ng/ml human epidermal growth factor (Thermo Fisher Scientific, Catalog No. 17005-042). VdH01 cells were cultured in a FAD⁺ medium that was composed of three parts of DMEM (Gibco, Catalog No. 21969) and one part of Ham's F12 medium (Thermo Fisher Scientific, Catalog No. 11765054) supplemented with 10% fetal bovine serum (Gibco, Catalog No. 10270106), $1.8 × 10^{-4}$ M adenine (Sigma-Aldrich, Catalog No. A2786-5G), 0.5 µg/ml hydrocortisone (Sigma-Aldrich, Catalog No. H4001-1G), 5 µg/ml insulin (Thermo Fisher Scientific, Catalog No. 12585014), 10 ng/ml epidermal growth factor (PreproTech, Catalog No. AF-100-15), $10^{-10}$ M cholera toxin (Sigma-Aldrich, Catalog No. C8052-5MG), and 2 mM L-glutamine (Gibco, Catalog No. 25030024).

**Three-dimensional organotypic cultures**. Exponentially growing keratinocytes maintained in CnT-Prime medium were detached using Accutase, centrifuged at 300 × g for 5 min at room temperature, and counted. $2 × 10^5$ cells were seeded onto polycarbonate inserts (Thermo Fisher Scientific, Catalog No. 140620) and cultured for 2 days in CnT-Prime medium. Upon confluency, the medium was changed to 3D-Barrier (CellnTec, Catalog No. CnT-PR-3D) and the air-lift was performed according to the manufacturer's instructions. 3D cultures were maintained for 12 days with three medium changes per week and finally processed for histological study as indicated above. When indicated, the inhibitors and corresponding vehicles were applied in the sixth day after carrying out the airlift and refreshed with the medium changes indicated above. Inhibitors used included FRAX597 (5 nM; Selleckchem, Catalog No. S7271), Y-27632 2HCl (1 µM; Selleckchem, Catalog No. S1049), 1A116 (500 nM)[39], 10058-F4 (500 nM; Selleckchem, Catalog No. S7153), and verteporfin (200 nM; Selleckchem, Catalog No. S1786). Concentrations of inhibitors were selected based on the induction of minor effect in the organotypic structures formed by control cells.

**Lentiviral-mediated expression of proteins in keratinocytes**. For the generation of stable cell clones, lentiviral particles were produced in HEK293T cells cultured in DMEM (supplemented with 10% fetal bovine serum and 2 mM L-glutamine) and subsequently concentrated using the LentiX concentrator kit (Clontech, Catalog No. 631232). Keratinocytes were then infected with control and protein-encoding lentiviral particles in KGM-Gold medium in the presence of 8 µg/ml of polybrene (Sigma-Aldrich, Catalog No. H9268) and selected by either antibiotic resistance or EGFP expression by flow cytometry. In the latter case, we used a FACSAria III flow cytometer (BD Biosciences) for cell purification. Transduced cells were puromycin selected as described elsewhere[17]. Proper protein expression was assessed by Western blot.

**RHO GTPase activation assays**. Total cellular lysates were obtained as above, snap frozen, thawed, quantified for total protein concentration, and analyzed using the Rac1, RhoA and Cdc42 G-LISA assay kits according to the manufacturer's instructions (Cytoskeleton, Catalog No. BK135).

**Microarray experiments**. Total RNA from the back epidermis of 8-week-old mice of indicated genotypes ($n = 3$ per genotype) was isolated using the RNAeasy Mini Kit (Qiagen, Catalog No. 74104) and analyzed using Affymetrix platform (GeneChip Mouse Gene 1.0 ST arrays) as previously described by us[17,55,69].

**In silico analyses of mouse expression microarray data.** Initial data processing was carried out as indicated for the human data (see above). Gene Ontology and KEGG pathways enrichment analyses were performed using DAVID and ToppFun. GSEA was performed using gene set permutations ($n = 1000$) for the assessment of significance and signal-to-noise metric for ranking genes. The gene sets used included signatures related to: epidermal differentiation (GSE52954)[70]; cSCC stem cells (GSE29328)[71]; embryonic stem cells (GSE10423)[72]; cSCC cells (GSE5576)[73]; and metastatic hnSCC cells (GSE72939)[35]. We also used lists for transcriptional targets for AP1[74], c-MYC[75], E2F (signature #M40 from the Molecular Signatures Database, Broad Institute), NFY (signature #C3:TFAC:0178 from the Molecular Signatures Database, Broad Institute), YAP/TAZ/TEAD (GSE66083)[5], and FOXO (signature #C3:TFAC:0256 from the Molecular Signatures Database, Broad Institute). To evaluate the similarity between the transcriptional programs activated by Vav2$^{Onc}$ and those differentially regulated in the indicated SCC and clinical conditions, the datasets GSE45216 and GSE13355 (human cSCC), GSE30784 (human hnSCC), and GSE21264 (mouse cSCC) were used as indicated above. To assess conservation of the Vav2$^{Onc}$ transcriptional signature across tumors, a 1.5-fold change threshold was used. Heatmaps were generated using the heatmap3 package. To evaluate the enrichment of the Vav2$^{Onc}$ gene signature across SCC tumors, ssGSEA were used.

For the discovery of transcription factor binding motifs in the promoters of the coregulated genes, the iRegulon software was used[47]. A collection of 9713 position weight matrices (PWMs) was applied to analyze 10 kb-long DNA sequences that were centered around the transcriptional start sites. Using as criteria a maximum false discovery rate (FDR) on motif similarity below 0.001, we performed motif detection, track discovery, motif-to-factor mapping, and target detection. Circos plots were generated using the OmicCircos package implemented with mouse genomic data from the UCSC Genome Browser (https://genome.ucsc.edu), gene ontology annotation from the Gene Ontology Consortium (http://www.geneontology.org), and transcription factor annotation from the iRegulon tool.

To evaluate the correlation of c-MYC or YAP gene signature with *VAV2* mRNA levels in tumors, we stratified patients according to *VAV2* expression levels (low, medium, and high, established by quantiles) and calculated the ssGSEA enrichment score for the aforementioned signatures.

**Generation of induced pluripotent stem cells.** Retroviral particles encoding the Yamanaka's factors were used to infect primary keratinocytes obtained from neonate mice in the presence of polybrene (8 μg/ml) and, after 2 days, transferred to an iPSC medium composed of DMEM-GlutaMAX (Thermo Fisher Scientific, Catalog No. 31966-021), 15% knockout serum replacement (Thermo Fisher Scientific, Catalog No. 10828-028), 1x nonessential amino acids (Thermo Fisher Scientific, Catalog No. 11140-035), 100 μM 2-mercapthoethanol (Thermo Fisher Scientific, Catalog No. 31350-010), and $1 \times 10^3$ u/ml LIF (Millipore, Catalog No. ESG1107). Three days later, cells were collected using Acutase and seeded on feeder cell layers generated as described elsewhere[76]. iPS cell colonies were allowed to grow for two weeks, with daily medium changes, and finally stained with an alkaline phosphatase assay kit (Sigma-Aldrich, Catalog No. AB0300) according to the manufacturer's instructions. The generation of bona-fide iPS cells was further confirmed using qRT-PCR analysis to detect *Nanog* expression.

**Analysis of mRNA abundance.** Total RNA was extracted using NZYol (NZYtech, Catalog No. MB18501) and quantitative RT-PCR performed using the Power SYBR Green RNA-to-CT 1-Step Kit (Applied Biosystems, Catalog No. 4389986) and the StepOnePlus Real-Time PCR System (Applied BioSystems, Catalog No. 4376600). Raw data were analyzed using the StepOne software (Applied Biosystems). We used the abundance of the endogenous *GAPDH* mRNA as internal normalization control. Primers used for transcript quantitation included: 5′-GAC GGG GAA CTG AAA GTC CG-3′ (forward, *VAV2*), 5′-TTT TCC CGT GAG ACT TCT TGA C-3′ (reverse, *VAV2*), 5′-ATG GCC TTC CGT GTC CCC ACT G-3′ (forward, *GAPDH*), 5′-TGA GTG TGG CAG GGA CTC CCC A-3′ (reverse, *GAPDH*), 5′-AAG TAC CTC AGC CTC CAG CA-3′ (forward, *Nanog*), 5′-GTG CTG AGC CCT TCT GAA TC-3′ (reverse, *Nanog*), 5′-TGC ACC ACC AAC TGC TTA GC-3′ (forward, *Gapdh*), 5′-TCT TCT GGG TGG CAG TGA TG-3′ (reverse, *Gapdh*).

**Promoter activation assays.** We utilized luciferase reporter plasmids to evaluate the activation status of the interrogated transcriptional factors of our study. To this end, exponentially growing human keratinocytes were transiently transfected with 40 ng of the pRL-SV40 vector encoding the *Renilla* luciferase for intersample normalization, 2 μg of the appropriate EGFP and EGFP-Vav2 encoding vector, and 2 μg of the appropriate firefly reporter plasmids containing the firefly luciferase gene under the regulation of AP1- (pAP1-Luc), c-Myc- (pBV-Myc-Del1-Luc), E2F- (pGL2-Chk1-WT-Luc), and TEAD- (p8xGTIIC-Luc) responsive promoters. After 24 h, cells were lysed with Passive Lysis Buffer (Promega, Cat No. E1960) and luciferase activities determined using the Dual Luciferase Assay System (Promega, Cat No. E1960). In all cases, the values of firefly luciferase activity obtained in each experimental point were normalized taking into account the activity of the *Renilla* luciferase obtained in the same samples. In addition, we analyzed aliquots of the same lysates by Western blot to assess the appropriate expression of the ectopically expressed proteins in each case. Values are represented in the figures as the n-fold change of the experimental sample relative to the activity shown by control cells (which was given an arbitrary value of 1 in each case). When needed, these experiments included the use of chemical inhibitors in concentrations identical to those used in the 3D organotypic cultures.

**Genomic characterization of hnSCC patient-derived cells.** DNA samples from indicated patient-derived cells were sequenced at Macrogen (Seoul, South Korea) to a 30× depth using 150 bp paired-end reads on Illumina NovaSeq 6000. Sequenced reads were mapped to human genome assembly GRCh37 using BWA-MEM (v0.7.17)[77] and duplicate reads were marked using Biobambam (v2.0.87)[78]. For variant calling, GATK tools (v4.1.4.1)[79] were used to detect substitutions and indels. Variants per sample were called using HaplotypeCaller in GVCF mode and a join-call cohort generated using both the CombineGVCFs, GenotypeGVCFs, and VariantFiltration tools. Structural variants were called using the germline implementation of DELLY (v0.8.1)[80]. Copy number status was inferred using Canvas (v1.40.0)[81]. For variant annotation, we used the Ensembl Variant Effect Predictor (v100.2)[82]. Substitutions and indels were defined as variants with potential clinical significance when meeting the following criteria: (i) The highest allele frequency observed for the variants in any population from 1KGP, ESP or gnomAD were less than 0.01 or not described. (ii) The variants were predicted to have high impact on protein function or to be deleterious based on SIFT and PolyPhen scores. Variants reported as pathogenic by ClinVar were also selected. (iii) Variants located within the Cancer Gene Census[83]. Known cancer hotspots were pinpointed using the Cancer Hotspots database[84]. Copy number alterations were considered clinically relevant if they involved oncogenic gains or tumor suppressor deletions. For structural variants, those affecting Cancer Gene Census genes at breakpoint level were included.

**shRNA-mediated transcript knockdown.** To target the *VAV2* mRNA, we infected the indicated hnSCC patient-derived and SSC-25 cells with lentiviruses encoding either empty (TR1.5-pLKO-1-puro; Sigma-Aldrich, Catalog No. SHC001) or a number of independent *VAV2* shRNAs (TRCN0000435647, referred to in the figures as shRNA#1; TRCN0000436728, referred to in the figures as shRNA#2; TRCN0000048227, referred to in the figures as shRNA#3; TRCN0000418821, referred to in the figures as shRNA#4 and TRCN0000419630, referred to in the figures as shRNA#5). Lentiviral vectors encoding those shRNAs were obtained from Sigma-Aldrich. Lentiviral particle production and infections were done as indicated above. Transduced cells were puromycin selected as described elsewhere[17]. Proper transcript knockdown was assessed by qRT-PCR.

**Development of diagnostic signatures.** The criteria used to develop the Vav2$^{Onc}$-Sig have been described in the Results section. To assess its tumor specificity, we performed principal component analyses using the datasets indicated in the figures. For the development of the MVGS from the Vav2$^{Onc}$-Sig, we carried out Cox proportional-hazards regression model using the survival R package (https://cran.r-project.org/web/views/Survival.html). Survival analyses were performed as indicated above, using datasets from the GEO (GSE41613) and the TCGA (hnSCC collection).

**Statistics.** The number of biological replicates ($n$), the type of statistical tests performed, and the statistical significance are indicated for each experiment either in the figure legends or in the main text. Data normality and equality of variances were analyzed with Shapiro-Wilk and Bartlett's tests, respectively. Parametric distributions were analyzed using Student's $t$-test (when comparing two experimental groups) or ANOVA followed by either Dunnett's (when comparing more than two experimental groups with a single control group) or Tukey's HSD test (when comparing more than two experimental groups with every other group). Nonparametric distributions were analyzed using either Mann-Whitney (for comparisons of two experimental groups) or the Kruskal-Wallis followed by Dunn's (for comparisons of three or more than three experimental groups) tests. The chi-squared test was used to determine the significance of the differences between expected and observed frequencies. In all cases, values were considered significant when $P \leq 0.05$. Data obtained are given as the mean ± SEM.

**Reporting summary.** Further information on research design is available in the Nature Research Reporting Summary linked to this article.

## Data availability

A Source Data file is provided with this paper. The microarray data reported in this paper has been deposited in GEO database under accession number GSE124019. In addition, public GEO datasets GSE45216, GSE13355, GSE30784, GSE21264, GSE52651, GSE26713 and the TCGA hnSCC, CESC, and LUSC datasets have been used to interrogate *VAV2* enrichment levels. Survival analyses have been performed using the GSE41613, GSE42743, and the TCGA HNSC datasets. GSEA were performed using the datasets GSE52954, GSE29328, GSE10423, GSE5576, GSE72939, and GSE66083. All relevant data are available from the authors.

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

## Acknowledgements

We thank M. Blázquez and CIC facilities' personnel for lab work and core unit technical assistance, respectively. X.R.B. is supported by grants from Worldwide Cancer Research (14-1248), the Castilla-León Government (CSI049U16, CLC-2017-01), the Spanish Ministry of Science and Innovation (MSI) (SAF2015-64556-R, RTI2018-096481-B-I00), the Ramón Areces Foundation, and the Spanish Association against Cancer (GC16173472GARC). X.R.B.'s institution is supported by the Programa de Apoyo a Planes Estratégicos de Investigación de Estructuras de Investigación de Excelencia of the Ministry of Education of the Castilla-León Government (CLC-2017-01). J.M.P. is supported by MSI grants (SAF2015-66015-R, PIE15/00076). S.A.B. was supported by the European Research Council, the Spanish MSIU, and the IRB-Barcelona. S.R.-F. and L.F. L.-M. contracts have been supported by funding from the MSI (BES-2013-063573) and the Spanish Ministry of Education, Culture, and Sports (L.F.L.-M., FPU13/02923), respectively. Both Spanish and Castilla-León government-associated funding is partially supported by the European Regional Development Fund.

## Author contributions

L.F.L.-M. participated in all experimental work, analyzed data, and contributed to both artwork design and manuscript writing. N.F.-P. and M.C. performed experiments with Vav2$^{WT}$-expressing keratinocytes. M.M.-M. and S.F. initiated the study. S.R.-F. and A.A. were involved in animal experimentation procedures. J.R.-V. characterized the immune system of Vav2$^{Onc}$ mice. G.P and S.A.B. contributed to work related to patient-derived cells. J.M.G.-P., M.A.P., R.G.-S., and J.P.R. provided SCC patient samples and data. S.Z. and J.M.C.T carried out the genomic characterization of patient-derived cells. M.C.G.-M., C.S., and J.M.P performed histopathological analyses. N.G. and P.L.-M. developed Rac1 inhibitors and contributed to some experiments done with them. X.R.B. conceived the work, analyzed data, wrote the manuscript, and performed the final editing of figures.

## Competing interests

The authors declare no competing interests.
