## [Peer Review File · Nature Communications]

Reviewers' comments:

Reviewer #1 (Remarks to the Author): Expert in Rho signalling

The study by Bustelo et al., is based on the observation that VAV2, a guanine nucleotide exchange factor for Rho GTPases, is frequently upregulated in cutaneous and head and neck cancer squamous cell carcinoma (SCCs). The authors propose that this overexpression is causative in terms of expansion of epithelial stem cells and subsequent carcinogenesis, based on the definition of VAV2-induced gene programs using cells and transgenic mice expressing VAV2 mutants. They also take advantage of a new "pharmacomimetic mice" that attempts to explore the impact of VAV2 inhibition within specific thresholds on tumorigenesis. The study is highly innovative. However, it may over rely (or over interpret) bioinformatics analyses performed on distinct subsets of SCCs, without defining the biological and clinical settings of each database used, and choosing/picking those that fit their statistical threshold without a specified rationale. For example, most correlations are performed in oral cancer datasets that may have extensive influence by their HPV status, and cell studies on oral cancer cells without much in terms of background info and compared to immortal skin cells. Most animal model studies are instead performed in skin cancer models. In addition, the authors appear to make multiple statements on causality based on correlative observations, and they do not consider that VAV2 tyrosine phosphorylation rather than expression levels may reflect better VAV2 activity, unless mutationally activated.

The datasets used for the different analyses are not defined at all. Only their corresponding GEO number is included, without any background regarding their corresponding publication and patient stratification. This becomes critical, as the authors appear to pick the ones aligned with their interest and expected outcome without a defined criteria, often relying on the results from one dataset to support one conclusion and other datasets for others. Each of the diseases is different, with different etiology and mutational landscape. For example, it is unclear what is the distinction between oral SCC (oSCC) and hnSCC, and unclear why the authors pick one or the other as the best one to support their conclusion. In principle, one of them may reflect viral infection (HPV), which is clearly distinct anatomically and genetically from tobacco-related cancers, with different response to treatment and prognosis. Similar to skin SCC (sSCC), whose causative role and driver oncogenes are clearly different from the others. In this regard, the use of "head-and-neck" SCC (best known simply as head and neck SCC or head and neck cancer) seems odd. The frequent statement that VAV2 is involved in "poor patient prognosis" may not be correct, as a cause effect relationship has not been established. Perhaps correlation or association would be a preferred way to refer to this potential functional relationship.

The authors may need to make a concerted effort on defining their clinical groups and corresponding datasets, and defining the criteria used to reach their conclusions for all SCCs or rationale for why specific subsets are chosen (or their conclusions more valid) with respect of the other ones. This perceived problem spills into the experimental approaches as most of the carcinogenesis models used (in which they monitor SCC formation as compared to benign hyperplasia) rely on chemical carcinogenesis models that only apply to sSCC, hence one would expect that the primary analysis would involve human sSCC and then exploring for commonalities with other SCCs.

For example, on page 6, the authors write: "Further highlighting the possible involvement of VAV2 in SCC, we found that the abundance of its transcripts correlated with shorter survival rates of oSCC patients (Fig. 1c)." Why only on oSCC? Also, "This stratification power is even better from a statistical point of view than that shown by the abundance of the mRNA for EGFR, a tyrosine kinase receptor frequently amplified in hnSCC (Figs. 1c and S1b). This stratification power is lost when using hnSCC patients that had not been separated according to the anatomical location of tumors (data not shown)." It seems that the authors are cherry picking information from these two groups of SCCs, without a strong rationale or statistical analysis of differences, including a statement of better prognostic value with respect to EGFR, without the corresponding statistical support. In addition, the dataset used for analysis of VAV2 expression in SCC (including oSCC),

and correlation with survival in oSCC are different?

Tumors are heterogeneous, with abundant immune cell and stromal contamination. This may influence the mRNA and protein expression throughout the study. Especially when addressing the relationship between VAV2 expression and "stage" and "tumor size". Is VAV2 perceived higher mRNA and protein expression due to immune/inflammatory infiltration?

On page 7, the authors comment that oSCC have a common hypopharyngeal origin, an anatomical area usually associated with very large cancerization fields. Is this HPV related? If so, what is the relevance of the comparison with non-HPV related lesions and experimental systems?

example, the authors compare all oSCC cell lines to KerCT, an immortalized, nontransformed primary human keratinocyte cell line derived from the skin.

The origin and description (and mutational status) of the cell lines used is also not described, the authors used several cells, but how they relate to the oSCC that they try to model (including HPV status) is not described, and why they compare the information to skin normal cells need to be addressed or corrected experimentally.

In figs 2A-B, is proliferation observed in the basal layers or suprabasal?

The rationale for the selection of the particular Rho GTPase mutants is not specified.

At the core of the uncertainty of this reviewer regarding the suitability of this study is the observation that the overall levels of the raw Vav2Onc-dependent transcriptome correlate with the poor prognosis of oSCC patients (Fig. 4d), but not sSCC, aligned with the comments (above) regarding concerns about the appropriate selection of datasets to compare with.

The analysis showing that c-Fos (an AP1 family member), c-Myc, Cyclin D1, and YAP are upregulated in the epidermis of Vav2Onc/Onc mice is of interest, but is this also observed in all proliferating (normal) cells? If so, how do they assess direct cause-effect relationship? It appears that this may be a consequence of cell proliferation. Same for experiments using cSCCs collected from the DMBA-treated Vav2Onc/Onc mice.

The Vav2L332A/L332A mice showed reduced tumorigenesis when subjected to the standard DMBA skin carcinogenesis method. However, VAV2 is affected in all tissues/cells, and it is unclear whether the effect is epithelial cell intrinsic or due to changes in immune infiltration or function, and inflammatory processes that characterize this tumor model (which in any case is compared to oSCC that is not what is being modeled).

In addition:

In Fig 1a, what is the dataset used to define oral dysplasia and criteria used?

In Fig 1c, what is the HPV status of the different VAV2 expression subgroups? It is possible that HPV+ patients, which have better prognosis, rather than VAV2 expression, may correlate with prognosis. What is the survival curve for sSCC patients, which is what they are modelling experimentally?

In Fig 1h, what is the HPV status of these cells, and why were they selected?

Reviewer #2 (Remarks to the Author): Expert in SCC

Lorenzo-Martin et al. provide an overwhelming amount of data on Vav2 and squamous cancer. The experiments are carried out well. Generally the paper is not very well focused and should in my opinion be focused more on the key findings. The key findings to me are Figs, 1, 2, 3, and 5. Fig 4 is too detailed and in fact is only relevant for the identification of the potential transcription factors involved. Fig 6 is less convincing by the use of shRNAs but do support the picture while Fig 7 is really out of scope. The key question for me in this MS is the role of Vav2, Rac1 and perhaps RhoA in squamous cell biology and their role in dedifferentiation and cancer proneness. The authors put Yes Myc Yap and E2F and AP1 forward as players, but how they precisely relate to increased Vav2-Rac1 signaling remains elusive.

Convincing is the hyperplasia in squamous cells, and the authors build a strong case on that phenotype. Also convincing is the proneness to cancer, but using DBMA is not the most elegant way and also the phenotype of the tumors is somewhat odd. Crossing with p53 and/or p16 KO mouse models and obtaining mucosal SCCs as phenotype, would have been stronger.

I miss convincing direct evidence of Vav2-Rac1 signaling and the phenotype. The presented evidence is very indirect.

There are other concerns:

- There are many typo's in the MS and the Abstract is very vague. It would help when it is corrected by a native English speaker.
- In 3f right low panel, the magnification seems different.
- The authors make a point on HNSCC and OSCC but their mouse phenotypes are skin SCCs and then also of sebaceous gland types. This is not very convincing, but might relate to the mouse model.
- There is prognostic impact in OSCC but not in HNSCC in general, which seems remarkable but neither the data are shown nor the patient characteristics. Moreover there was no multivariate analysis performed. Was Vav2 expression related to T stage, N stage, age, histology (differentiation!), HPV status in oropharyngeal cancers? Did that interfere with prognostic impact?
- Vav2 mutations are very rare in HNSCC (2.8%Vav2, 5% Rac1, and 2.8% RhoA, and some are inactivating). Moreover, the activating mutations may not even be functionally comparable to the VAFonc mutation. This was not studied at all and is a flaw in the MS.
- The authors suggest field cancerization in hypopharynx, but it is everywhere and should either be studied by genetic markers or this attempt to explain unwanted observations removed.
- Expression levels of Vav2onc may be changed by the large deletion and this is not shown. Perhaps this was demonstrated in a previous study, but should be presented here as well.
- Specificity of VAFonc for substrate GTPases may have been changed. Should Ras not have been investigated in the G-ELISA?

Reviewer #3 (Remarks to the Author): Expert in mouse models

Rho-GTPases are long thought to contribute to tumorigenesis. Still, how their deregulation contributes to carcinogenesis remains largely elusive. In this manuscript, Lorenzo-Martin et al. provide compelling evidence that increased Vav2 activity promotes squamous carcinogenesis in skin and oral epithelia. This manuscript is well written, the experimental data are interesting, clearly presented, and correctly interpreted. The main strength of the study are the gain and loss of function mouse models demonstrating a role for Vav2 in squamous carcinogenesis, combined with links to published patient data and potential mechanistic links that had been established by various studies in the field. Although this paper stops short in providing a direct molecular mechanism for how Vav2 activity controls a cancer promoting transcriptional program and it remains unclear whether the transcriptional changes are a direct or indirect result of aberrant Vav2 function, it provides compelling evidence that pharmacological Vav2 inhibition can inhibit SCC growth. In conclusion, this is an interesting and well executed study that seems suitable for publication in Nature Communications after some minor questions have been addressed.

Specific comments:

- It is unclear where within the stratified epithelium and a SCC Vav2 is expressed and active. What is its sub-cellular localization?
- How does Vav2 become over-expressed in SCCs? What regulates its expression?
- The authors provide a significant amount of meta analyses of previously published data sets, providing evidence that Vav2 expression is up-regulated in human SCCs and linked to poor outcome in patients. They also compare Vav2 dependent changes in gene expression to various other, previously published data sets. These analyses are well presented and they significantly strengthen the work. However, the authors only reference the data sets in the experimental methods sections and they fail to cite the papers that provided the basis for their meta analyses. The authors must clearly and adequately reference the papers that generated the data sets for their meta analyses within the main text of the manuscript as it is difficult for the reader familiar with the field to understand what studies have contributed to the conclusions of the presented work.
- Upon closer inspection of the previously published data sets that have been used for the meta analyses in this paper it becomes apparent that normal and psoriatic patient data were compared to SCC data generated by different labs in different papers. Although the presented approach resulted in compelling evidence for how aberrant Vav2 expression contributes to the pathogenesis in patients, it is unclear how these data have been analyzed, normalized, and if/how potential batch effects have been appropriately considered. The authors should add supplementary data to clearly demonstrate to the reader that the analyses have been adequately normalized and potential "batch effects" have been accounted for.
- The authors demonstrate in Fig. 1c that increased Vav2 expression correlates with shorter survival rates in oSCC patients, underscoring the medical relevance of this study. However, a simple comparison of Vav2 expression and survival in the HNSCC TCGA data set on cBioportal does not support this conclusion. The authors must explain this discrepancy and if possible, use independent patient data sets to substantiate their important findings and their translational potential. The authors mention that anatomical location is an important contributor (data not shown) but it seems to this reviewer that these data should be shown and more clearly explained. Why is Vav2 function important in one but not another anatomical location? Is Vav2 expression correlated with HPV status of the patient tumors?
- Why has Vav2 expression in patient specimens been assessed by immune-precipitation and western blot analyses, rather than western blotting alone? It seems difficult to normalize data between different experiments with this technique. Furthermore, how much variation in the tumor epithelial fraction has there been between individual specimens? Did the expression data on the authors patient specimens correlate with patient survival?
- On page 17: The authors state: As control, we used the appropriate parental cells transduced with empty lentiviral vector. What does appropriate parental cells mean exactly?
- On page 17: the authors state: When orthotopically transplanted in the tongue of partially immunocompromised mice, (which strain was used).

COMMENTS TO REFEREES
MANUSCRIPT NCOMMS-19-07108-T

GENERAL COMMENT ON THE REFEREES' DECISION:

Despite the negative final opinion on our NCOMMS-19-07108-T submission, we would like to thank the Referees for their criticisms, comments, and advice. Certainly, we do believe that the views of those Referees are totally sound. And, retrospectively, we acknowledge that we could have done a better job at presenting in a more clear and understandable manner the data presented. The data overload associated with that submission did not help either to explain in detail many of the aspects and uncertainties that were pointed out by the Referees.

However, after analyzing the concerns raised by the Referees, we believe that the key concerns pointed out by them could be addressable. Furthermore, we still consider that the take-home message of our work is interesting for the readership of *Nat Commun*. Therefore, we have decided to ask the Referees to reconsider the possible reevaluation of a fully reformatted work in which we have addressed the key problems associated with the original NCOMMS-19-07108-T submission. In particular, we have made an effort to:

- (a)** Explain in a much better way the rationale used to select the datasets employed in our in silico analyses.
- (b)** Simplify the experimental data (for example, eliminating the Vav2^{L332A}-related data and some of the in vivo carcinogenesis-based experiments) to present in a much clearer way the main message of the work (and to provide more room to explain the rest of experiments made).
- (c)** Incorporate new data in the **new Fig. 1**.
- (d)** Modify the text (Abstract, Results, Discussion), figures, and **Table S1** (info about datasets).

We hope that, with these changes, some of the problems associated with the first negative decision can be solved.

Below, we address all the comments made by the three Referees.

REVIEWER #1:

General Comment #1. *The study by Bustelo et al., is based on the observation that VAV2, a guanine nucleotide exchange factor for Rho GTPases, is frequently upregulated in cutaneous and head and neck cancer squamous cell carcinoma (SCCs). The authors propose that this overexpression is causative in terms of expansion of epithelial stem cells and subsequent carcinogenesis, based on the definition of VAV2-induced gene programs using cells and transgenic mice expressing VAV2 mutants. They also take advantage of a new “pharmacomimetic mice” that attempts to explore the impact of VAV2 inhibition within specific thresholds on tumorigenesis. The study is highly innovative.*

However, it may over rely (or over interpret) bioinformatics analyses performed on distinct subsets of SCCs, without defining the biological and clinical settings of each database used, and choosing/picking those that fit their statistical threshold without a specified rational. For

example, most correlations are performed in oral cancer datasets that may have extensive influence by their HPV status, and cell studies on oral cancer cells without much in terms of background info and compared to immortal skin cells.

Most animal model studies are instead performed in skin cancer models.

In addition, the authors appear to make multiple statements on causality based on correlative observations, and they do not consider that VAV2 tyrosine phosphorylation rather than expression levels may reflect better VAV2 activity, unless mutationally activated.

Authors' response: We thank the Referee for considering our work as “highly innovative”. Regarding his/her criticisms, we fully agree with them. They were, in fact, quite useful to reformat in a more coherent and informative manner the new version of the manuscript.

The issues regarding the datasets used, HPV status, etc. will be addressed in the sections below to avoid redundancies (see **General Comments #2** and **#3**). We consider all of them sound and, in fact, apologize for not including this important piece of information in the first version of the manuscript. Indeed, we do agree that all the missing information that has been pointed out by the Referee is critical to understand the experimental approaches used and, perhaps more importantly, to assess that the conclusions obtained are not associated with analytical caveats.

Issues regarding the type of animal models used will be also discussed in subsequent sections (see our answer to **General Comment #3** below, page 6).

Finally, we would like to respectfully indicate that we do not concur with the latter general statement made by this Referee (that we *seem to “make multiple statements on causality based on correlative observations and they do not consider that VAV2 tyrosine phosphorylation rather than expression levels may reflect better VAV2 activity, unless mutationally activated”*). Of course, it is clear that all the data obtained using in silico analyses only unveils potential clues and serves to make just functional correlations. However, we believe that we have made a significant effort at corroborating the key in silico observations using independent, wet-lab approaches. For example, we believe that our data obtained using the gain-of-function mouse model do support the idea that the upregulation of Vav2 signaling promotes a regenerative proliferative state in several parts of the skin as well as in specific head and neck areas such as the tongue and the oral epithelium (**old** and **new Fig. 2a-c**). These data were subsequently corroborated using 3D organotypic experiments with both mouse and human keratinocytes (making emphasis on using primary cells rather than transformed cell lines). Perhaps more importantly, the subsequent loss-of-function experiments using genetic and pharmacological approaches have given the “expected” opposite results (molecular and pathobiological) in primary keratinocytes and, more importantly, in hnSCC patient-derived cells. Collectively, we consider that the main mechanistic model described in our manuscript is solid from an experimental point of view.

Of course, we are fully aware that the action of Vav2^{WT} in this system must be likely dependent both on its expression and its phosphorylation state. The Vav2^{Onc} mouse model was only used as a genetically-clean experimental tool to test the effect of upregulated Vav2 signaling in vivo.

By no means we wanted to imply that the VAV2 mutations were important for human hnSCC tumorigenesis. In fact, we underscored in the first version of the manuscript (and in the new, reformatted one), that the frequency of VAV2 mutations found in this type of tumors is very low (**old Fig. S1b**, now **new Fig. S1d**). Our data does support, however, the concept that the signaling of Vav2^{WT} is important for both cutaneous (data from Vav2^{L332A}, **old Fig. S7i,j**) and hnSCC (data from knockdown VAV2 patient-derived cells; **old Fig. 6**, **new Fig. 7**). And, certainly, given our extensive previous work on these proteins, we infer that this action has to be mediated in a tyrosine phosphorylation-dependent manner in the case of the WT protein.

We would also like to emphasize that, despite the obvious differences that have to exist between the WT and the oncogenic versions of Vav2, the results obtained with the Vav2^{Onc} mice probably reflect to some extent the role of the endogenous WT protein in the system. This is clearly exemplified by:

(a) The mirror-image phenotypes obtained with Vav2 KO and the Vav2^{Onc} mice. For example, we have found that a large percentage of the Vav2^{Onc}-regulated transcriptome in the skin shows an inverse behavior to that observed in the case of the TPA-stimulated skin of Vav2 KO mice (**old and new Figs. 4 and S4**).

(b) The opposite phenotypes found in the case of VAV2^{WT} knockdown and Vav2^{Onc}-expressing cells in terms of both proliferation and differentiation state (even when considering cells from different sources such as the skin and oral epithelium).

(c) Our in silico data indicating that the levels of expression of the Vav2^{Onc}-regulated gene signature in hnSCC tumor samples shows high correlation with the overall levels of the VAV2 mRNA found in them (see **new Fig. 8c**).

General Comment #2. *The datasets used for the different analyses are not defined at all. Only their corresponding GEO number is included, without any background regarding their corresponding publication and patient stratification. This becomes critical, as the authors appear to pick the ones aligned with their interest and expected outcome without a defined criteria, often relying on the results from one dataset to support one conclusion and other datasets for others. Each of the diseases is different, with different etiology and mutational landscape. For example, it is unclear what is the distinction between oral SCC (oSCC) and hnSCC, and unclear why the authors pick one or the other as the best one to support their conclusion. In principle, one of them may reflect viral infection (HPV), which is clearly distinct anatomically and genetically from tobacco-related cancers, with different response to treatment and prognosis. Similar to skin SCC (sSCC), whose causative role and driver oncogenes are clearly different from the others. In this regard, the use of “head-and-neck” SCC (best known simply as head and neck SCC or head and neck cancer) seems odd. The frequent statement that VAV2 is involved in “poor patient prognosis” may not be correct, as a cause effect relationship has not been established. Perhaps correlation or association would be a preferred way to refer to this potential functional relationship.*

Authors’ response: This point of concern was also raised by the other two Referees as well. **We entirely agree that having that information is really important to assess the relevance**

of our in silico data. Such missing information was available at the time of the submission of the first version of the manuscript. However, due to the space constraints imposed by the large amount of data included in the NCOMMS-19-07108-T version, we mistakenly chose to eliminate that information to shorten the main text. Following the recommendations of the three Referees, we now present a detailed description of the datasets used in the new version of the manuscript. This new information includes:

(a) The rationale used to select the datasets (see pages 6-7 of the new version).

(b) A detailed description of the datasets in the main text (pages 5-7 of the new version), the figures, and in the **new Table S1** (which describes in detail the main features of each dataset used).

(c) The references, when available, of the publications that have reported the datasets used in our study (see new **Methods** section, pages 28-29 and 37 of the new version).

Although this information can now be found in the text, we include here the rationale followed to choose the datasets used in our in silico analyses:

(a) For the human expression studies (page 6 of new manuscript version), *“we interrogated four independent gene expression datasets from either cSCC (GEO GSE13355 and GSE45216) or hnSCC (GEO GSE30784 and TCGA; for further information, see both **Table S1** and **Methods**) patients. In the case of hnSCC, the TCGA collection harbored samples with (21% of cases) and without information on HPV status. The GEO GSE30784 lacks information on HPV status, although the percentage of HPV⁺ samples has been estimated to be in the 75% range¹⁸ (**Table S1**). These datasets were chosen because: (i) They contained expression data from both healthy and tumor samples. (ii) They included a minimum of 100 samples (to facilitate the generation of statistically robust data). (iii) They included samples with clinical information (e.g., dysplasia, stage of tumor progression, etc.)”*.

Importantly, all these datasets were also used for: (i) The expression studies conducted with *VAV1* (**new Fig. S1a**), *ACTL6A* (**new Fig. S1f**), *SARDH* and *BRD3* (**new Fig. S1g**) mRNAs. (ii) The enrichment analyses of c-MYC- and YAP/TAZ-regulated gene signatures according to *VAV2* mRNA levels (**new Fig. 5e,f**).

(b) For the mouse expression studies, we used the GSE21264 gene expression microarray dataset (generated at Balmain’s lab, PMID: 21244661). This expression dataset is, to our knowledge, the best one available in terms of: (i) Number of samples contained ($n = 273$). (ii) The presence of data from healthy tissues and progressive cSCC development steps (papilloma, cSCC).

(c) For the survival studies with the *VAV2* mRNA, we originally used the GSE41613 (old **Fig. 1c**, which gave a prognostic value for the *VAV2* mRNA) and the unstratified TCGA (which did not give any significant stratification value) hnSCC datasets. The reason for choosing the GSE41613 was the fact that it contained a large number of samples ($n = 97$). This dataset was

composed of HPV⁻ cases, although this information was not included in the first version of our manuscript. In the new version, we have made the following changes:

(c.1) In addition to the GSE41613 (now, **Fig. 1b**, left panel), we now have included two additional datasets: **(i)** the TCGA hnSCC, this time stratified according to HPV status following the recommendations of the Referees. As it will be seen, the VAV2 mRNA levels also display prognostic value for the HPV⁻ cases present in this dataset (see new **Fig. 1b**, right panel). **(ii)** A third expression hnSCC dataset (Ref. GSE42743, $n = 103$ HPV⁻ cases), in which the VAV2 mRNA levels also exhibit stratification power in terms of long-term patient survival (see new **Fig. S1c**).

(c.2) We included in the text the rationale used to select the foregoing datasets. As indicated in the new page 7 of the new manuscript version: *“These datasets were chosen because: (i) They contained information of at least 90 independent clinical cases. (ii) They have information on long-term survival, HPV status, and other clinical criteria of patients”*.

(c.3) We included detailed information about each of those datasets both in the **Methods** section and in the reformatted **Table S1**.

Taken together, the use of these new datasets indicates that the levels of the VAV2 mRNA correlate with the poor prognosis of HPV⁻ hnSCC patients (new Fig. 1b and S1c).

(d) For the survival studies with the Vav2^{Onc}-regulated gene signatures (old **Fig. 4e-h**), we originally used the GSE41613 and the non-stratified TCGA hnSCC dataset for the reasons already explained in **Point c** above. Now, we have extended these studies to three datasets described in **Point c**.

As above, the new data confirm that the Vav2^{Onc}-regulated gene signatures have prognostic value for HPV⁻ hnSCC patients (see new Fig. 8).

We would also like to add that the selection of different datasets for the expression and survival analyses was exclusively determined by the presence of healthy samples (in the case of expression studies) and of clinical data on patient survival (in the case of survival studies). There was not cherry-picking at all in these type of analyses. Of course, we are willing to consider other datasets that the Referees could suggest and that have escaped our attention in the current studies.

In the same context, we would like to underscore that we have also included a lot of controls to check the relevance of our VAV2 data in comparison to genes that play key roles in this tumorigenic process (*EGFR* and, in the new version of the manuscript, *ACTL6A*). For example, we have found a similar pattern between VAV2 and *EGFR* in HPV⁻ hnSCC (see **new Figs. 1b and S1e**). By contrast, *ACTL6A* shows some prognostic value both in HPV⁻ and non-stratified hnSCC patients (see **new Figs. S1e and f**). We have also included negative controls (e.g., the genes located up- and downstream of the VAV2 locus). These negative controls show no expression alterations and no prognostic value in the same test done with the VAV2 mRNA (**new Figs. S1g**).

Regarding other points raised by this Referee, we would like to add that:

(a) We have substituted and unified the terms “oral” and “head-and-neck” by the one suggested by the Referee (head and neck).

(b) We have eliminated any reference to VAV2 being “involved” in poor prognosis. In the new version of the manuscript, we always indicate that VAV2 “correlates” or “associates” with poor prognosis. We have also removed the reference to “poor prognosis” from the new title of the manuscript.

General Comment #3. *The authors may need to make a concerted effort on defining their clinical groups and corresponding datasets, and defining the criteria used to reach their conclusions for all SCCs or rational for why specific subsets are chosen (or their conclusions more valid) with respect of the other ones.*

This perceived problem spills into the experimental approaches as most of the carcinogenesis models used (in which they monitor SCC formation as compared to benign hyperplasia) rely on chemical carcinogenesis models that only apply to sSCC, hence one would expect that the primary analysis would involve human cSCC and then exploring for commonalities with other SCCs. For example, on page 6, the authors write: “Further highlighting the possible involvement of VAV2 in SCC, we found that the abundance of its transcripts correlated with shorter survival rates of oSCC patients (Fig. 1c).” Why only on oSCC? Also, “This stratification power is even better from a statistical point of view than that shown by the abundance of the mRNA for EGFR, a tyrosine kinase receptor frequently amplified in hnSCC (Figs. 1c and S1b). This stratification power is lost when using hnSCC patients that had not been separated according to the anatomical location of tumors (data not shown).” It seems that the authors are cherry picking information from these two groups of SCCs, without a strong rationale or statistical analysis of differences, including a statement of better prognostic value with respect to EGFR, without the corresponding statistical support.

In addition, the dataset used for analysis of VAV2 expression in SCC (including oSCC), and correlation with survival in oSCC are different?

Authors’ response: Agree. We believe that all the issues regarding the datasets and survival analyses have been already discussed above (see **General Comment #2**, page 3). Following such discussion, we have unified the description of these tumors as hnSCCs.

Regarding the issue of the animal models used, we would like to point out the following:

(a) The phenotypic analysis of the Vav2^{Onc} does indicate that the increased activity of Vav2 is associated with the development of epithelia hyperplasia both in the skin (back skin, ear skin; **Fig. 2a-c**) and head and neck (palate, tongue; **Fig. 2a,b**) areas. Therefore, this is consistent with the latter analysis of both skin keratinocytes, hnSCC patient-derived cells and the SCC-25 hnSCC cancer cell line.

(b) We have simplified a lot the carcinogenesis experiments included in the new version of the manuscript. Specifically, we have made the following changes:

(b.1) We have kept the DMBA single dose-based treatment of new-born animals to convey that most important take-home message of these experiments: that the upregulated VAV2 signaling favors the emergence of tumors when combined with additional genetic lesions (**new Fig. 2d-g**). Certainly, we could also include here a model for hnSCC (e.g., treatments of control and Vav2^{onc} with the carcinogen 4-NQO to trigger oral cancer). However, we believe that the DMBA data illustrates well the point we want to convey.

(b.2) We have removed the standard DMBA+TPA models used in our previous manuscript. In our opinion, the DBMA-model using newborn mice offers a very clean readout to illustrate the take-home message of our work (both in terms of simplicity of the type of carcinogenic insult and the impact that Vav2^{onc} signaling has in tumor kinetics and final cSCC burden obtained).

(b.3) Following the indications made by Referee #2, we have eliminated all the carcinogenesis experiments involving the use of the catalytically hypomorphic Vav2^{L332A} mice that were included in our first submission (**old Figs. 7a-c and S7i-j**). This modification has also allowed us to focus our attention on the main message of original manuscript: the impact of Vav2 signaling in both naïve and cancer cells.

(c) We believe that the interconnections between cSCC and hnSCC made in the manuscript are fully coherent and logical given that: **(i)** The phenotype observed in the skin and oral epithelial areas of Vav2^{onc} mice (see **Point a** above). **(ii)** The similar deregulation of the VAV2 mRNA in cSCC and hnSCC cases (**new Fig. 1a**). **(iii)** The conservation of Vav2^{onc}-regulated skin gene signatures in human hnSCCs. **(iv)** The prognostic value of the VAV2 mRNA (**new Figs. 1b and S1c**) and Vav2^{onc}-regulated gene signatures (obtained from skin samples) in HPV⁻ hnSCC cases (**new Figs. 1b, S1c and 8**). This is specific, according to principal component analyses (**new Fig. 8d, old Fig. 4g**). **(v)** The knockdown of VAV2 in the hnSCC patient-derived cells elicits a phenotype (**new Fig. 7, old Fig. 6**) opposite to that found with Vav2^{onc} in skin keratinocytes (**new Figs. 2, 3 and 5**).

General Comment #4. *Tumors are heterogeneous, with abundant immune cell and stromal contamination. This may influence the mRNA and protein expression throughout the study. Especially when addressing the relationship between VAV2 expression and “stage” and “tumor size”. Is VAV2 perceived higher mRNA and protein expression due to immune/inflammatory infiltration?*

Authors' response: Agree. We hope that the new information generated in the reformatted version of the manuscript can dispel these concerns. This information includes:

(a) New in silico-based expression data, demonstrating that the hematopoietic-specific VAV1 mRNA is not enriched in any of the samples derived from cSCC and HPV⁻ hnSCC (see **new Fig. S1a**). By contrast, we did find the VAV1 mRNA slightly elevated in psoriatic cases (see **new Fig. S1a**, left panel). This information has also been included in the new main text (page 7): “We

did not observe any statistically significant elevation in VAV1, a mRNA encoding a VAV family member that shows a hematopoietic-specific pattern of expression^{19,20}, in any of those pre- and neoplastic stages (Fig. S1a). By contrast, the VAV1 mRNA does show a small, although statistically significant enrichment in the case of samples from psoriatic patients (Fig. S1a, left panel). These data suggest that the changes seen in VAV2 mRNA levels are probably intrinsic to tumor cells rather than an indirect consequence of the presence of varying amounts of infiltrating hematopoietic cells in the interrogated samples”.

(b) New immunohistochemical data using a hnSCC tissue microarray showing that the VAV2 immunoreactivity is mostly detected in cancer cells but not in stromal components or hematopoietic infiltrates (see **new Fig. 1e**). We have included this information in the main text as well (page 9): *“This expression is concentrated in the cancer cells rather than in the surrounding stromal components (Fig. 1e), further demonstrating that the upregulation of VAV2 is a cancer cell-intrinsic phenomenon”.*

Specific Point #1. *On page 7, the authors comment that oSCC have a common hypopharyngeal origin, an anatomical area usually associated with very large cancerization fields. Is this HPV related? If so, what is the relevance of the comparison with non-HPV related lesions and experimental systems?*

Authors’ response: Sorry, this is a misunderstanding. We were referring in that case to the few healthy samples that showed moderate-to-high levels of expression in our immunoprecipitation experiments, not to all the samples used in these analyses.

Regarding the use of “healthy” samples, it is worth noting that our analyses used lysates obtained from healthy tissues that were in the vicinity of the tumors extracted from the same patients (not from different patients or from different, far-away-located anatomical locations). Due to this, they were assigned the same identification numbers that the match-pair tumor samples (see **Fig. 1c**). This information is given both in the main text and the **Methods** section. In the former case, we state (page 9): *“We next performed immunoprecipitation analyses to monitor the levels of VAV2 in cellular extracts from 83 HPV⁻ hnSCC patient samples. As control, we included lysates from healthy tissues that were in the vicinity of the tumors extracted from the same patients”.* In the latter case (page 29), we indicate: *“Healthy mucosa, when included in the experiments, were obtained from the same hnSCC patients (due to this, they were assigned the same identification number shown in Fig. 1c)”.*

Specific Point #2. *Example, the authors compare all oSSC cell lines to KerCT, an immortalized, nontransformed primary human keratinocyte cell line derived from the skin.*

Authors’ response: Agree. These data have been eliminated in the new version of the manuscript.

Specific Point #3. *The origin and description (and mutational status) of the cell lines used is also not described, the authors used several cells, but how they relate to the oSCC that they try*

to model (including HPV status) is not described, and why they compare the information to skin normal cells need to be addressed or corrected experimentally.

Authors' response: They are HPV⁻. SCC-25 is a cancer cell line widely used in hnSCC. The VdH01 and VdH15 cells are patient-derived and generated at the Vall d'Hebron Hospital. Both of them derived from hnSCC originated in the tongue. We have made modifications in the main text to specify the nature of these cells (page 18): *“To this end, we knocked down the endogenous VAV2 transcripts using several independent shRNAs in two independent clones of HPV⁻ hnSCC patient-derived cells (VdH15, VdH01) (Fig. S6a). In addition, we utilized a similar approach in the case of the widely-used SCC-25 hnSCC cell line (Fig. S6a). All these cells were obtained from tongue-derived hnSCCs and have been described elsewhere^{25,26}”*.

Specific Point #4. *In figs 2A-B, is proliferation observed in the basal layers or suprabasal?*

Authors' response: According to the data shown in **Fig. 5a** (left panel), the proliferation seems to occur both in basal and suprabasal cells (using cyclin D1 as marker). The same seems to take place when using 3D organotypic cultures from Vav2^{Onc} mice and Vav2^{Onc}-expressing keratinocytes, respectively. In this case, we have used cyclin D1 (**Fig. 5a**, rest of panels) and Ki67 (**Fig. 3a,c**) as markers. The same result is observed in the case of Rac1+RhoA-expressing human keratinocytes (**Fig. 3h**).

Specific Point #5. *The rationale for the selection of the particular Rho GTPase mutants is not specified.*

Authors' response: Totally agree. As indicated in the new version of the text (page 11): *“As positive controls, we used human keratinocyte derivatives stably expressing mutant versions of Rac1 (F28L mutant), RhoA (F30L mutant) and Cdc42 (F28L mutant) (Fig. S3c). These mutant proteins show constitutive activity due to high intrinsic rates of GDP/GTP exchange in the absence of upstream stimulation²⁴”*. Sorry for not including this information. The nature of these mutants is well-known by scientist working in the Rho and Ras field, but it is clear that they have to be introduced to non-specialists in this area.

Specific Point #6. *At the core of the uncertainty of this reviewer regarding the suitability of this study is the observation that the overall levels of the raw Vav2^{Onc}-dependent transcriptome correlate with the poor prognosis of oSCC patients (Fig. 4d), but not sSCC, aligned with the comments (above) regarding concerns about the appropriate selection of datasets to compare with.*

Authors' response: Thanks again for pinpointing this problem. As described in the new version of the manuscript, the prognostic value of the levels of the VAV2 mRNA (new **Figs. 1b** and **Fig. S1c**) and the Vav2^{Onc}-regulated gene signatures (now, **new Fig. 8**) is maintained in all HPV⁻ hnSCC cohorts analyzed. The lack of stratification previously found in the TCGA hnSCC dataset was due to the fact that we did not considered the HPV status of the samples used. In

fact, in our new analyses, we still cannot find correlations with survival when we use the TCGA data without considering HPV status (**new Fig. S1e**). In this context, it is worth noting that the proportion of TCGA hnSCC cases with information on HPV status is relatively small (21% of total cases; this information is given in the main text of the new version of the manuscript, page 6).

To avoid the problem raised by the Referee, we now specify in all the appropriate figures the hnSCC datasets (stratified or non-stratified according to HPV status) that were used to establish the prognostic value of the levels of the VAV2 mRNA (**Fig. 1** and **S1**) and the Vav2^{Onc}-regulated signatures (**Fig. 8**). We have also done the same in the case of other in silico analyses (e.g., see **Specific Point 7c** below).

Regarding the issue of the lack of prognostic value of the Vav2^{Onc}-driven gene signature in skin cancer (sSCC) patients pointed out by the Referee, we have not included this information in the original manuscript because it is not feasible to obtain enough numbers of sSCC patients to carry out long-term survival analyses given their usually good survival rates (>95%). Therefore, the omission of that information was not due to the fact that the Vav2^{Onc}-regulated gene signature did not give a positive score in those analyses. We have included this clarification in the new version of the manuscript (page 7): *“We next investigated whether the abundance of VAV2 transcripts correlated with disease outcome. We focused these analyses on hnSCC, given that no data are available on cSCC due to the typically high survival rates exhibited by these patients²¹”*.

Specific Point #7. *The analysis showing that c-Fos (an AP1 family member), c-Myc, Cyclin D1, and YAP are upregulated in the epidermis of Vav2^{Onc/Onc} mice is of interest, but is this also observed in all proliferating (normal) cells? If so, how do they assess direct cause-effect relationship? It appears that this may be a consequence of cell proliferation. Same for experiments using cSCCs collected from the DMBA-treated Vav2^{Onc/Onc} mice.*

Authors' response: Disagree. Several experiments rule out this possibility, although we agree that we have not explained them well in the former version of the manuscript. We describe them below:

(a) As explained now in the new text version of the manuscript (page 15): *“The immunoreactive signals obtained with each of those antibodies always display much higher intensities in the Vav2^{Onc} samples than in controls, indicating that they are not merely the consequence of the expansion of proliferative cell layers that already contain very high levels of activation of these transcriptional factors. Consistent with this, we also observed that the immunoreactivity of c-Fos, c-Myc and YAP is detected in upper cell layers of the organotypic cultures that are negative for the proliferative markers Cyclin D1 (**Fig. 5a**) and Ki67 (not shown)”*.

(b) We have shown using short-term luciferase assays in two-dimensional keratinocyte cultures that the transient expression of Vav2^{Onc} leads to the activation of promoters under the regulation of the transcriptional factors AP1, c-Myc, E2F and TEAD (**Fig. 5c,d**). As stated in the

new version of the manuscript (see page 16), under these conditions there is no obvious proliferative advantage of the transiently-transfected Vav2^{Onc} in the time-frame analyzed.

(c) In the case of c-Myc and TEAD, we have also detected direct correlations between the expression levels of the VAV2 mRNA and those of both c-MYC- and YAP/TAZ-regulated gene signatures (**Fig. 5e,f**) both in cSCC and hnSCC. Furthermore, we show that such correlation is primarily observed in the latter case in HPV⁻ tumors (consistent with the pattern of expression and prognostic value of VAV2 shown in **Fig. 1**). We now include detailed information on the datasets used in these experiments in the new version of the manuscript.

Specific Point #8. *The Vav2^{L332A/L332A} mice showed reduced tumorigenesis when subjected to the standard DMBA skin carcinogenesis method. However, VAV2 is affected in all tissues/cells, and it is unclear whether the effect is epithelial cell intrinsic or due to changes in immune infiltration or function, and inflammatory processes that characterize this tumor model (which in any case is compared to oSCC that is not what is being modeled).*

Authors' response: Agree. The Referee is totally right in his/her remark. However, the intention of using these mice was to recapitulate the action of the systemic administration of a putative drug targeting the catalytic activity of VAV2. As in the case of our mice, this type of drug administration is also expected to elicit effects both in cancer and normal cells. Furthermore, with this approach, we could also distinguish, for the first time in the field, whether there were inhibitory thresholds that could impair tumorigenesis while preserving the function of normal cells (e.g., thus avoiding the hypertension and glaucoma that develops in the absence of Vav2). In our opinion, the use of these animals was quite instrumental to demonstrate that: **(i)** We do not need to inhibit 100% the activity of Vav2 to obtain therapeutic effects in cancer cells. **(ii)** We can obtain in some cases therapeutic benefits without collateral, negative effects in other tissues depending on the level of catalytic inhibition of Vav2 achieved. To our knowledge, this was the first case in which these two observations have been made in the field.

In any case, and following the advice of Referee # 2 (**General Comment**, page 13), we have decided to remove this part of our work from the new version of the manuscript (he/she felt that this part of the work was quite unconnected to the rest of the data presented).

Regardless of the issue of the cell type responsible for the anti-tumorigenic effects found in vivo with these mice, we would like to point out that our 3D organotypic experiments fully support that the Vav2 catalysis-regulated pathways do contribute to the tumorigenesis of hnSCC cells. This is demonstrated, for example, by the detection of VAV2 knockdown-like effects upon the administration of Rac1 inhibitors to hnSCC patient-derived cells (**Fig. S7a-c**). The Vav2^{Onc}-induced hyperplasia of keratinocytes is also abolished upon the administration of Pak (**Fig. S3e,f**) and Rock (**Fig. S3e,f**) inhibitors.

Specific Point #9. *In Fig 1a, what is the dataset used to define oral dysplasia and criteria used?*

Authors' response: The data are derived from the dataset GSE30784 (this info is now included in both **Fig. 1a** and **Table S1**). The classification of the samples was made by the authors reporting that dataset (PMID 18669583), so we do not know the specific criteria used. We think that were the standard ones used by pathologists.

Specific Point #10. *In Fig 1c, what is the HPV status of the different VAV2 expression subgroups? It is possible that HPV⁺ patients, which have better prognosis, rather than VAV2 expression, may correlate with prognosis. What is the survival curve for sSCC patients, which is what they are modelling experimentally?*

Authors' response: Agree. Thank you for raising this point. We have remade the analyses of **Fig. 1a** to address this point. As seen in the new data, VAV2 is preferentially upregulated in HPV⁻ patients (see **Fig. 1a**, third panel from left). We have changed the main text accordingly (pages 6-7): *“Using this approach, we found that the VAV2 mRNA is consistently upregulated in human cSCC (**Fig. 1a**, left panel) and hnSCC (**Fig. 1a**, second and third panels from left) when compared to healthy tissue samples. In the case of hnSCCs, higher expression is detected in HPV⁻ than in HPV⁺ tumors (**Fig. 1a**, third panel from left)”*.

It is worth noting, however, that the survival analyses that have given positive scores in terms of prognostic value of the VAV2 mRNA and the Vav2^{Onc}-regulated signature have been performed exclusively with HPV⁻ samples (**new Figs. 1b, S1c and 8f**). Therefore, the prognostic value found for the VAV2 mRNA in those cases cannot be merely attributed to changes in expression of VAV2 transcripts between HPV⁻ and HPV⁺ tumors.

Regarding the issue of the lack of prognostic value of the VAV2 mRNA and the Vav2^{Onc}-driven gene signature in skin cancer (sSCC) patients, we have not included this information in the first version of the manuscript because it is not feasible to obtain enough numbers of sSCC patients to carry out long-term survival analyses (this is due to their good (>95%) survival rates). Therefore, the omission of that information was not due to the fact that the VAV2 mRNA or the Vav2^{Onc}-regulated gene signature did not give positive scores in those analyses. We have included this clarification in the new version of the manuscript (page 7): *“We next investigated whether the abundance of VAV2 transcripts correlated with disease outcome. We focused these analyses on hnSCC, given that no data are available on cSCC due to the typically high survival rates exhibited by these patients²¹”*.

Specific Point #11. *In Fig 1h, what is the HPV status of these cells, and why were they selected?*

Authors' response: See answer to this question in our comments to this Referee's **Specific Point #3** (page 9). In any case, these cells have been eliminated from **Fig. 1h**. They are used, as indicated above, in the **new Fig. 7 (old Fig. 6)**.

REVIEWER #2:

General Comment. *Lorenzo-Martin et al. provide an overwhelming amount of data on Vav2 and squamous cancer. The experiments are carried out well. Generally the paper is not very well focused and should in my opinion be focused more on the key findings. The key findings to me are Figs, 1, 2, 3, and 5. Fig 4 is too detailed and in fact is only relevant for the identification of the potential transcription factors involved. Fig 6 is less convincing by the use of shRNAs but do support the picture while Fig 7 is really out of scope. The key question for me in this MS is the role of Vav2, Rac1 and perhaps RhoA in squamous cell biology and their role in dedifferentiation and cancer proneness.*

The authors put Yes Myc Yap and E2F and AP1 forward as players, but how they precisely relate to increased Vav2-Rac1 signaling remains elusive.

Convincing is the hyperplasia in squamous cells, and the authors build a strong case on that phenotype. Also convincing is the proneness to cancer, but using DBMA is not the most elegant way and also the phenotype of the tumors is somewhat odd. Crossing with p53 and/or p16 KO mouse models and obtaining mucosal SCCs as phenotype, would have been stronger. I miss convincing direct evidence of Vav2-Rac1 signaling and the phenotype. The presented evidence is very indirect.

Authors' response: Thank you a lot for your comments. Following them, we have entirely reformatted the manuscript in this new version. As indicated, we have removed all the information that was contained in **old Figs. S7 and 8** (Vav2^{L332A}-related) to concentrate on the issues remarked by the Referee. In addition, we took this opportunity to make extensive changes in the text to: **(i)** Explain better some experiments that had been described in the former version in a rather cursory manner. **(ii)** Reorganize some parts of the text (e.g., the data on the prognostic value of the Vav2^{Onc}-regulated gene signatures have now been transferred to the final section of the **Results** section). **(iii)** Make extensive modifications in the **Abstract** and **Discussion** section. **(iv)** Redesign some of the figures.

Regarding the mechanistic issue, we consider that our data do support the direct signaling connection between Vav2-GTPase signaling and the activation of the downstream transcriptional factors. For example, our transient transfection experiments indicate that Vav2 activates those factors in a catalysis-dependent manner. Our gain- and loss-of-function experiments also suggest that Rac1 and RhoA cooperate in the generation of the Vav2^{Onc} phenotype. We have also provided drug-based data indicating that the downstream elements Pak and Rock are probably downstream of Vav2^{Onc} in this pathway. We could dissect these pathways more deeply, but we believe that the main point here is the unveiling of the connection of upregulated Vav2 signaling with the engagement of the proliferative and undifferentiated state in naïve cells and the requirement of such signaling for the maintenance of such malignant traits in hnSCC patient-derived cells.

Regarding the issue of the carcinogenic experiments, we would like to point out that these experiments were only carried out to test whether Vav2^{Onc} favored full-blown transformation upon the generation of additional genetic lesions. Certainly, we could complement these experiments with additional ones focused on hnSCC (the ones suggested by the Referee, 4-

NQO carcinogenesis experiments). However, in addition to the time delay associated with the crosses and background homogenization required, the added value of these data is not clear. In this case, we thought that was more appealing (and interesting) to verify the impact of the inhibition of Vav2 in the tumorigenic potential of hnSCC patient-derived cells that, eventually, recapitulate much better what is going on in this type of tumors.

Specific Point #1. *There are many typo's in the MS and the Abstract is very vague. It would help when it is corrected by a native English speaker.*

Authors' response: Sorry for this. We have modified extensively the text and, in addition, made a totally new version of the Abstract. We thank the Referee for pointing out these problems to us.

Specific Point #2. *In 3f right low panel, the magnification seems different.*

Authors' response: We have rechecked it. No, it is indeed the same magnification.

Specific Point #3. *The authors make a point on HNSCC and OSCC but their mouse phenotypes are skin SCCs and then also of sebaceous gland types. This is not very convincing, but might relate to the mouse model.*

Authors' response: Partially agree. As discussed in the case of Referee #1 (**General Comment #3**, page 6), we believe that the mouse models are useful because:

(a) The phenotypic analysis of the Vav2^{Onc} does indicate that the increased activity of Vav2 is associated with the development of epithelia hyperplasia both in the skin (back skin, ear skin; **Fig. 2a-c**) and head and neck (palate, tongue; **Fig. 2a,b**) areas. Therefore, this is consistent with the latter analysis of both skin keratinocytes, hnSCC patient-derived cells and the SCC-25 hnSCC cancer cell line.

(b) We believe that the interconnections between cSCC and hnSCC made in the manuscript are fully coherent and logical given that: **(i)** The phenotype observed in the skin and oral epithelial areas of Vav2^{Onc} mice (see **Point a** above). **(ii)** The similar deregulation of the VAV2 mRNA in cSCC and hnSCC cases (**new Fig. 1A**). **(iii)** The conservation of Vav2^{Onc}-regulated skin gene signatures in human hnSCCs. **(iv)** The prognostic value of the VAV2 mRNA (**new Figs. 1b** and **S1c**) and Vav2^{Onc}-regulated gene signatures (obtained from skin samples) in HPV⁻ hnSCC cases (**new Figs. 1b, S1c** and **8**). This is specific, according to our principal component analyses (**new Fig. 8d, old Fig. 4g**). **(v)** More importantly, because the knockdown of VAV2 in the hnSCC patient-derived cells yields the opposite phenotype (**new Fig. 7, old Fig. 6**) to that found with Vav2^{Onc} in skin keratinocytes (**new Figs. 2, 3** and **5**).

(c) Regarding the issue of the features of the tumors, the development of both epithelial and sebaceous tumors is quite common (at least in our hands), when using the single application of DMBA to the skin of new-born animals. Regardless of this, the data obtained here support the

idea that upregulated Vav2 signaling favors full-blown epithelial transformation when combined with additional genetic lesions (that have to be very minor, given that we have applied DMBA only once to the animals; **old Figs. 2d-f; new Fig. 2d-g**). Consistent with this, we found a specific increase in the percentage of cSCC formed at the expense of the numbers of papillomatous hyperplasia and papilloma (see **new Fig. 2g**). By contrast, there are no changes in the normal percentages of sebaceous tumors (**new Fig. 2g**).

Regarding the other models used in our original submission (which have been removed from the current one, see below), we do find some cases of sebaceous adenomas. It is important to note, however, that sebaceous gland anomalies are observed in the case of mice expressing Myc (e.g., PMIDs: 11369200 and 23403291). Rac1 has been also involved in the differentiation of sebaceous tumors (PMID: 25659584).

We would also like to indicate that we have significantly simplified the carcinogenesis experiments included in the new version of the manuscript. Specifically, we have made the following changes:

(a) We have kept the DMBA single dose-based treatment of new-born animals to convey that most important take-home message of these experiments: that the upregulated VAV2 signaling favors the emergence of tumors when combined with additional genetic lesions (**new Fig. 2d-g**). Certainly, we could also include here a model for hnSCC (e.g., treatments of control and Vav2^{onc} with the carcinogen 4-NQO to trigger oral cancer). However, we believe that the DMBA data illustrates well the point we want to convey.

(b) We have removed the standard DMBA+TPA models used in our previous manuscript to simplify the message conveyed to the readers. In our opinion, the DBMA-model using newborn mice offers a very clean readout (both in terms of simplicity of the type of carcinogenic insult provided and the impact that Vav2^{onc} signaling has in tumor kinetics and final cSCC burden obtained).

(c) Following the indications made by this Referee, we have eliminated all the data obtained with the catalytically hypomorphic Vav2^{L332A} mice that were included in our first submission (**old Figs. 7a-c and S7i-j**).

Specific Point #4. *There is prognostic impact in OSCC but not in HNSCC in general, which seems remarkable but neither the data are shown nor the patient characteristics. Moreover there was no multivariate analysis performed. Was Vav2 expression related to T stage, N stage, age, histology (differentiation!), HPV status in oropharyngeal cancers? Did that interfere with prognostic impact?*

Authors' response: Totally agree. We have made significant changes to eliminate these problems in the new version of the manuscript:

(a) For the issue of oSCC/hnSCC and the datasets used, please see our answer to Referee #1 **General Comment #2** (page 3).

(b) Regarding the multivariate analyses, we did find correlation between high levels of expression of the VAV2 mRNA levels and the HPV⁻ status of tumors (see **new Fig. 1a,b**). We could not find any additional correlation with other clinical factors. This information has been included now in the new version of the text (page 8): *“Further multivariable analyses indicated that the VAV2 mRNA levels do not show any statistically significant association with tumor stage, smoking status or therapeutic response (data not shown)”*.

Importantly, it is worth noting that the survival analyses that have given positive scores in terms of prognostic value of the VAV2 mRNA have been performed exclusively with HPV⁻ samples (**new Figs. 1b** and **S1c**). Therefore, the prognostic value found for the VAV2 mRNA in those cases cannot be merely attributed to changes in expression of the VAV2 transcripts between HPV⁻ and HPV⁺ tumors.

Specific Point #5. *Vav2 mutations are very rare in HNSCC (2.8% Vav2, 5% Rac1, and 2.8% RhoA, and some are inactivating). Moreover, the activating mutations may not even be functionally comparable to the Vav2^{onc} mutation. This was not studied at all and is a flaw in the MS.*

Authors' response: Agree. As discussed in one of our answers to Referee #1 (**General Comment #1**, page 1), we have used the Vav2^{onc} mouse model only as a genetically-clean experimental tool to test the effects of upregulated Vav2 signaling in the skin. By no means we wanted to imply that the VAV2 mutations were important for hnSCC tumorigenesis. In fact, we underscored in the first version of the manuscript (and in the new, reformatted one), that the frequencies of VAV2 mutations found in hnSCC are very low to be considered as convincing causative agents (**old Fig. S1b**, now **new Fig. S1d**). Given these data, we have not considered the characterization of the few VAV2 mutations found in cSCC and hnSCC in this work.

Despite the above, we believe that our data do support the concept that the upregulated signaling from the Vav2^{WT} protein is important for both cutaneous (data from Vav2^{L332A}, **old Fig. S7i,j**) and hnSCC (data from knockdown VAV2 patient-derived cells; **old Fig. 6**, **new Fig. 7**). The use of inhibitors for Rac1, Rock and Pak also support this idea in the case of hnSCC patient-derived cells (**old Fig. 6e**, **new Fig. S7**).

We would also like to emphasize that, despite the obvious differences that have to exist between the WT and the oncogenic versions of Vav2, the results obtained with the Vav2^{onc} mice probably reflect to some extent the role of the endogenous WT protein in the system. This is clearly exemplified by:

(a) The mirror-image phenotypes obtained with Vav2 KO and the Vav2^{onc} mice. For example, we have found that a large percentage of the Vav2^{onc}-regulated transcriptome in the skin shows an inverse behavior with that observed in the case of the TPA-stimulated skin of Vav2 KO mice (**old and new Figs. 4** and **S4**).

(b) The opposite phenotypes found in the case of VAV2^{WT} knockdown and Vav2^{onc}-expressing cells in terms of both proliferation and differentiation state (even when considering cells from different sources such as the skin and oral epithelium).

(c) Our in silico data indicating that the levels of expression of the Vav2^{onc}-regulated gene signature in hnSCC tumor samples shows high correlation with the overall levels of the VAV2 mRNA found in them (see **new Fig. 8c**).

Specific Point #6. *The authors suggest field cancerization in hypopharynx, but it is everywhere and should either be studied by genetic markers or this attempt to explain unwanted observations removed.*

Authors' response: Agree. We do not have any data supporting this idea. We have eliminated this part of the text from the new revised version. In any case, it is clear that these type of cases are not that frequent according to our experimental data (**old Fig. 1d,e**; now, **new Fig. 1c,d**).

Specific Point #7. *Expression levels of Vav2^{onc} may be changed by the large deletion and this is not shown. Perhaps this was demonstrated in a previous study, but should be presented here as well.*

Authors' response: Following the recommendations of this Referee, we have eliminated this part of our experiments from the new version of the manuscript. The expression of Vav2^{onc} is similar to that of the WT allele, as previously described (PMID: 25288640).

Specific Point #8. *Specificity of Vav2^{onc} for substrate GTPases may have been changed. Should Ras not have been investigated in the G-ELISA?*

Authors' response: We have checked the activity of the three main Rho GTPases and, as expected, the activity is confirmed by Rac1 and RhoA. The rationale for looking at Ras is unclear, given that Rho GEFs do not target this GTPases. It is structurally impossible. Of course, it is feasible, however, that Ras could be indirectly activated via the stimulation of autocrine loops.

REVIEWER #3:

General Comment. *Rho-GTPases are long thought to contribute to tumorigenesis. Still, how their deregulation contributes to carcinogenesis remains largely elusive. In this manuscript, Lorenzo-Martin et al. provide compelling evidence that increased Vav2 activity promotes squamous carcinogenesis in skin and oral epithelia. This manuscript is well written, the experimental data are interesting, clearly presented, and correctly interpreted. The main strength of the study are the gain and loss of function mouse models demonstrating a role for Vav2 in squamous carcinogenesis, combined with links to published patient data and potential mechanistic links that had been established by various studies in the field. Although this paper stops short in providing a direct molecular mechanism for how Vav2 activity controls a cancer promoting transcriptional program and it remains unclear whether the transcriptional changes are a direct or indirect result of aberrant Vav2 function, it provides compelling evidence that*

pharmacological Vav2 inhibition can inhibit SCC growth. In conclusion, this is an interesting and well executed study that seems suitable for publication in Nature Communications after some minor questions have been addressed.

Authors' response: Thank you for your encouraging comments.

Specific Point #1. *It is unclear where within the stratified epithelium and a SCC Vav2 is expressed and active. What is its sub-cellular localization?*

Authors' response: Agree. We now provide new information indicating that VAV2 is preferentially detected in basal precursors using both immunohistochemical (**new Figure S1h**, oral epithelium) and in silico (**new Fig. S1i**, skin keratinocytes) analyses. The text associated with these data can be found in page 9 of the new version of the manuscript: *“These immunohistochemical studies also revealed that VAV2 is detected in the basal but not in the more differentiated layers of the normal oral epithelium obtained from the same patients (Fig. S1h, left panel). However, we found that the VAV2 immunoreactivity expands to suprabasal cell layers when analyzed in samples exhibiting histological signs of dysplasia (Fig. S1h, right panel). The levels of immunoreactivity found in the foregoing cases were lower than those detected in tumors (Fig. 1e). The preferential expression of the VAV2 mRNA in keratinocyte precursors and its downmodulation in more mature derivatives is also observed in the skin when using gene expression data from organotypic cultures that recapitulate this differentiation process (Fig. S1i). By contrast, differentiation-associated transcripts show the expected upregulation as these cell precursors progressively give rise to a more mature progeny in those cultures (Fig. S1i)”*.

Specific Point #2. *How does Vav2 become over-expressed in SCCs? What regulates its expression?*

Authors' response: Good question. We do not know at this moment the cause involved. We have posited several possibilities in the **Discussion** section (spurious activation of superenhancers, mRNA translatability, feed-back mechanisms that boost the basal expression seen in cell precursors; see pages 24-25). We can also exclude other potential causes (gene amplification events). What is clear is that: **(i)** It is specific for cSCC and hnSCC (we do not find the same consistent upregulation, for example, in lung or cervix SCCs). **(ii)** It is specific for the VAV2 locus, given that the neighboring genes area not deregulated. **(iii)** It has to involve some kind of increased translation of the VAV2 mRNA, given that the fold changes observed at the protein levels are always higher than those found at the transcript level.

Specific Point #3. *The authors provide a significant amount of meta analyses of previously published data sets, providing evidence that Vav2 expression is up-regulated in human SCCs and linked to poor outcome in patients. They also compare Vav2 dependent changes in gene expression to various other, previously published data sets. These analyses are well presented and they significantly strengthen the work. However, the authors only reference the data sets in*

the experimental methods sections and they fail to cite the papers that provided the basis for their meta analyses. The authors must clearly and adequately reference the papers that generated the data sets for their meta analyses within the main text of the manuscript as it is difficult for the reader familiar with the field to understand what studies have contributed to the conclusions of the presented work.

Authors' response: Agree. Following the recommendations of the three Referees, we now present a detailed description of the datasets used. The new information provided includes:

- (a) The rationale used to select the datasets (pages 6-7 of new manuscript version).
- (b) A detailed description of them (both in the main text, pages 5-7, and in a modified **Table S1**).
- (c) The references, when available, of the publications that have reported them (see **new Methods** section of new version, pages 28-29 and 37).

We thank the Referee for pinpoint this main problem in the manuscript. This information was available but, due to the space constraints imposed by the extension of our original manuscript, we mistakenly chose not to include it to make the text shorter. We agree that having that information was really important to assess the relevance of our in silico data.

Specific Point #4. *Upon closer inspection of the previously published data sets that have been used for the meta analyses in this paper it becomes apparent that normal and psoriatic patient data were compared to SCC data generated by different labs in different papers. Although the presented approach resulted in compelling evidence for how aberrant Vav2 expression contributes to the pathogenesis in patients, it is unclear how these data have been analyzed, normalized, and if/how potential batch effects have been appropriately considered. The authors should add supplementary data to clearly demonstrate to the reader that the analyses have been adequately normalized and potential "batch effects" have been accounted for.*

Authors' response: Agree. When using different datasets (with data generated using the same analytical gene expression platform), we used the frozen robust multiarray analysis (fRMA) to avoid the batch effects referred to by the Referee. This information is given in **Methods** and further specified in the legend to **Figure 1a** where we now indicate (page 54 of new manuscript version): *"When data were generated using two datasets at the same time, we used frozen robust multiarray analyses (fRMA) to avoid batch effects as indicated in Methods"*.

Specific Point #5. *The authors demonstrate in Fig. 1c that increased Vav2 expression correlates with shorter survival rates in oSCC patients, underscoring the medical relevance of this study. However, a simple comparison of Vav2 expression and survival in the HNSCC TCGA data set on cBioPortal does not support this conclusion. The authors must explain this discrepancy and if possible, use independent patient data sets to substantiate their important findings and their translational potential. The authors mention that anatomical location is an important contributor (data not shown) but it seems to this reviewer that these data should be*

shown and more clearly explained. Why is Vav2 function important in one but not another anatomical location? Is Vav2 expression correlated with HPV status of the patient tumors?

Authors' response: Agree. The lack of stratification found in the TCGA hnSCC dataset used by us (and the cBioPortal dataset indicated by the Referee) is due to the fact that such analyses did not take into account the HPV status of the samples used. As shown in our new version of the manuscript, we do find statistically significant stratification of HPV⁻ patients in terms of both VAV2 mRNA levels (**new Fig. S1b**, right panel) and Vav2^{Onc}-regulated gene signatures (**new Fig. 8f**). In this context, it is worth noting that the proportion of TCGA hnSCC cases with information on HPV⁻ status is relatively small (21% of total cases; this information is given in the main text of the new version of the manuscript, page 6).

See further comments on this in our answers to Referee #1's **Points 6** (page 9) and **10** (page 12).

Specific Point #6. *Why has Vav2 expression in patient specimens been assessed by immunoprecipitation and western blot analyses, rather than western blotting alone? It seems difficult to normalize data between different experiments with this technique. Furthermore, how much variation in the tumor epithelial fraction has there been between individual specimens? Did the expression data on the authors patient specimens correlate with patient survival?*

Authors' response: Agree. Honestly, we have chosen this approach because I always prefer to “see” the bands rather than trusting immunohistochemical signals.

However, since the original submission, the pathologists that have collaborated with us have managed to optimize one of the available Vav2 antibodies to carry out new immunohistochemical studies in tissue microarrays. These new data are included in the **new Fig. 1** (panel e). In agreement with the previous data from our in silico and immunoprecipitation experiments, we have detected elevated levels of VAV2 immunoreactivity in patient-derived tumor samples. These data are also discussed in the main text of the new manuscript version (page 9): *“Finally, we found medium (41.8%) to high (14.7%) VAV2 immunoreactivity levels in a significant fraction of the 232 independent hnSCC samples present in an in-house generated tissue microarray (Fig. 1e). This expression is concentrated in the cancer cells rather than in the surrounding stromal components (Fig. 1e), further demonstrating that the upregulation of VAV2 is a cancer cell-intrinsic phenomenon”*.

We would like to indicate that the lysates used in the immunoprecipitations have been always normalized according to protein content prior to carrying out the experiments. The Ponceau S staining of the starting material is also presented as control in these experiments. And, whenever possible, we used “matched” samples of healthy and tumoral material from the same patients.

Specific Point #7. *On page 17: The authors state: As control, we used the appropriate parental cells transduced with empty lentiviral vector. What does appropriate parental cells mean exactly?*

Authors' response: We were referring to the corresponding control of each cell type. We have rephrased this sentence to avoid this confusion. The new text says (page 18): *“As negative control, we used VdH15, VdH01, and SCC-25 cells that were transduced with an empty lentiviral vector (referred to as “pLKO”).”*

Specific Point #8. *On page 17: the authors state: When orthotopically transplanted in the tongue of partially immunocompromised mice, (which strain was used).*

Authors' response: We have used the compound $Vav1^{-/-};Vav2^{-/-};Vav3^{-/-}$ mice that, in our hands, works similarly to some of the standard immunocompromised mice in xenograft experiments using different cancer cells, including those used in this study. This model is much more cost-saving than the standard strains that are commercially available. This information was present in the **Methods** section of the NCOMMS-19-07108-T version. However, in the new version of the manuscript, we have also included this information in the main text as well (page 19): *“When orthotopically transplanted into the tongue of partially immunocompromised $Vav1^{-/-};Vav2^{-/-};Vav3^{-/-}$ mice, we found that the hnSCC patient-derived cells bearing the VAV2 shRNA show reduced tumorigenicity (Fig. 7b,c) and metastatic potential (Fig. 7d) than controls”.*

Reviewers' comments:

Reviewer #1 (Remarks to the Author); expert in signalling Rho/GTPases:

The revised version of the study by Bustelo et al., is highly improved. The primary observation is that VAV2, a guanine nucleotide exchange factor for Rho GTPases, is frequently upregulated in cutaneous and head and neck cancer squamous cell carcinoma (SCCs). The authors propose that this overexpression is causative in terms of expansion of epithelial stem cells and subsequent carcinogenesis, based on the definition of VAV2-induced gene programs using cells and transgenic mice expressing VAV2 mutants. The authors have tamed down their prior strong causative conclusions based on bioinformatics revealing statistical correlations, and have provided much more extensive description of the datasets used and computational pipeline. They have also addressed the HPV status of the patients from which these datasets were derived, concluding now that most observations and correlations are primarily applicable to HPV- HNSCC cases. They have also extended the studies to oral and skin cancer biology. Overall, the new version is much more concise and to the point, and the additional information and results provided have made this study stronger and more conclusive.

Few areas may still need to be addressed.

There are countless references to "data not shown". As per the journal policy, include the data or refrain from including related statements.

On Page 9, the text "They also indicate that such an upregulation can forecast poor prognosis in the case of HPV- hnSCC patients," may not be accurate, only correlation was analyzed, not predictive value as biomarker.

In Figure 6, verteporfin and 10058-F4 can block YAP and MYC signaling, respectively, but many studies have questioned the initially described specificity of this pharmacological approach. In addition, the activity of these drugs was demonstrated using transfected luciferase reporters driven by these transcription factors. The authors could address this easily using siRNAs and reporting the expression of endogenous genes regulated by these transcription factors.

The authors comment that the human cells were obtained from tongue-derived hnSCCs and have been described elsewhere^{25,26}. What is their genetic background and HPV status? Given that figure 7 is the only figure in which the authors ask the role of the endogenous VAV2 in human hnSCCs, more biological context of the models selected may be necessary.

In this context, given that endogenous Vav2^{WT} in human SCCs must be likely dependent both on its expression and its phosphorylation state, the authors may need to show that this is the case in the human hnSCC cell lines, considering that they acknowledge that "the Vav2^{Onc} mouse model was only used as a genetically-clean experimental tool to test the effect of upregulated Vav2 signaling in vivo". The response from the authors to this issue is accurate, but may need to be reflected in the manuscript.

In Figure 7g, the authors may have selected a highly differentiated area for the IHC analysis in the shVAV2#3 derived tumors. Please, confirm that this is representative.

Reviewer #2 (Remarks to the Author); expert in human SCC:

Lorenzo-Martin et al revised their MS on VAV2 in SCC. It is still an avalanche of data, and to my opinion it lacks focus.

In Fig 1 the point is made that SCCs overexpress VAV2 both at the RNA and protein level, and this associates with prognosis. A real problem is the lack of information on HPV. Would be important to analyze VAV2 expression in HPV+ve and HPV-ve HNSCC when known, to exclude HPV as confounding factor. Suggestions: b remove EGFR graphs, make x-axis in increments of 12 mths and stop at 60 mths. P-values in max two digits. In c how reproducible are these data? In some patients two bands are identified. What is visualized with Ponceau S, the IgG bands?

In Fig 2 the point is made that VAV2onc has a carcinogenic effect on skin and mucosa in mice models. Evidence that overexpression of VAV2 has the same effect as the mutant, is lacking.

In Fig 3, the point is made that VAV2onc cells from mice have an increased proliferative capacity in organotypic models, confirming the in vivo data. Also Rac and Rho mutants show this effect.

In Fig 4 the point is made that VAV2onc causes expression profiles associated with dedifferentiation and increased stemness.

In Fig 5 the associated transcription factors that were identified in Fig 4 are further analyzed. Fig 5c and 5d are unclear and what precisely is shown does not become clear from the legend. The point the authors want to make with 5e and 5f is also unclear to me.

In Fig 6 the authors show with inhibitors that myc and yap/tead impact the morphological changes of hyperplasia. 6c is insufficiently explained in the legend. The role of myc is supported by ectopic expression, of yap/tead not.

In Fig 7 inhibition of VAV2 by shRNA is studied. Very confusing that Vav1^{-/-} etc immunocompromised mice were used.

In Fig 8 prognostic gene expression profiles related to the VAV2 signaling are identified. To me this is beyond the scope of this paper.

Despite a lot of experimental work: there is no evidence presented that upregulated VAV2 has the same effect as the mutant, which is the assumption to link the cancer data to the functional data. In addition it is not VAV2-specific as Rac and Rho do the same: hence it seems that GEF activity in general elicit these phenotypes. The main question remaining is how relevant this is for HNSCC in man. The experimental data look convincing, but which important question was answered? There are so many molecules that have essential functions in cells and also tumor cells. There is no data presented that VAV2 is a fantastic therapeutic target in HNSCC or that it is a key pathway in carcinogenesis.

Reviewer #3 (Remarks to the Author); expert in mouse models for SCC:

Building on mouse and human models, the manuscript by Lorenzo-Martin et al provides compelling evidence that increased VAV2 activity promotes squamous carcinogenesis in skin and oral epithelia. The authors have adequately addressed my concerns with the initial version of the manuscript by (a) adding new experimental data; and (b) clarifying some sections in the text. I believe this study is solid, interesting, well presented and suitable for publication in Nature Communications.

COMMENTS TO REFEREES
MANUSCRIPT NCOMMS-19-07108A-Z

REVIEWER #1

General Comment. *The revised version of the study by Bustelo et al., is highly improved. The primary observation is that VAV2, a guanine nucleotide exchange factor for Rho GTPases, is frequently upregulated in cutaneous and head and neck cancer squamous cell carcinoma (SCCs). The authors propose that this overexpression is causative in terms of expansion of epithelial stem cells and subsequent carcinogenesis, based on the definition of VAV2-induced gene programs using cells and transgenic mice expressing VAV2 mutants. The authors have tamed down their prior strong causative conclusions based on bioinformatics revealing statistical correlations, and have provided much more extensive description of the datasets used and computational pipeline. They have also addressed the HPV status of the patients from which these datasets were derived, concluding now that most observations and correlations are primarily applicable to HPV- HNSCC cases. They have also extended the studies to oral and skin cancer biology. Overall, the new version is much more concise and to the point, and the additional information and results provided have made this study stronger and more conclusive.*

Authors' response: Thank you for your kind comments. This improvement, certainly, has been done due to the insightful inputs from you and the rest of reviewers.

Minor issue #1. *There are countless references to “data not shown”. As per the journal policy, include the data or refrain from including related statements.*

Authors' response: Agree. Those statements have been eliminated and, when the data were kept, have been associated with specific statistical values indicated in the main text.

Minor issue #2. *On Page 9, the text “They also indicate that such an upregulation can forecast poor prognosis in the case of HPV- hnSCC patients,” may not be accurate, only correlation was analyzed, not predictive value as biomarker.*

Authors' response: Agree. Sorry for this oversight. In the new version of the text (page 9), we now say that: “They also indicate that this upregulation correlates with poor prognosis in the case of HPV⁻ hnSCC patients.”

Minor issue #3. *In Figure 6, verteporfin and 10058-F4 can block YAP and MYC signaling, respectively, but many studies have questioned the initially described specificity of this pharmacological approach. In addition, the activity of these drugs was demonstrated using transfected luciferase reporters driven by these transcription factors. The authors could address this easily using siRNAs and reporting the expression of endogenous genes regulated by these transcription factors.*

Authors' response: Agree. It is important to indicate, however, that this type of experiments have to be performed with inducible expression systems given that we are using 3D organotypic cultures. Only with this type of approach we could address the effect of the inhibition of these pathways in the context of an epithelia that is undergoing both proliferation and differentiation decisions. In addition, it is only the proper way to adequately mimic the data obtained with the chemical inhibitors, which were added to the cultures 6 days after the initiation of the 3D conditions (a time in which we already have extra layers of undifferentiated cells in these cultures).

Based on this, we have selected two inducible systems to carry out these proposed experiments. In the case of MYC, we chose a previously described 4-hydroxytamoxifen-inducible system based on the expression of a Myc peptide inhibitor (the “omomyc” dominant negative for c-MYC) fused to the ER protein (pBP-Omomer; obtained from Addgene, Catalog No. 113169). This reagent has been developed by Saucek’s group (Saucek et al, *Cancer Res* 2002; PMID: 12067996). In the case of YAP, we have selected an inducible system based on the doxycycline-dependent expression of a shRNA for *YAP1* (plus a red fluorescent protein to monitor that the inducible system works, pTetON-shYAP; obtained from Addgene, Catalog No. 115667). This vector has been developed in Massagué’s lab (Er et al, *Nat Cell Biol* 2018; PMID: 30038252). These plasmids were introduced both in the control (previously transduced with the empty pLKO lentiviral vector) and the Vav2^{Onc}-expressing human keratinocytes.

However, we failed in getting the system to work in our 3D cultures despite extensive efforts. This is because, under these conditions, both the control and the Vav2^{Onc}-expressing cells carrying the foregoing inhibitory vectors cannot form the expected 3D tissue structures. Given that this problem is also observed in the control cells, we guess that it is due to some type of technical question (e.g., some leakiness of the constructs used, long-term toxicity of either the 4-doxycycline or the 4-hydroxytamoxifen treatments). This problem persisted upon trying different doses of the chemical inducers. We can keep trying more technical variables, although this is a highly time consuming process given that each 3D culture takes a minimum of 15 days to be carried out (plus the additional time required for generating the histological sections and subsequent analyses). This is further complicated by the time shifts that have been implemented in our Center to follow up the mandatory anti-Covid-19 health policies currently active in our country.

In any case, we would like to vindicate the data obtained with the chemical inhibitors. Firstly, there is extensive evidence that the compounds chosen do act selectively on the disruption of the c-Myc/Max (10058-F4) and the YAP/TEAD (Verteporfin) interactions. Secondly, we have selected concentrations of the compounds in our experiments that did not interfere with the growth and the differentiation state of the control cells. As a result, the effects elicited by these inhibitors in our experiments are specific for the Vav2^{Onc}-expressing keratinocytes. This approach also rules out the possibility of potential off-target effects for those drugs, an idea supported by the observation of the lack of effects of those inhibitors in the epithelia formed by control keratinocytes. In this context, it is also worth noting that the concentrations used for these inhibitors in our experiments are always in the nM range, not in the μ M range typically used in other studies. Finally, we have shown that 10058-F4 and Verteporfin block the activation of c-Myc- and TEAD-dependent promoters when tested in transient luciferase reporter assays in keratinocytes, respectively (see **old Figure 6e**, now **new Figure 7c**).

Minor issue #4. *The authors comment that the human cells were obtained from tongue-derived hnSCCs and have been described elsewhere^{25,26}. What is their genetic background and HPV status? Given that figure 7 is the only figure in which the authors ask the role of the endogenous VAV2 in human hnSCCs, more biological context of the models selected may be necessary.*

Authors' response: Agree. Following the reviewer's advice, we have characterized the whole genome of the two patient-derived cells (PDCs) using next generation sequencing. The main genetic alterations found are now indicated both in the main text (pages 20-21) and in the new **Supplementary Figure S6**. These data indicate that:

1. VdH01 and VdH015 are clearly different according to the catalogue of both deleterious single nucleotide (**Supplementary Figure S6a**) and copy number (**Supplementary Figure S6a**) variations present in them. This is to some extent expected, given that they were obtained from independent patients.

2. Despite such molecular disparity, these two patient-derived cells show alterations in common loci (*APOBEC3B, AXIN1, BCOR, CASP8, CREBBP, DDX3X, EP300, FAT1, FHIT, GRIN2A, MRTFA, MYH9, NTHL1, ROBO2, TP53, TRAF7, TSC2, ZRSR29*) (**Supplementary Figure S6a-c**). They also show a gain in the long arm of chromosome 9, a frequent event in hnSCC from oral regions (the original location of the tumors from which the patient-derived cells were obtained from).

3. Confirming our previous data using PCR detection, the two PDCs are HPV⁻ according to the next generation sequencing data (**new Supplementary Figure S6a**).

4. Many of the genetic alterations found in each of those PDCs have been previously detected at high frequency rates in hnSCC cases (*CASP8, CDKN2A, EP300, FAT1, NOTCH1, MYC, PABPC1, RAD21, RECQL4, TERT, SPEN, TP53*) (**new Supplementary Figure S6a-d**).

5. The two PDCs lack gain-of-function point mutations in the *VAV2* gene (**new Supplementary Figure S6a**)

6. We found that the VdH15 PDC bears a small deletion of 1,068 bp in a region downstream of the *VAV2* locus (**new Supplementary Figure S6e**; coordinates chr9:136625194-136626262). The functional relevance of this deletion for either *VAV2* expression or hnSCC etiology is unknown. We have to indicate, however, that it does not seem to be a frequent event in hnSCC according to available TCGA cancer genome data.

All these data have been obtained in collaboration with Sonia Zumalave and José M. C. Tubío, two experts on cancer genomics, who now are part of the authorship of this work.

Minor issue #5. *In this context, given that endogenous Vav2^{WT} in human SCCs must be likely dependent both on its expression and its phosphorylation state, the authors may need to show that this is the case in the human hnSCC cell lines, considering that they acknowledge that "the Vav2^{Onc} mouse model was only used as a genetically-clean experimental tool to test the effect*

of upregulated Vav2 signaling in vivo". The response from the authors to this issue is accurate, but may need to be reflected in the manuscript.

Authors' response: Agree. To address this issue, we have reblotted the VAV2 immunoprecipitates originally presented in **old Figure 1c** using antibodies specific for general tyrosine-phosphorylated residues. The data obtained indicate that endogenous VAV2 is indeed tyrosine phosphorylated in most of the hnSCC patient samples surveyed (see **new Figure 2a** and quantitation of data obtained in **new Figure 2c**). This is probably an under-estimation of the total levels of tyrosine phosphorylation of VAV2, given that the long experimental procedures typically associated with tumor collection and storage can negatively affect the phosphorylation levels of the protein. In addition, we show that ectopically expressed Vav2^{WT} is also tyrosine phosphorylated in keratinocytes (**new Figure 8b**). Collectively, these data indicate that both endogenously and ectopically expressed Vav2^{WT} are tyrosine phosphorylated in hnSCC and primary keratinocytes, respectively. In line with these observations, it must be mentioned that a previous report has shown high levels of tyrosine phosphorylation of endogenous Vav2 in a number of established hnSCC cell lines (Patel et al, Carcinogenesis 2007; PMID: 17234718). We have slightly changed one of our original sentences in the manuscript to indicate this point to the future readers of the paper. The new sentence now says (Results section, page 6): *"In favor of the connection of VAV2 with this type of tumors, work with human hnSCC cell lines has revealed that this GEF is frequently tyrosine phosphorylated and involved in the stimulation of RAC1 downstream of the epidermal growth factor receptor (EGFR) in a significant number of human hnSCC cell lines (Ref 11)."*

Minor issue #6. *In Figure 7g, the authors may have selected a highly differentiated area for the IHC analysis in the shVAV2#3 derived tumors. Please, confirm that this is representative.*

Authors' response: Agree. This is indeed a general event in the tumor sections obtained from the orthotopic transplantation experiments. We have included a lower magnification of these images in the **new Supplementary Figure S8d** to demonstrate this issue to the future readers of this work.

REVIEWER #2

Comment #1. *Lorenzo-Martin et al revised their MS on VAV2 in SCC. It is still an avalanche of data, and to my opinion it lacks focus.*

Authors' response: We respectfully disagree. In our opinion, we believe that the data are presented in a coherent way to demonstrate the importance of the deregulation of the VAV2 pathway in the context of both keratinocyte and HPV⁻ hnSCC biology. This has been done in a stepwise manner: human tumor-derived in silico data, → protein data in human tumors, → mouse models to assess the impact of deregulated Vav2 signaling in naïve keratinocytes in vivo, → 3D organotypic cultures to demonstrate that this is a cell autonomous event, → genomics and functional studies to decipher the pathobiological mechanisms involved, → loss-of-function approaches in patient-derived cells to demonstrate the importance of endogenous

VAV2 in hnSCC tumorigenesis and, finally, → in silico work to identify Vav2-regulated gene signatures that have prognostic value in these tumors. As presented, our data also indicate that this Vav2-regulated program is common to both cutaneous and head and neck SCCs but not to SCCs present in other locations (e.g., lung). We would also like to point out that the other two referees did not find any problem in how the manuscript was organized.

In any case, we have decided to make some changes in the original structure of the manuscript in order to get the data as grouped as possible in the same figures. With these modifications, we believe that we will also avoid that the readers will have to move back and forth between the main figures and the supplementary ones as in the previous format. We described the main changes made below:

1. We now present the VAV2 mRNA (**new Figure 1**) and protein/phosphorylation (**new Figure 2**) data in two separate figures instead of a single one (**old Figure 1**). We have also moved some of the information originally present in the **old Supplementary Figure S1** (panels h and i) to the **new Figure 2** (panels e and f). As a result, the **new Supplementary Figure S1** only contains some of the panels (a to g) originally present in the **old Supplementary Figure S1** (mRNA expression and survival data).

2. In the **new Figure 5**, we have combined the information originally presented in the **old Figure 4** (panels a, b and c) and the **old Supplementary Figure S4** (panels b to e). With these changes, this figure now contains all the information associated with the functional annotation of the Vav2^{Onc}-dependent transcriptome. As a result, the new version of the **old Supplementary Figure S4** has been limited to the inclusion of only one panel (old panel a).

3. In the **new Figure 6** (corresponding to the **old Figure 5**), we now have incorporated all the information originally presented in **old Figure 4d,e** (as panels **5a** and **5b**). With this change, we have consolidated all the data regarding the identification and validation of the transcriptional factors that participate in the orchestration of the Vav2^{Onc} transcriptome in a single figure.

4. The information originally given in **old Supplementary Figure S6** now has been split in two new supplementary figures. In the **new Supplementary Figure S7**, we show the reduction in VAV2 mRNA levels in the knocked-down cells (these data were originally included in the **old Supplementary Figure S6a**). In the **new Supplementary Figure S8**, we include the quantitation of all the immunohistochemical data shown in the main figures (originally depicted in the **old Supplementary Figure S6b-d**).

Of course, we are open to any suggestion to improve the organization or structure of the manuscript by the Reviewer.

Comment #2. *In Fig 1 the point is made that SCCs overexpress VAV2 both at the RNA and protein level, and this associates with prognosis. A real problem is the lack of information on HPV. Would be important to analyze VAV2 expression in HPV+ve and HPV-ve HNSCC when known, to exclude HPV as confounding factor. Suggestions: b remove EGFR graphs, make x-axis in increments of 12 mths and stop at 60 mths. P-values in max two digits. In c how*

reproducible are these data? In some patients two bands are identified. What is visualized with Ponceau S, the IgG bands?

Authors' response: We respectfully disagree. Indeed, this information was missing in the initial version. However, it was already included in the second version of the manuscript following, in fact, the insightful suggestions made by the three referees after seen the first version of the manuscript. Indeed, the inclusion of this information has been crucial to clearly establish that the deregulation of VAV2 seems to be mainly associated with HPV⁻ hnSCC cases. Information on HPV status can be already found the **old Figures 1, S1, S5e,f, and 8**. We also indicated in the previous version of the manuscript that the two hnSCC patient-derived cells used in our work were HPV⁻ as well. This latter feature has been confirmed upon the characterization of genomes of these two patient-derived cells (see new **Supplementary Figure S6a**).

Regarding the issue of the EGFR graphs, we now present the correlation of survival curves with the levels of VAV2 (**new Figure 1b**, upper panels) and EGFR (**new Figure 1b**, lower panels) mRNAs in separate graphs. With this change, the information is provided in a much cleaner way than before. We do believe that the inclusion of the EGFR (and ACTLA in **Supplementary Figure S1e**) data provides a good reference control given for our work. In any case, we can remove them if the referee believes that these data are not needed.

Following the referee's advice, we have also modified the presentation of the time intervals in the x axes whenever possible in the old **Figures 1b** (right panel; now **new Figure 1b**, upper right panel) and **8f** (now **new Figure 10f**, second panel from left). In the rest of panels is not possible, given that the time intervals used in those analysis do not fit well with the use of the 12 month intervals.

Regarding the *P* values shown in the survival curves, we reduced them down to three digits both in **old Figures 1 and 8a** (now, **new Figure 10a**). Further reductions to two digit values is, in our opinion, an important loss in information for this type of analyses.

Regarding the data shown in **old Figure 1c** (now, **new Figure 2a**), we have performed a total of 83 independent hnSCC samples (some of them with matched healthy controls) that were immunoprecipitated using antibodies to Vav2. As seen in that figure, the immunoprecipitations are rather clean in the majority of cases and identify the expected molecular weight band for endogenous Vav2. The two bands indicated by the referee (exclusively seen in samples #541 and #547) can correspond to either differentially spliced isoforms (that are known to exist in the case of VAV2) or to some spurious band. At this moment we cannot distinguish between both possibilities. However, we would favor the former explanation given that, if they were nonspecific bands, they would have to show up in all the immunoprecipitations shown in this panel. In any case, the quantitation shown in **old Figure 1d** (now, **new Figure 2b**) has been done with the upper band that is detected in all the immunoprecipitations.

In the new data presented in the new version of the manuscript, we also show that the immunoprecipitated VAV2 is tyrosine phosphorylated (see **new Figure 2a,c**). We would like to emphasize that these data have been obtained from human tissues and tumors rather than cell

lines. Despite this, the quality of the immunoprecipitations obtained with the antibody to Vav2 is rather high in our opinion.

Regarding the reproducibility of this finding, it is worth noting that we have carried out an independent analysis of a panel of 232 hnSCCs using standard immunohistochemical analyses as requested in our previous review cycle of this manuscript. This analysis has revealed a percentage of samples with elevated levels of VAV2 similar to those found in the immunoblots.

Finally, the most prominent band seen in the Ponceau staining cannot be the IgG (given that these samples correspond to different aliquots of the same extracts that were used in the immunoprecipitation experiments). Given its abundance, we believe that this band must correspond to the endogenous keratin molecules present in the extracts.

Comment #3. *In Fig 2 the point is made that VAV2^{onc} has a carcinogenic effect on skin and mucosa in mice models. Evidence that overexpression of VAV2 has the same effect as the mutant, is lacking.*

Authors' response: Agree. Addressing this issue in vivo is difficult, given that it would imply the generation of a new mouse model. However, following the advice of Referee #1 and #2, we have investigated whether the overexpression of Vav2^{WT} can also elicit a Vav2^{Onc}-like phenotype in keratinocytes. As shown in the **new Figure 8c,d**, this is in fact the case. This result suggests that both Vav2^{WT} and Vav2^{Onc} can trigger a hyperplastic phenotype when overexpressed in nontransformed keratinocytes. Of course, this phenotype is stronger in the case of Vav2^{Onc}, as expected considering that it is a constitutively active version of the protein.

Conversely, we remind the referee that we have demonstrated that the shRNA-mediated elimination of the endogenous VAV2 mRNA severely affects the fitness of hnSCC patient-derived cells when tested both in 3D organotypic cultures (reduced proliferation, increased differentiation, reduced levels of activation of Vav2-regulated transcriptional factors; **old Figure 7a,f**, now **new Figure 9a,f**) and in in vivo orthotopic transplants (reduced tumorigenesis, reduced metastasis, increased differentiation, reduced levels of activation of Vav2-regulated transcriptional factor; **old Figure 7b,c,d,e-h**, now **new Figure 9b,c,d,e-h**). This is, in our view, a clear demonstration that the endogenous wild-type version of VAV2 does play a role in this tumorigenic process.

Finally, we would like to indicate that during the elaboration of this version of our manuscript a gain-of-function mutation in the VAV2 gene has been found in familial cases of oral SCC (Huang et al. *Cell Discov* 2019, PMID: 31798960). This new observation has given further interest to the analyses carried out with the gain-of-function Vav2^{Onc} mice. Indeed, the results obtained with this mouse knock-in strain suggest that the VAV2 gain-of-function mutation, in combination with other genetic alterations, should have a significant role in the etiology of this type of hereditary cancers. In addition, they provide a pathological framework to understand the contribution of mutant VAV2 to this rare hereditary cancer. This information has been included both in the **Results** (pages 8-9) and **Discussion** (page 28) sections of the new version of the manuscript. It should be noted that, according to the new regulatory mechanism recently established by us for Vav family proteins (Barreira et al, *Sci Signal* 2014; PMID: 24736456), the

CH-Ac deletions (present in Vav2^{Onc}) and the mutation in the most C-terminal domain of Vav2 must induce the same activation effect in the protein.

Comment #4. *In Fig 3, the point is made that VAV2onc cells from mice have an increased proliferative capacity in organotypic models, confirming the in vivo data. Also Rac and Rho mutants show this effect.*

Authors' response: Agree. This is precisely what the data show.

Comment #5. *In Fig 4 the point is made that VAV2onc causes expression profiles associated with dedifferentiation and increased stemness.*

Authors' response: Agree. This is precisely what these results demonstrate.

Comment #6. *In Fig 5 the associated transcription factors that were identified in Fig 4 are further analyzed. Fig 5c and 5d are unclear and what precisely is shown does not become clear from the legend. The point the authors want to make with 5e and 5f is also unclear to me.*

Authors' response: Agree. Sorry for this. We have made changes in the main text and figure legends of the new version of the manuscript to clarify the significance of these experiments.

In the case of **old Figure 5c,d** (now, **new Figure 6e-i**), we wished to demonstrate that the transient expression of Vav2^{Onc}, but not of the catalytically dead Vav2^{Onc+E200A} mutant, does promote the activation of AP1-, c-MYC-, E2F- and TEAD-regulated promoters in keratinocytes. With this approach, we demonstrate that the stimulation of these transcriptional factors takes place downstream of Vav2^{Onc}. The reason to carry out these experiments is indicated in the main text: “To further confirm that the stimulation of these factors is the result of the direct downstream signaling of Vav2^{Onc}, we investigated whether the transient expression of the active version of this GEF could promote the stimulation of these transcriptional factors using luciferase reporter assays in 2D keratinocyte cultures. In agreement with the organotypic data, we found that the transient expression of Vav2^{Onc} leads to the stimulation of the transcriptional activity of the endogenous AP1 factors (**Fig. 6e**), c-MYC (**Fig. 6f**), E2F proteins (**Fig. 6g**), and the YAP/TAZ/TEAD complex (**Fig. 6h**). These effects are severely reduced when using the catalytically dead Vav2^{Onc+E200A} protein (**Fig. 6e to h**). Immunoblot analyses confirmed the proper expression of the ectopically expressed proteins used in these experiments (**Fig. 6i**). These data indicate that Vav2^{Onc} stimulates all these transcriptional factors in a catalysis-dependent and signaling autonomous-manner”.

To further clarify this issue, we have also changed the text in the legend to this figure (page 70): “**(e to h)** Transiently transfected Vav2^{Onc} triggers the rapid activation of endogenous AP1- (**e**), c-MYC- (**f**), E2F- (**g**), and TEAD (**h**) in human keratinocytes. Activity was measured using luciferase-encoding vectors containing promoter regions for each of the interrogated transcriptional factors, as described in Methods. **(i)** Expression of indicated proteins in one of

the experiments performed in panels e to h. Tubulin was used as loading control (bottom panel) ($n = 3$).”

Regarding the issue of **old Figure 5e,f** (now, **new Figure 6j,k**), this figure summarizes the results obtained with our *in silico* analyses indicating that there is a statistically significant correlation between the levels of expression of the VAV2 mRNA and previously characterized c-MYC- and YAP/TAZ/TEAD-regulated gene signatures in hnSCC samples. These data further suggest a link between VAV2 signaling and the activation of both c-MYC and YAP/TAZ/TEAD complexes in hnSCCs. Furthermore, they show again that such correlation is associated with HPV⁻ cases. To clarify this issue, we have modified the main text in the new version of the manuscript. This new version now says (page 17) : “Further linking Vav2 with the stimulation of these factors, we observed using *in silico* analyses that the expression of well-known c-MYC- (**Fig. 6j**) and YAP/TAZ/TEAD-regulated (**Fig. 6k**) gene signatures is directly proportional to the abundance of the VAV2 transcripts in both human cSCC and hnSCC patient samples. The correlation between VAV2 transcript levels and the c-MYC- (**Fig. 6j**, bottom right panel) and YAP/TAZ-regulated (**Fig. 6k**, bottom right panel) gene signatures is lost in most cases when the hnSCC samples are not stratified according to HPV status, further linking this Vav2^{Onc}-driven pathobiological program to HPV⁻ tumor cases.”

We have also modified the legend to this figure to further clarify this issue (page 71): “**(j and k)** Correlation between the levels of the VAV2 mRNA and the expression of c-MYC- (j) and YAP/TEAD-regulated- (k) gene signatures in the indicated (top) cSCC ($n = 40$) and hnSCC ($n = 685$) gene expression datasets.”

Comment #7. *In Fig 6 the authors show with inhibitors that MYC and YAP-TEAD impact the morphological changes of hyperplasia. 6c is insufficiently explained in the legend. The role of MYC is supported by ectopic expression, of YAP/TEAD not.*

Authors’ response: Agree. Sorry for this. We have modified the main text to clarify the meaning of these experiments (which are now shown in the **new Figure 7c**). The new text now indicates (page 18): “To demonstrate the inhibition of the expected targets by the foregoing inhibitors, we evaluated c-MYC and YAP/TAZ/TEAD transcriptional activity in human keratinocyte 2D cultures transiently transfected with the indicated Vav2^{Onc} versions plus luciferase reporter plasmids containing either c-MYC- or TEAD-responsive promoters, respectively. We observed that the addition of 10058-F4 blocks both the basal and the Vav2^{Onc}-driven MYC activity in those cells (**Fig. 7c**, left panel). On the other hand, Verteporfin reduces the basal and the Vav2^{Onc}-triggered transcriptional activity of TEAD in these culture conditions (**Fig. 7c**, right panel).”

We have also modified the legend for the panel of this figure (page 73): “**(c)** Demonstration that the transcriptional activity of c-MYC and TEAD is inhibited by MYC (10058-F4) and YAP/TEAD interaction (Verteporfin) inhibitors, respectively.”

Regarding the issue of testing the effect of the overexpression of MYC and YAP/TEAD, we decided to evaluate only the effect of c-MYC in our system because previous reports indicated quite distinct effects of this transcriptional factor in keratinocytes. Thus, some studies proposed

a proliferative action (which would be consistent with our data). By contrast, other studies proposed a role of c-MYC in the induction of terminal differentiation (these data would be inconsistent with the results found in our study). This issue was clearly explained in the previous version of the manuscript (and in the new one as well, see page 18). Our data clearly show that, in our experimental system, the overexpression of c-MYC favors proliferation rather than terminal differentiation (**old Figure 6d,e**; now **new Figure 7d,e**).

We have not done the same experiments with the overexpression of YAP because it is already well established that this complex promotes epithelial hyperplasia when activated in keratinocytes (see references in main text). Due to this, we did not find the value of repeating again these experiments in our work.

Comment #8. *In Fig 7 inhibition of VAV2 by shRNA is studied. Very confusing that Vav1^{-/-} etc immunocompromised mice were used.*

Authors' response: As indicated in our previous rebuttal letter, we have used the compound Vav1^{-/-};Vav2^{-/-};Vav3^{-/-} mice in these experiments because, in our hands, they work similarly to some of the standard immunocompromised mice in xenograft experiments due to their lymphopenic status. This model is also much more cost-saving for us when compared to the use of standard strains of immunocompromised mice that are commercially available.

We have modified the text in the new version of the manuscript to make this issue clearer to the future readers. The new version now says (page 21): “**To analyze the behavior of VAV2 knockdown PDCs *in vivo*, we transplanted them orthotopically in the tongue of Vav1^{-/-};Vav2^{-/-};Vav3^{-/-} mice. As these mice are lymphopenic (Ref. 38, 39) and generated in our own animal facility, they represent a cheap alternative to the use of the commercial immunodeficient mouse strains typically used for this type of experiments,**”

It is also worth noting that, according to the data obtained with the control PDCs, the Vav family-deficient background does not affect extrinsically the growth and tumorigenic properties of cancer cells.

Comment #9. *In Fig 8 prognostic gene expression profiles related to the VAV2 signaling are identified. To me this is beyond the scope of this paper.*

Authors' response: We respectfully disagree. We believe that these data give useful information for: **(i)** An extra proof in support of the correlation existing between upregulated VAV2 signaling and poor patient prognosis. **(ii)** The identification of a minimal gene signature, connected to a well-defined pathobiological process, that can be potentially used as a diagnostic tool. In addition, we believe that the inclusion of these data adds a nice corollary to the functional data presented in the rest of figures.

Of course, we can remove these data from the final version of the manuscript if the Referee remains adamant about it. However, we would like to indicate that the other two referees did

not make any indication about the lack of interest of this part of our work (now, the **new Figure 10** of the manuscript).

Comment #10. *Despite a lot of experimental work: there is no evidence presented that upregulated VAV2 has the same effect as the mutant, which is the assumption to link the cancer data to the functional data.*

Authors' response: We do understand to some extent the concern raised by the referee regarding the fact that we had not provided formal proof in the manuscript that Vav2^{WT} elicits a phenotype similar to Vav2^{Onc} in keratinocytes. This issue has been solved in the new version of the manuscript, where we clearly show that the ectopic expression of the full-length version of Vav2 also promotes epithelial hyperplasia when tested in 3D organotypic models (**new Figure 8**). It is also worth noting that the key role of endogenous VAV2^{WT} in hnSCC has been also demonstrated in our loss-of-function experiments using both hnSCC patient-derived cells and the SCC-25 cell line (**old Figure 7, now new Figure 9**).

Having said this, we believe that the experiments carried out with Vav2^{Onc} are highly valuable, since they allowed us to address the intrinsic and signaling autonomous role of the phenotypes observed without any other input. Moreover, the interest of using this approach has been further increased upon the recent discovery of a recurrent gain-of-function mutation of VAV2 in familial oral hnSCC cases (Huang et al, *Cell Discov* 2019, PMID: 31798960). In this context, the results obtained with the Vav2^{Onc/Onc} mouse strain used in our current study indicate that this gain-of-function mutation, in combination with other genetic alterations, can probably contribute quite significantly to the etiology of this type of hereditary hnSCC cancers. In addition, our data provide a good pathobiological and functional framework to understand the contribution of VAV2 to the development of this rare familial cancer (this information has been included in the **new Discussion** section, see page 28).

Comment #11. *In addition it is not VAV2-specific as Rac and Rho do the same: hence it seems that GEF activity in general elicit these phenotypes.*

Authors' response: We respectfully disagree. Certainly, the coexpression of active versions of Rac and Rho elicit a Vav2^{Onc}-like phenotype in keratinocytes (**old Figure 3f-h, now new Figure 4f-h**). This is consistent with the observation that the role of Vav2^{Onc} in this process is catalysis-dependent as indicated in our manuscript (**old Figure 3c,d, now new Figure 4c,d**). It is also consistent with the observation that endogenously and ectopically expressed Vav2^{Onc} triggers the activation of these two GTPases in primary mouse and human keratinocytes, respectively (**old Figure 3e, now new Figure 4e**). Further indicating the connection of Vav2 with these two GTPases, we could demonstrate that the hyperplastic phenotype triggered by Vav2^{Onc} is eliminated when the organotypic cultures are treated with either Pak1 or Rock inhibitors (**old Supplementary Figure S2e,f; now new Supplementary Figure S3e,f**). In our view, this is not lack of specificity: it is, in fact, a mechanistic demonstration that these GTPase-regulated pathways are working downstream of Vav2 in this biological process.

We would also like to emphasize that we find the same phenotype in mice under conditions in which the Vav2^{Onc} protein is maintained under normal physiological levels of expression (**old**

Figures 2 and 3, now new Figures 3 and 4, respectively). Whether other GEFs trigger a similar phenotype in vivo remains to be determined. This is, however, beyond the scope of our paper. In any case, against this lack of specificity, we have shown that the depletion of endogenous VAV2^{WT} impairs the tumorigenic features of hnSCC cell lines and patient-derived cells (**old Figure 7, now new Figure 9**). It is difficult to envision how this might occur in the case that many other Rho GEFs would play the same redundant function in keratinocytes.

Finally, we have extra data in the lab, not included in this work, regarding the potential implication of other Rho GEFs in both cSCC and hnSCC. Using a similar approach to those utilized in this manuscript, we have found that there only are three additional GEFs upregulated in these two SCC types and that, only two of them have prognostic value. Our functional data indicate that these GEFs are indeed important for hnSCC tumorigenesis. However, they do not seem to elicit redundant, Vav2^{Onc}-like downstream pathways in keratinocytes and hnSCC patient-derived cells (data unpublished).

In this context, it is worth noting that the lack of functional redundancy of Rho GEFs that are expressed in the same tissues is still a poorly understood issue in this field. As a token, both the initiation and promotion phases of skin tumors can be indistinctly blocked by inhibiting the catalytic activity of Vav2 (*Oncogene*, in press; doi: 10.1038/s41388-020-1353-x; PMID: 32528129) or by depleting the Rac1 GEF Tiam1 (Malliri et al, *Nature* 2002; PMID: PMID: 12075356). Similar observations have been found with other combinations of Rho GEFs in other tumor types (for a review, see Bustelo, *Biochem Soc Trans* 2018; PMID: 29871878). This suggests that each Rho GEF might target distinct GTPase pools in an oncogenic insult-, subcellular localization- or cancer stage-dependent manner. It is also plausible that Rho GEFs could associate with specific signalosomes that could lead to the engagement of different spectra of signaling pathways and biological responses. Clearly, more work is needed to fully understand the specific role of Rho GEFs that share overlapping expression patterns in tumors.

Comment #12. *The main question remaining is how relevant this is for HNSCC in man. The experimental data look convincing, but which important question was answered? There are so many molecules that have essential functions in cells and also tumor cells. There is no data presented that VAV2 is a fantastic therapeutic target in HNSCC or that it is a key pathway in carcinogenesis.*

Authors' response: We respectfully disagree. In our view, we believe that the relevance of this new pathway in human hnSCC is supported by a number of observations: **(i)** Frequent overexpression of both the VAV2 mRNA and the VAV2 protein in HPV⁻ hnSCCs cases. **(ii)** Correlation of the levels of both the VAV2 mRNA and Vav2^{Onc}-regulated gene signatures with the long-term survival of hnSCC patients. **(iii)** Demonstration that the elimination of endogenous VAV2^{WT} leads to reduced proliferation, increased differentiation, and reduced tumorigenesis of HPV⁻ hnSCC patient-derived cells. **(iv)** The recent finding of a recurrent gain-of-function mutation in the VAV2 gene in familial oral hnSCC (Huang et al, *Cell Discov* 2019, PMID: 31798960), as now discussed in the new version of our manuscript (see **new Discussion** section, page 28).

In the first version of this manuscript, we also demonstrated using a knock-in mouse strain that the expression of a catalytic hypomorphic Vav2 mutant eliminates skin tumorigenesis while preserving normal Vav2-dependent physiological functions in mice. These data, which this referee requested to be removed out from our work in the first reviewing cycle of the manuscript, also support the hypothesis that the inhibition of the catalytic activity of Vav2 could be a valuable therapeutic approach for these type of tumors. These data have been recently accepted for publication in *Oncogene* (in press, doi: [10.1038/s41388-020-1353-x](https://doi.org/10.1038/s41388-020-1353-x); PMID: [3252812](https://pubmed.ncbi.nlm.nih.gov/3252812/)). These data, together with the negative effects induced by the VAV2 knockdown in hnSCC patient-derived and SSC-25 cells, suggest in our opinion that VAV2 can be considered a potential therapeutic target (although perhaps not a “fantastic” one as the referee indicates). Of course, we will have to wait to get chemical inhibitors to be able to eventually approach this issue.

Regardless of the therapeutic interest of Vav2, we honestly believe that the data presented in this work offers a comprehensive view of the role of Vav2 in the context of both naïve and fully transformed keratinocytes that, in addition, provides interesting links to human pathology. As such, we believe that these data do have an intrinsic interest for the readership of *Nat Commun*.

REVIEWER #3

Building on mouse and human models, the manuscript by Lorenzo-Martin et al provides compelling evidence that increased VAV2 activity promotes squamous carcinogenesis in skin and oral epithelia. The authors have adequately addressed my concerns with the initial version of the manuscript by (a) adding new experimental data; and (b) clarifying some sections in the text. I believe this study is solid, interesting, well presented and suitable for publication in Nature Communications.

Authors' response: Thank you very much for your comments and the positive opinion of the new revised version. We also appreciate it very much your prior comments, which helped us significantly to improve the information contained in our first submission.

REVIEWERS' COMMENTS:

Reviewer #1 (Remarks to the Author):

The authors should be commended for their efforts in addressing all major and minor issues raised during the review process. The study has been certainly strengthened with the exciting and rigorous new data.

Reviewer #2 (Remarks to the Author):

Lorenzo-Martin revised their manuscript. It continuously improves including the readability. Readability may further increase when edited by a native speaker. I still have a few comments.

1) The Abstract starts with: 'Regenerative proliferation and cell undifferentiation are histological features linked to poor ...' I doubt whether the term 'cell undifferentiation' exists. I would revise to 'Regenerative proliferation capacity and poor differentiation are histological features linked to unfavorable ...' Second sentence I would change 'oral locations' to 'oral mucosa'.

2) Second sentence Introduction "limited availability of therapeutic tools" 'should be limited efficacy of therapeutic tools'.

3) Also in the Introduction: correct spelling is human papillomavirus

4) Fig 2. The authors indicated in their answer to my question that the Ponceau S band is likely keratin. Keratin is an intrinsically poor loading control, particularly in the context of well and poor differentiated squamous tissues. The authors should indicate that in the legend of 2b. I do believe the upregulation in HNSCC but question the quantitation. I Further miss an association of VAV2 staining with differentiation grade: poor, moderate and well-differentiated. The authors have data of 232 cases available. These were all HPV-negative? In legend 2f IVL and LOR shouldd be explained.

5) Fig 8/9 I would suggest to bring the molecular characterization of the HNSCC cell lines to the supplementary. It distracts from the main message.

COMMENTS TO REFEREES
MANUSCRIPT NCOMMS-19-07108B

REVIEWER #1

General Comment. *The authors should be commended for their efforts in addressing all major and minor issues raised during the review process. The study has been certainly strengthened with the exciting and rigorous new data.*

Authors' response: Thank you for your kind comments.

REVIEWER #2

Comment #1. *The Abstract starts with: 'Regenerative proliferation and cell undifferentiation are histological features linked to poor ...' I doubt whether the term 'cell undifferentiation' exists. I would revise to 'Regenerative proliferation capacity and poor differentiation are histological features linked to unfavorable ...' Second sentence I would change 'oral locations' to 'oral mucosa'.*

Authors' response: Agree. These modifications have been made in the new version of the manuscript (see page 2).

Comment #2. *Second sentence Introduction "limited availability of therapeutic tools" 'should be limited efficacy of therapeutic tools'.*

Authors' response: Agree. This modification has been incorporated to the new version of the manuscript (see page 3).

Comment #3. *Also in the Introduction: correct spelling is human papillomavirus*

Authors' response: Agree. The suggested correction has been made (see page 3 of new manuscript version).

Comment #4a. *Fig 2. The authors indicated in their answer to my question that the Ponceau S band is likely keratin. Keratin is an intrinsically poor loading control, particularly in the context of well and poor differentiated squamous tissues. The authors should indicate that in the legend of 2b. I do believe the upregulation in HNSCC but question the quantitation.*

Authors' response: The determination of the total amount of protein used in these experiments is not based on the levels of this highly expressed band (which can be keratin or any other abundant protein since we have not identified it). As indicated in the Methods section of our

manuscript (pages 34-35), we have determined the total amount of protein used in the immunoprecipitations using two complementary methods. Firstly, we quantified the protein concentration in the tissue lysates using the Bradford technique prior to the immunoprecipitation step. Based on that quantitation, we then used aliquots of the tissue extracts (each containing 400 µg of total protein) to carry out the immunoprecipitation with antibodies to Vav2. Secondly, the total amount of protein present in each tissue lysate was rechecked by running aliquots of the total cellular extracts used in the immunoprecipitation experiments in SDS-PAGE gels and, after transfer to nitrocellulose filters, staining with Ponceau S solution. To avoid confusions about this, we now have included images of the whole Ponceau S-stained filters in **Figure 2a**. We have also included the information about this issue in the main text (see page 9-10).

Comment #4b. *I further miss an association of VAV2 staining with differentiation grade: poor, moderate and well-differentiated.*

Authors' response: Agree. We have found that VAV2 staining does not significantly correlate with the differentiation grade in our patient cohort. However, it is worth mentioning that VAV2 staining was consistently absent in all terminal differentiation areas of the samples (e.g., keratin pearls) even in the case of VAV2-positive tumors (an example of this is given in the **new Figure 2e**). This new information has been included in the main text of the new version of the manuscript (page 10).

Comment #4c. *The authors have data of 232 cases available. These were all HPV-negative?*

Authors' response: Most samples (96.55%) used in these experiments were HPV⁻. Only 8 out of the 232 samples utilized were HPV⁺ in fact. This information has been included both in the main text (page 10) and in the Methods section (page 38). The protocols used to determine HPV status in the tumor samples have also been included in the Methods section (page 34).

Comment #4d. *In legend 2f IVL and LOR should be explained.*

Authors' response: Agree. *IVL* and *LOR* are the recognized symbols for involucrin and loricrin, two transcripts that are associated with specific moments of the terminal differentiation of keratinocytes. This info has been included in the new version of the legend to this figure (now, new **Fig. 2g**; page 64). We have also included this information in the main text (page 10).

Comment #5. *Fig 8/9 I would suggest to bring the molecular characterization of the HNSCC cell lines to the supplementary. It distracts from the main message.*

Authors' response: Agree. This information has been transferred to the **Supplementary Note 1** in the new version of the manuscript.